# Structure-Aware Fusion with Progressive Injection for Multimodal Molecular Representation Learning

**Zihao Jing**[1], **Yan Sun**[1], **Yan Yi Li**[2], **Sugitha Janarthanan**[2], **Alana Deng**[1], **Pingzhao Hu**[1,2*]

[1]Department of Computer Science, Western University, London, ON, Canada
[2]Department of Biochemistry, Western University, London, ON, Canada
zjing29@uwo.ca, phu49@uwo.ca

## Abstract

Multimodal molecular models often suffer from 3D conformer unreliability and modality collapse, limiting their robustness and generalization. We propose **MuMo**, a structured **mu**ltimodal fusion framework that addresses these challenges in **mo**lecular representation through two key strategies. To reduce the instability of conformer-dependent fusion, we design a Structured Fusion Pipeline (SFP) that combines 2D topology and 3D geometry into a unified and stable structural prior. To mitigate modality collapse caused by naive fusion, we introduce a Progressive Injection (PI) mechanism that asymmetrically integrates this prior into the sequence stream, preserving modality-specific modeling while enabling cross-modal enrichment. Built on a state space backbone, MuMo supports long-range dependency modeling and robust information propagation. Across 29 benchmark tasks from Therapeutics Data Commons (TDC) and MoleculeNet, MuMo achieves an average improvement of 2.7% over the best-performing baseline on each task, ranking first on 22 of them, including a 27% improvement on the LD50 task. These results validate its robustness to 3D conformer noise and the effectiveness of multimodal fusion in molecular representation. The code is available at: github.com/selmiss/MuMo.

## 1 Introduction

Molecular property prediction is fundamental to computational chemistry and drug discovery, offering a cost-effective alternative to experimental screening. According to the Tufts Center for the Study of Drug Development, developing a new drug costs over $2.6 billion on average, with much of this attributed to early-stage trial inefficiencies [Chatterjee, 2015].

Accurate silico prediction substantially reduces time and cost by early elimination of suboptimal candidates [Graff et al., 2021]. To improve prediction, recent advances in molecular representation learning have explored large-scale pretraining or multimodal architectures, typically combining SMILES [Weininger, 1988], 2D graphs, and 3D geometries. Sequence-based models [Fabian et al., 2020, Ross et al., 2022] leverage mature language modeling but often miss structural detail, while 3D-aware models [Stärk et al., 2022] capture geometric context at the cost of scalability and stability. These limitations highlight the necessity for an efficient and structure-aware fusion framework.

Specifically, we identify two key challenges in molecular representation learning: **(1) Conformer-dependent fusion is unreliable.** First, conformers generated by tools like RDKit often differ significantly in local arrangement even with the same molecule. As shown in Figure 1(a), two RDKit-generated conformers exhibit clear geometric differences in the rotation and orientation of terminal groups, despite sharing identical 2D topology and SMILES string. These conformers may present

---

*Corresponding author. Department of Computer Science, Western University, 1400 Western Road, London, Ontario N6G 2V4, Canada. E-mail: phu49@uwo.ca (P.H.)

39th Conference on Neural Information Processing Systems (NeurIPS 2025).

different surface areas or spatial constraints, leading to changes in predicted properties [Adams and Coley, 2025, Brethomé et al., 2019]. Second, some different molecules share nearly identical embeddings, making them difficult to distinguish. Figure 1(b) illustrates this conformer sensitivity using two drugs (Ibuprofen and Ketoprofen). Despite being chemically distinct, their conformer embeddings from DimeNet [Gasteiger et al., 2020] exhibit considerable overlap in Principal Component Analysis (PCA) space, indicating the risk that existing embedding methods fail to distinguish between structurally similar yet functionally distinct molecules due to conformational noise.

**(2) Modality collapse stems from naive fusion.**
In many multimodal models, different modalities are treated as equally important and are fused in the same phase using simple operations such as early concatenation or token-level attention. This is based on an untenable assumption that all modalities are clean and semantically aligned. However, in molecular data, 3D inputs are often noisy, and different modalities (e.g., geometry and SMILES) operate at distinct levels of abstraction. These can lead to modality collapse, where the 3D signal dominates or distorts the information from other modalities [Su et al., 2020, Li et al., 2020]. Prior studies in vision-language and chemistry [Rong et al., 2020, Zeng et al., 2023] also observe that symmetric fusion often leads to unstable optimization or degraded generalization. These findings inspire a shift toward asymmetric fusion, allowing for precise and properly timed information exchange between modalities.

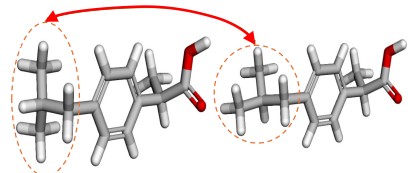

(a) Two conformers show local 3D variation

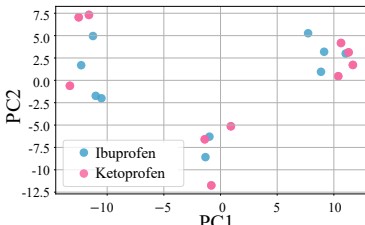

(b) Conformers reveals PCA embedding instability

Figure 1: Illustration of Limitations in molecular representation learning.

To address these challenges, we propose **MuMo**, a **mu**ltimodal fusion model for **mo**lecular representation learning with two key components. To mitigate the unreliability of conformer-dependent fusion, we introduce a Structured Fusion Pipeline (SFP) that combines 2D and 3D inputs, as well as local and global information, into a unified and aligned graph representation. It serves as a stable structural prior for the subsequent inference. To mitigate modality collapse from naive fusion, we propose a Progressive Injection (PI) mechanism that asymmetrically integrates the fused structural prior into the main sequence stream, while preserving the independent propagation and evolution of the modality information throughout the model to grasp the long-range dependencies in complex molecules. Together, these enable MuMo to model molecules into a consistent representation in a robust and structure-aware manner. We summarize our key contributions as follows:

- We propose a Structured Fusion Pipeline that aligns and encodes 2D and 3D inputs into a unified and stable structural prior, addressing the inconsistency of conformer-dependent modeling.

- We introduce a Progressive Injection mechanism that asymmetrically integrates structural prior into the mainstream, mitigating modality collapse caused by inappropriate fused 3D signals.

- Our MuMo ranks top on 22 out of 29 molecular tasks across fusion baselines, 3D-heavy models, and larger pretrained models, averagely outperforming previous methods by 2.7%, even up to 27% on the LD50 dataset, showing the leading practical value in molecular property prediction.

## 2  Related Work

**Single-modality molecular models.** Sequence-based models such as MolBERT [Fabian et al., 2020], MoLFormer [Ross et al., 2022], and ChemBERTa [Chithrananda et al., 2020] treat SMILES as language and leverage Transformer pretraining, but lose structural fidelity due to serialization. Graph neural networks like GCN [Kipf and Welling, 2016], HiGNN [Zhu et al., 2022], and FPGNN [Cai et al., 2022] preserve 2D connectivity, capturing local atomic patterns but lacking geometric awareness. 3D models, including SchNet [Schütt et al., 2018], DimeNet [Gasteiger et al., 2020], and Uni-Mol [Zhou et al., 2023], encode spatial coordinates and bond angles, but depend on force field-derived conformers [Oliveira et al., 2020], being expensive and fragile for large and flexible molecules.

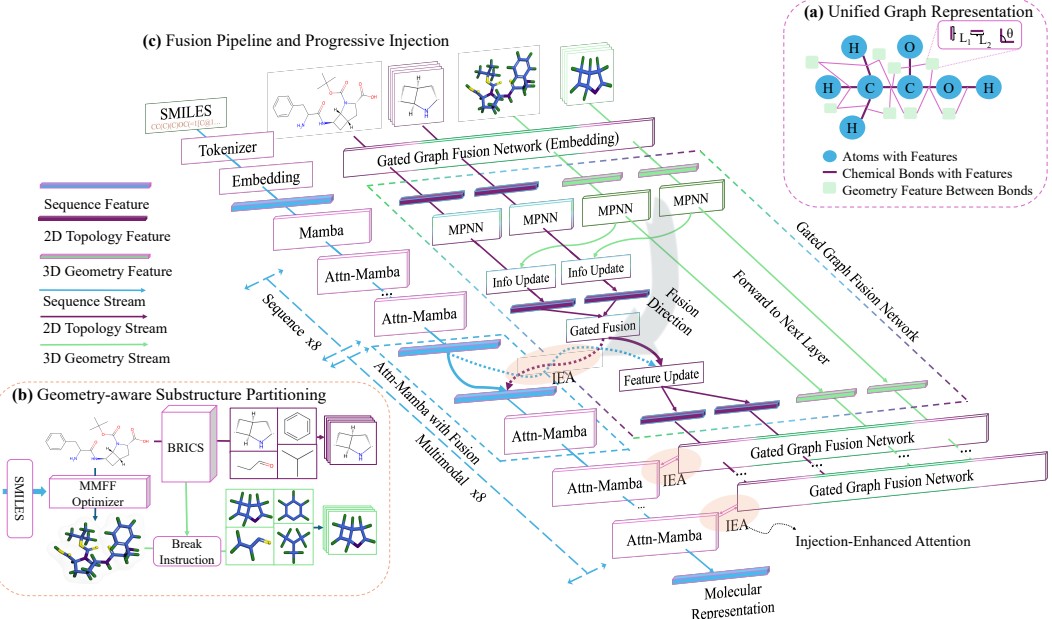

Figure 2: Overview of the **MuMo** architecture. (a) Structural Unified Representation for 2D/3D modalities encoding, (b) Substructure Partitioning for multiscale molecular feature, (c) Fusion Pipeline (right) of 2D topology & 3D geometric priors, and Progressive Injection to integrate cross-modal structural information into the main sequence (left).

**Multimodal and pretrained models.** Several models explore the fusion of multiple molecular modalities. GraphMVP [Liu et al., 2022] and MolCLR [Wang et al., 2022] combine 2D graphs with 3D conformers using contrastive pretraining, while Uni-Mol [Zhou et al., 2023] integrates topological and geometric features via coordinate-aware graph encoders. These models typically adopt symmetric fusion strategies, which can entangle noisy 3D signals and suffer from conformer perturbation. Separately, large-scale pretrained models such as ChemBERTa [Chithrananda et al., 2020], TranFoxMol [Gao et al., 2023], and MLM-FG [Peng et al., 2024] focus on SMILES-only inputs, improving sequence modeling through masked prediction or fragment-aware encoding. However, they lack explicit mechanisms for integrating structural priors, and their performance is often tied to scale rather than architectural robustness. We also discussed the connection between reused modules and our contributions in Appendix B.1 and B.2.

## 3 MuMo: Structured Fusion and Progressive Injection

We present MuMo, a molecular representation framework comprising two core components. Section 3.1 provides an overview of the MuMo architecture. Section 3.2 and 3.3 introduce the Structured Fusion Pipeline and the Progressive Injection mechanism, which address the inconsistency of conformer-dependent fusion and the modality collapse caused by naive integration.

### 3.1 Overview of MuMo Architecture

The fusion pipeline of MuMo consists of two key pathways, as illustrated in Figure 2(c). (1) Structural modality fusion stream (Section 3.2): The 2D molecular graph (purple lines) and 3D geometric information (green lines) are jointly fused to generate unified structural representations, which evolve through independent propagation. (2) Semantic sequence stream (Section 3.3): The SMILES sequence is first tokenized and embedded, being input into the stacked modeling blocks as the main stream (blue line). And the structural modality information stream will be injected into the main stream at later layers for subsequent inference, and generate the ultimate molecular representation.

During inference, MuMo introduces the progressive injection mechanism to selectively integrate the structural priors from the modal fusion stream to the main sequence stream. Unlike symmetric fusion strategies, the structural priors are treated as auxiliary guidance and are asymmetrically injected via dedicated attention layers (purple dashed arrows). This allows the sequence to first establish its own contextual semantics before receiving structural guidance, avoiding signal distortion while effectively

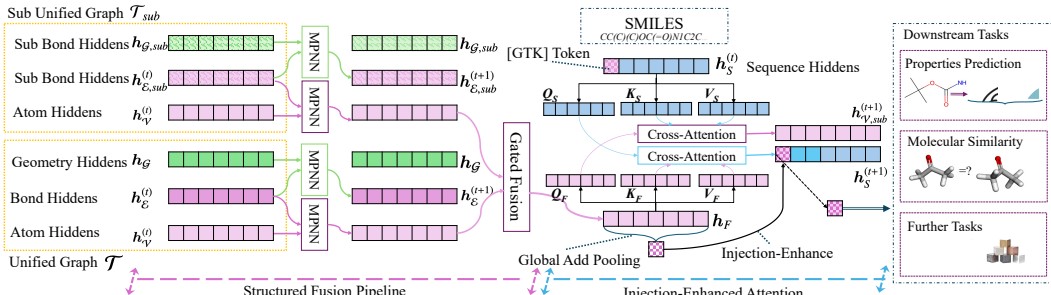

Figure 3: Multimodal Fusion. It illustrates how structural modalities (2D and 3D) are fused via the Structured Fusion Pipeline (SFP), then injected into the SMILES sequence stream through the Injection Enhanced Attention (IEA, within PI) module.

enriching the multimodal structural information. The final molecular representation is derived from the structurally enhanced sequence output.

## 3.2 Structured Fusion Pipeline for 2D and 3D Modalities Integration

To establish a unified molecular structure, we propose a graph-based representation that seamlessly integrates both 2D typological, 3D geometric information at both local and global scales in the graph, serving as a joint structural prior for guiding sequential semantic modeling.

### 3.2.1 Structural Unified Graph Representation for 2D and 3D Modalities

To jointly represent and process multimodal molecular data, MuMo introduces a Unified Graph formulation, a two-step message passing mechanism, and a batch-aware graph aggregation scheme.

**Unified Graph Structure:** To better integrate molecular information in 2D and 3D modalities, we define a Unified Graph structure $\mathcal{T} = (\mathcal{V}, \mathcal{E}, \mathcal{G})$, where $\mathcal{V}$ denotes atoms (nodes), $\mathcal{E}$ chemical bonds (edges), and $\mathcal{G}$ auxiliary geometric linkages. As illustrated in Figure 2(a), each atom $v_i \in \mathcal{V}$ is associated with a feature vector $v_i \in \mathbb{R}^{d_V}$, and each bond $e_{ij} \in \mathcal{E}$ with $e_{ij} \in \mathbb{R}^{d_E}$. The geometric linkage set $\mathcal{G}$ encodes spatial relations between edge pairs $(e_{ij}, e_{jk})$ sharing a central atom $v_j$, represented as a triplet $g_{ij,k} = (l_{ij}, l_{jk}, \theta_{ij,k})$, where $l_{ij} = \|\boldsymbol{p}_i - \boldsymbol{p}_j\|_2$ is the Euclidean bond length and $\theta_{ij,k}$ the angle between bond vectors. To guarantee geometric consistency across conformations, we formally prove the rotational invariance of the proposed representation in Appendix A.1.

**Message Updating in the Unified Graphs:** As Figure 3 (left) shows, to jointly propagate 2D and 3D geometric information, we perform a two-step message updating throughout the graph. At each iteration $t$, edge-centered and node-centered updates are computed as follows, here $\boldsymbol{h}_{e_{ij}}^{(t)}$, $\boldsymbol{h}_{v_i}^{(t)}$, and $\boldsymbol{h}_{g_{ij,k}}$ denote the hidden states of edges, nodes, and geometric descriptors, respectively:

$$\boldsymbol{m}_{e_{ij}}^{(t+1)} = \sum_{e_{jk} \in N_{\mathcal{E}}(e_{ij})} Message_{\mathcal{E}}^{(t)}(\boldsymbol{h}_{e_{ij}}^{(t)}, \boldsymbol{h}_{e_{jk}}^{(t)}, \boldsymbol{h}_{g_{ij,k}}), \quad (1)$$

$$\boldsymbol{h}_{e_{ij}}^{(t+1)} = Update_{\mathcal{E}}^{(t)}(\boldsymbol{h}_{e_{ij}}^{(t)}, \boldsymbol{m}_{e_{ij}}^{(t+1)}), \quad (2)$$

$$\boldsymbol{m}_{v_i}^{(t+1)} = \sum_{v_j \in N_{\mathcal{V}}(v_i)} Message_{\mathcal{V}}^{(t)}(\boldsymbol{h}_{v_i}^{(t)}, \boldsymbol{h}_{v_j}^{(t)}, \boldsymbol{h}_{e_{ij}}^{(t)}), \quad (3)$$

$$\boldsymbol{h}_{v_i}^{(t+1)} = Update_{\mathcal{V}}^{(t)}(\boldsymbol{h}_{v_i}^{(t)}, \boldsymbol{m}_{v_i}^{(t+1)}), \quad (4)$$

the final node embeddings $\boldsymbol{h}_{v_i}$ are subsequently passed to the sequence stream via the PI (Progressive Injection) detailed in Section 3.3.1.

---

**Algorithm 1** Unified Batching Scheme

**Input:** List of Unified Graphs $\{\mathcal{T}_1, \ldots, \mathcal{T}_N\}$;
$\quad \mathcal{T}^{(batch)} \leftarrow new(\mathcal{T}), \delta_v \leftarrow 0, \delta_e \leftarrow 0$
**Output:** $\mathcal{T}^{(batch)}$
1: **for** $k = 1$ to $N$ **do**
2: $\quad$ /* (1) Merge Entity Features */
3: $\quad \mathcal{V}^{(batch)} \leftarrow \mathcal{V}^{(batch)} \cup \mathcal{T}_k.\mathcal{V}$
4: $\quad \mathcal{E}^{(batch)} \leftarrow \mathcal{E}^{(batch)} \cup \mathcal{T}_k.\mathcal{E}$
5: $\quad \mathcal{G}^{(batch)} \leftarrow \mathcal{G}^{(batch)} \cup \mathcal{T}_k.\mathcal{G}$
6: $\quad$ /* (2) Adjust Constraints */
7: $\quad C_{\mathcal{E}}^{(batch)} \leftarrow C_{\mathcal{E}}^{(batch)} \cup \{(i + \delta_v, j + \delta_v) \mid (i, j) \in \mathcal{T}_k.C_{\mathcal{E}}\}$
8: $\quad C_{\mathcal{G}}^{(batch)} \leftarrow C_{\mathcal{G}}^{(batch)} \cup \{(Idx(e_{ij}) + \delta_e, Idx(e_{jk}) + \delta_e) \mid \{Idx(e_{ij}), Idx(e_{jk})\} \in \mathcal{T}_k.C_{\mathcal{G}}\}$
9: $\quad$ /* (3) Update Offsets */
10: $\quad \delta_v \leftarrow \delta_v + |\mathcal{T}_k.\mathcal{V}|, \delta_e \leftarrow \delta_e + |\mathcal{T}_k.\mathcal{E}|$
11: **end for**
12: **Batch Idx:** $\mathcal{T}^{(batch)}.Idx \leftarrow [k \cdot \mathbf{1}_{|\mathcal{T}_k.\mathcal{V}|}]_{k=1}^N$
13: **return** $\mathcal{T}^{(batch)}$

---

**Unified Batching Scheme:** To enable batch processing of Unified Graph, we adopt a unified batching scheme that merges $N$ Unified Graphs $\{\mathcal{T}_1, \ldots, \mathcal{T}_N\}$ into a single batched graph $\mathcal{T}^{(batch)}$, as outlined in

Algorithm 1. This process proceeds in three stages: (1) merging entity features (e.g., node and edge attributes) from individual graphs, (2) adjusting constraint sets, including topological and geometric linkages, to ensure consistency across graphs, and (3) updating node and edge index offsets to preserve intra-graph referential integrity. This process aggregates constraint sets into global representations. The resulting batched graph supports vectorized message passing while maintaining the structural integrity of each constituent molecule. Given a 6 Å distance graph, multi-layer message passing allows implicit reconstruction of dihedral angles (Appendix A.3).

### 3.2.2 Geometry-Aware Substructure Partitioning for Multiscale Representation

Conventional molecular segmentation typically relies on 2D topology or SMILES sequences, often neglecting the rich spatial information in 3D conformers. To address this, we adopt a geometry-aware substructure partitioning strategy that incorporates 3D cues to refine molecular decomposition.

**Graph Partitioning:** We extend Breaking of Retrosynthetically Interesting Chemical Substructures (BRICS) rules [Seo et al., 2023] (details in Appendix B.2) to spatial graphs by severing a subset of topological edges $\mathcal{E}_{\text{cut}} \subset \mathcal{E}_{\text{global}}$ identified from the fused Unified Graph $\mathcal{T}_{\text{global}} = (\mathcal{V}, \mathcal{E}_{\text{global}}, \mathcal{G}_{\text{global}})$. We retain the remaining connectivity and angular constraints to construct a segmented graph $\mathcal{T}_{\text{sub}}$ composed of a batch of $N$ disconnected subgraphs.

$$\mathcal{E}_{\text{sub}} = \mathcal{E}_{\text{global}} \setminus \mathcal{E}_{\text{cut}}, \quad \mathcal{G}_{\text{sub}} = \mathcal{G}_{\text{global}} \setminus \mathcal{G}_{\text{cut}}, \quad \text{Batch}(\{\mathcal{T}_{\text{sub}}^n\}) = \mathcal{T}_{\text{sub}}. \tag{5}$$

**Message Updating in Multiscale Graphs:** To extract multiscale structural representations, we perform message passing over both the global graph $\mathcal{T}_{\text{global}}$ and the segmented graph $\mathcal{T}_{\text{sub}}$. The resulting node embeddings $h_{\mathcal{V}}$ from the global view and $h_{\mathcal{V},\text{sub}}$ from the substructures are then combined via a gated fusion scheme. This fusion adaptively balances coarse-grained global semantics and fine-grained local features:

$$\beta = \sigma\left(\varphi\left(\text{concat}[h_{\mathcal{V}}, h_{\mathcal{V},\text{sub}}, h_{\mathcal{V}} - h_{\mathcal{V},\text{sub}}]\right)\right), \tag{6}$$

$$h'_{\mathcal{V}} = \beta \cdot h_{\mathcal{V}} + (1 - \beta) \cdot \phi\left(\text{concat}[h_{\mathcal{V}}, h_{\mathcal{V},\text{sub}}]\right), \tag{7}$$

where $\varphi(\cdot)$ and $\phi(\cdot)$ are learnable transformations.

**Aggregation of Unified Structural Prior:** We combine multiscale structural signals into a unified structural prior for subsequent injection. Specifically, the fused node representations $h'_{\mathcal{V}}$ are aggregated into a global structural prior representation, which encapsulates both global and local structural features. This structural prior is subsequently used to enhance the semantic modeling stream via PI (Progressive Injection) in Section 3.3.

## 3.3 Progressive Injection for Structural Prior and Sequence Fusion

Given the fused structural prior in Section 3.2, we then integrate it into the sequence stream without disrupting the dominant modality-specific semantics. To this end, we introduce the **Progressive Injection (PI)** to asymmetrically inject structural information into a designated token rather than performing complete fusion. Next, the **Structural Prior Evolution** mechanism propagates the structural information independently across layers to enable a high-level semantic awareness.

### 3.3.1 Injection Enhanced Attention for Structural Priors and Sequence Stream

As shown in Figure 3 (right), injection Enhanced Attention (IEA) performs as the core module in PI in each fusion layer, which integrates structural priors into the SMILES sequence stream through three sequential operations: priors extraction, sequence and structure alignment, and global semantic injection. then

**Step 1: Extract structural priors from the Unified Graph.** As shown in Algorithm 2, we begin by extracting node embeddings $h_{\mathcal{V}}^{(t)}$ from the batched graph $\mathcal{T}^{(batch)}$, and unbatch them into per-graph node features $h_F^{(t)}$ (line 1–2), which serve as structural priors. In parallel, the SMILES sequence is represented as $h_S^{(t)}$ for subsequent semantic modeling.

**Step 2: Align sequence and structure via cross-attention.** We first apply self-attention over $h_S^{(t)}$ to capture intra-sequence context (line 5), then perform bidirectional cross-attention: structure attends to sequence (line 6), and sequence attends to structure (line 7). The updated structural representation $h_F^{(t+1)}$ is rebatched into graph format $h_{\mathcal{V}}^{(t+1)}$ (line 8).

**Step 3: Inject structural prior into the global anchor token.** To inject the fused structure into the sequence stream, we aggregate $h_{\mathcal{V}}^{(t+1)}$ into a pooled prior $h_{\mathcal{V}}^{pooled}$ (line 9), which is added to the [GTK] (global token) via a residual update (line 10). The updated graph $\mathcal{T}'^{(batch)}$ is then passed forward (line 11) for independent evolution for a higher-level perception, which we discuss in Section 3.3.2.

We adopt a staged modeling strategy that separates initial sequence encoding from cross-modal integration. In the early layers, the Mamba backbone operates exclusively on SMILES tokens, capturing intrinsic sequence-level semantics without external interference. In the latter part of the model, we progressively inject the fused structural prior into the sequence stream. This delayed injection allows the model to establish stable token representations before being modulated by structural information, preserving modality autonomy and improving convergence.

---

**Algorithm 2** Injection Enhanced Attention

**Input:** Sequence hiddens $h_S^{(t)}$, graph $\mathcal{T}^{(batch)}$.

**Output:** $h_S^{(t+1)}, \mathcal{T}'^{(batch)}$

    /* Step1: Prior Extraction */

1: $h_{\mathcal{V}}^{(t)} \leftarrow \mathcal{T}^{(batch)}.\mathcal{V}$

2: $h_F^{(t)} \leftarrow \text{Unbatch}(h_{\mathcal{V}}^{(t)}, \mathcal{T}^{(batch)}.\boldsymbol{batch})$

3: $\boldsymbol{Q}_S, \boldsymbol{K}_S, \boldsymbol{V}_S \leftarrow \text{Linear}(h_S^{(t)})$

4: $\boldsymbol{Q}_F, \boldsymbol{K}_F, \boldsymbol{V}_F \leftarrow \text{Linear}(h_F^{(t)})$

    /* Step2: Sequence and Structure Alignment */

5: $h_S^{(t)} \leftarrow \text{SelfAttention}(\boldsymbol{Q}_S, \boldsymbol{K}_S, \boldsymbol{V}_S)$

6: $h_F^{(t+1)} \leftarrow \text{CrossAttention}(\boldsymbol{Q}_F, \boldsymbol{K}_S, \boldsymbol{V}_S)$

7: $h_S^{(t+1)} \leftarrow \text{CrossAttention}(\boldsymbol{Q}_S, \boldsymbol{K}_F, \boldsymbol{V}_F)$

8: $h_{\mathcal{V}}^{(t+1)} \leftarrow \text{Batch}(h_F^{(t+1)}, \mathcal{T}^{(batch)}.\boldsymbol{batch})$

    /* Step3: Prior Injection */

9: $h_{\mathcal{V}}^{pooled} \leftarrow \text{GlobalPool}(h_{\mathcal{V}}^{(t+1)}, \mathcal{T}^{(batch)}.\boldsymbol{batch})$

10: $h_{S,[GTK]}^{(t+1)} \leftarrow \text{Norm}(h_{S,[GTK]}^{(t+1)} + \alpha h_{\mathcal{V}}^{pooled})$

11: $\mathcal{T}'^{(batch)}.\mathcal{V} \leftarrow h_{\mathcal{V}}^{(t+1)}$

12: **return** $h_S^{(t+1)}, \mathcal{T}'^{(batch)}$

---

### 3.3.2 Structural Prior Evolution by State Space Propagation

To further enhance the global perception of structural representations, we propose an evolution strategy based on state space modeling. Inspired by the well-known characteristic of neural networks that shallow layers typically capture local textures while deeper layers learn higher-level semantics, we allow the fused 2D and 3D modal signals to propagate independently across network layers.

We adopt Mamba blocks as the backbone for temporal consistency and continuous evolution of structural priors through state-space dynamics. Unlike Transformers, which rely on layerwise token-to-token attention, Mamba maintains a recurrent latent state that evolves across layers. This architecture naturally accommodates injected priors $g^{(t)}$, regulating the state trajectories driven by inputs without directly disrupting the interactions among local tokens. Consequently, structural priors are seamlessly integrated throughout the semantic modeling stream. At each layer $t$, we maintain a latent state $z^{(t)}$ the state-space update at each layer is:

$$z^{(t+1)} = \boldsymbol{A}z^{(t)} + \boldsymbol{B}_s h_s^{(t)} + \boldsymbol{B}_g g^{(t)}, \quad h_s^{(t+1)} = \boldsymbol{C}z^{(t+1)} + \boldsymbol{D}h_s^{(t)}, \tag{8}$$

where $h_s^{(t)}$ is the sequence state, $g^{(t)}$ is the structural prior, and $\boldsymbol{A}, \boldsymbol{B}_s, \boldsymbol{B}_g, \boldsymbol{C}, \boldsymbol{D}$ are learnable parameters. Our evolution scheme by latent recurrence allows structural priors to persist and influence downstream layers in a controlled, interpretable manner. Therefore, it gradually increases the receptive field and enables progressive abstraction of structural features.

## 4 Experiments & Downstream Analysis

In this section, we conduct comprehensive experiments to evaluate the performance, robustness, and consistency of MuMo across diverse molecular tasks. We pretrained MuMo on the ChEMBL-1.6M dataset [Gaulton et al., 2012] via masked language modeling (MLM), followed by task-specific fine-tuning (see Appendix C.4). In addition, we present ablation studies and visualization analysis to show the contribution of each component in enhancing the multimodal integration and improving the overall quality of molecular prediction. Extended experiments and ablation studies can be found in Appendix C and Appendix D, respectively.

### 4.1 Datasets and Baselines

**Datasets**. To evaluate performance and generalization ability, we benchmark MuMo on 29 tasks from three widely used platforms: 14 from the TDC [Huang et al., 2021], which provides rigorous

Table 1: Results on selected benchmark tasks from TDC and MoleculeNet. We report AUROC for classification (↑) and MAE/RMSE for regression (↓) tasks. We provide the results of "mean$_{std}$" over 5 runs. The top 2 scores per task are highlighted in pink. "Tox-Avg" and "CYP-Avg" indicate average AUROC over {DILI, hERG, Ames} and {CYP2C9-I, CYP2D6-I, CYP3A4-I}, respectively. Notably, MolBERT does not natively support multi-objective tasks (SIDER, TOX21).

| Models | BBB | HIA | Pgp | Bioav. | Tox-Avg. | CYP-Avg. | Top2Cnt/10 |
|---|---|---|---|---|---|---|---|
| TDC Datasets - Classification - AUROC ↑ | | | | | | | |
| AttentiveFP | $0.855_{0.011}$ | $0.974_{0.007}$ | $0.892_{0.012}$ | $0.632_{0.039}$ | $0.842_{0.010}$ | $0.749_{0.008}$ | 0 |
| FPGNN | $0.888_{0.018}$ | $0.958_{0.012}$ | $0.930_{0.007}$ | $0.666_{0.035}$ | $0.860_{0.017}$ | $0.866_{0.004}$ | 4 |
| DMPNN | $0.864_{0.010}$ | $0.976_{0.004}$ | $0.889_{0.005}$ | $0.617_{0.050}$ | $0.821_{0.019}$ | $0.819_{0.004}$ | 2 |
| AttrMasking | $0.892_{0.012}$ | $0.978_{0.006}$ | $0.929_{0.006}$ | $0.577_{0.087}$ | $0.846_{0.021}$ | $0.817_{0.005}$ | 4 |
| ContextPred | $0.897_{0.004}$ | $0.975_{0.004}$ | $0.923_{0.005}$ | $0.671_{0.026}$ | $0.818_{0.017}$ | $0.827_{0.003}$ | 1 |
| TranFoxMol | $0.868_{0.019}$ | $0.951_{0.036}$ | $0.875_{0.011}$ | $0.619_{0.019}$ | $0.837_{0.017}$ | $0.860_{0.006}$ | 0 |
| DeepMol | $0.774_{0.023}$ | $0.880_{0.012}$ | $0.821_{0.007}$ | $0.509_{0.026}$ | $0.735_{0.015}$ | $0.770_{0.008}$ | 0 |
| **MuMo** | $0.899_{0.014}$ | $0.979_{0.013}$ | $0.942_{0.019}$ | $0.714_{0.021}$ | $0.840_{0.015}$ | $0.880_{0.017}$ | 7 |

| Models | BACE-R | BACE-S | BBBP-R | BBBP-S | CLINTOX | SIDER | TOX21 |
|---|---|---|---|---|---|---|---|
| MoleculeNet - Classification - AUROC ↑ | | | | | | | |
| FPGNN | $0.831_{0.011}$ | $0.831_{0.011}$ | $0.904_{0.020}$ | $0.892_{0.019}$ | $0.732_{0.068}$ | $0.661_{0.014}$ | $0.833_{0.004}$ |
| TransFoxMol | $0.780_{0.032}$ | $0.780_{0.032}$ | $0.907_{0.024}$ | $0.881_{0.015}$ | $0.830_{0.047}$ | $0.636_{0.022}$ | $0.816_{0.011}$ |
| ChemBERTa-2 | $0.848_{0.037}$ | $0.848_{0.037}$ | $0.932_{0.037}$ | $0.892_{0.019}$ | $0.933_{0.054}$ | $0.708_{0.090}$ | $0.809_{0.029}$ |
| MoLFormer | $0.873_{0.009}$ | $0.833_{0.009}$ | $0.889_{0.028}$ | $0.868_{0.013}$ | $0.888_{0.044}$ | $0.651_{0.016}$ | $0.804_{0.013}$ |
| MolBERT | $0.882_{0.015}$ | $0.832_{0.015}$ | $0.955_{0.008}$ | $0.949_{0.013}$ | $0.875_{0.041}$ | - | - |
| GROVER | $0.779_{0.059}$ | $0.779_{0.059}$ | $0.849_{0.008}$ | $0.823_{0.020}$ | $0.685_{0.066}$ | $0.635_{0.034}$ | $0.808_{0.014}$ |
| Uni-Mol | $0.840_{0.031}$ | $0.84_{0.031}$ | $0.889_{0.025}$ | $0.886_{0.016}$ | $0.818_{0.065}$ | $0.666_{0.021}$ | $0.812_{0.007}$ |
| **MuMo** | $0.878_{0.046}$ | $0.849_{0.014}$ | $0.962_{0.007}$ | $0.957_{0.011}$ | $0.985_{0.011}$ | $0.677_{0.009}$ | $0.834_{0.009}$ |

| Models | LD50 | Caco-2 | PPBR | LIPO | Models | ESOL | Freesolv |
|---|---|---|---|---|---|---|---|
| TDC Datasets - Regression - MAE ↓ | | | | | MoleculeNet - Regression - RMSE ↓ | | |
| AttentiveFP | $0.678_{0.012}$ | $0.401_{0.032}$ | $9.373_{0.335}$ | $0.572_{0.007}$ | ChemBERTa-2 | $0.633_{0.132}$ | $1.219_{0.206}$ |
| FPGNN | $0.638_{0.024}$ | $0.326_{0.040}$ | $8.465_{1.709}$ | $0.544_{0.011}$ | FPGNN | $0.658_{0.006}$ | $1.106_{0.195}$ |
| DMPNN | $0.607_{0.022}$ | $0.388_{0.077}$ | $8.158_{0.314}$ | $0.448_{0.014}$ | GROVER | $0.617_{0.077}$ | $1.901_{0.459}$ |
| AttrMasking | $0.685_{0.025}$ | $0.546_{0.052}$ | $10.075_{0.202}$ | $0.547_{0.024}$ | MoLFormer | $0.653_{0.029}$ | $1.190_{0.046}$ |
| ContextPred | $0.669_{0.030}$ | $0.502_{0.036}$ | $9.445_{0.224}$ | $0.535_{0.012}$ | MolBERT | $0.617_{0.091}$ | $1.311_{0.257}$ |
| TranFoxMol | $0.645_{0.036}$ | $0.487_{0.068}$ | $9.055_{0.523}$ | $0.525_{0.024}$ | TranFoxMol | $0.930_{0.261}$ | $1.225_{0.155}$ |
| DeepMol | $0.589_{0.006}$ | $0.327_{0.012}$ | $9.533_{0.162}$ | $0.660_{0.004}$ | Uni-Mol | $0.769_{0.153}$ | $1.598_{0.153}$ |
| **MuMo** | $0.426_{0.031}$ | $0.315_{0.055}$ | $7.324_{0.323}$ | $0.448_{0.007}$ | **MuMo** | $0.536_{0.061}$ | $1.082_{0.088}$ |

Table 2: Evaluation on QM9 benchmarks from Uni-Mol-v2 [Ji et al., 2024]. Results are MAE (↓). Standard errors are in gray subscript. The top and second top results are highlighted in pink.

| Model | HOMO/LUMO/GAP ↓ | $\alpha$ ↓ | $C_v$ ↓ | $\mu$ ↓ | $R^2$ ↓ | ZPVE ↓ |
|---|---|---|---|---|---|---|
| GROVER-base | $0.0079_{3\text{E-}04}$ | $2.365_{0.302}$ | $1.103_{0.339}$ | $0.618_{0.002}$ | $113.01_{4.206}$ | $0.0035_{3\text{E-}04}$ |
| GROVER-large | $0.0083_{6\text{E-}04}$ | $2.240_{0.385}$ | $0.853_{0.186}$ | $0.623_{0.006}$ | $85.85_{6.816}$ | $0.0038_{5\text{E-}04}$ |
| GEM | $0.0067_{4\text{E-}05}$ | $0.589_{0.0042}$ | $0.237_{0.0137}$ | $0.444_{0.0015}$ | $25.67_{0.743}$ | $0.0011_{2\text{E-}05}$ |
| Uni-Mol | $0.0043_{2\text{E-}05}$ | $0.363_{0.009}$ | $0.183_{0.002}$ | $0.155_{0.0015}$ | $4.805_{0.055}$ | $0.0011_{3\text{E-}05}$ |
| Uni-Mol2 310M | $0.0036_{1\text{E-}05}$ | $0.315_{0.003}$ | $0.143_{0.002}$ | $0.092_{0.0013}$ | $4.672_{0.245}$ | $0.0005_{1\text{E-}05}$ |
| Uni-Mol2 570M | $0.0036_{2\text{E-}05}$ | $0.315_{0.004}$ | $0.147_{0.007}$ | $0.089_{0.0015}$ | $4.523_{0.080}$ | $0.0005_{3\text{E-}05}$ |
| Uni-Mol2 1.1B | $0.0035_{1\text{E-}05}$ | $0.305_{0.003}$ | $0.144_{0.002}$ | $0.089_{0.0004}$ | $4.265_{0.067}$ | $0.0005_{8\text{E-}05}$ |
| **MuMo 505M** | $0.0030_{1\text{E-}05}$ | $0.283_{0.003}$ | $0.126_{0.003}$ | $0.400_{0.0018}$ | $18.08_{0.533}$ | $0.0005_{1\text{E-}05}$ |

absorption, distribution, metabolism, excretion, and toxicity (ADMET) challenges and leaderboard baselines, and 12 from MoleculeNet [Wu et al., 2018], along with 3 chemical tasks from Reaxtica [Lin et al., 2022] which enables evaluation against strong unimodal and pretrained models. These tasks cover a range of molecular properties, including bioactivity and ADMET-related endpoints, for both classification and regression.

**Baselines**. We compare MuMo against various competitive baselines spanning diverse modalities and pretrained algorithms. These include 3D-aware models like FPGNN [Cai et al., 2022] and Uni-Mol [Zhou et al., 2023] that incorporate spatial geometry, 2D graph-based models such as HiGNN [Li et al., 2021] and GCN [Kipf and Welling, 2016] that rely solely on molecular topology, and pretrained models include GROVER [Rong et al., 2020], MoLFormer [Ross et al., 2022], and

Table 3: Evaluation on catalytic activity and reaction yield benchmarks from Reaxtica Lin et al. [2022]. MuMo achieves the best performance on three tasks. Standard deviations are shown in gray subscript where available. The best result is highlighted.

| BHC ($R^2 \uparrow$, Reaction Yield) | | CPA (MAE $\downarrow$, Catalytic Activity) | | HTE ($R^2 \uparrow$, Reaction Yield) | |
|---|---|---|---|---|---|
| Models | Value | Models | Value | Models | Value |
| Reaxtica | 0.94 | Reaxtica | 0.144 | Reaxtica | 0.87 |
| MFF | 0.92 | MFF | 0.144 | rxnfp | 0.81 |
| rxnfp | 0.95 | Denmark et al. | 0.152 | DRFP | 0.85 |
| **MuMo** | **0.952**$_{0.002}$ | **MuMo** | **0.144**$_{0.000}$ | **MuMo** | **0.873**$_{0.002}$ |

Table 4: Ablation study on the effectiveness of Structural Fusion Pipeline. "2D" column refers to the use of either 2D topological information or SMILES sequence alone. Mean and standard deviation are reported for two classification tasks (AUROC) and two regression tasks (RMSE).

| | | | Classification | | Regression | | |
|---|---|---|---|---|---|---|---|
| 2D | SUG | GSP | BACE $\uparrow$ | BBBP $\uparrow$ | ESOL $\downarrow$ | LIPO $\downarrow$ | IMPACT |
| √ | √ | √ | 0.849$_{0.014}$ | 0.957$_{0.011}$ | 0.536$_{0.061}$ | 0.577$_{0.027}$ | 0.00% |
| √ | × | √ | 0.821$_{0.005}$ | 0.956$_{0.003}$ | 0.664$_{0.025}$ | 0.615$_{0.018}$ | -7.46% |
| √ | √ | × | 0.849$_{0.003}$ | 0.960$_{0.002}$ | 0.585$_{0.030}$ | 0.614$_{0.017}$ | -3.00% |
| √ | × | × | 0.841$_{0.004}$ | 0.949$_{0.003}$ | 0.654$_{0.027}$ | 0.630$_{0.016}$ | -7.29% |
| × | × | × | 0.766$_{0.006}$ | 0.956$_{0.004}$ | 0.719$_{0.022}$ | 0.655$_{0.015}$ | -13.11% |

ChemBERTa-2 [Ahmad et al., 2022] that requires large-scale self-supervised learning before fine-tuning on specific tasks. The comprehensive comparisons forcefully demonstrate the effectiveness of our MuMo in molecule representation across architectures, input modalities, and learning paradigms.

**Settings.** We follow official protocols or recommendations for fair comparison in each benchmark. AUROC is used for classification; MAE (TDC) and RMSE (MoleculeNet) for regression. MoleculeNet tasks use scaffold split for single-objective classification; otherwise, random. Each task is run 5 times: we use the official leaderboard splits for TDC and generate 5 splits for MoleculeNet (Train:Valid:Test=8:1:1). Hyperparameters follow each baseline's official setup or defaults if unspecified. Additional details about datasets and settings are provided in Appendix C.5.

## 4.2 Main Performance and Analysis

As Table 1 shows, MuMo outperforms the best baseline by an average of 2.7% across 21 benchmark tasks from TDC and MoleculeNet, ranking first on 17 of them, and even improves up to 27% on LD50 compared to DeepMol [Correia et al., 2024]. Compared to other fusion models like Uni-Mol [Zhou et al., 2023] and TranFoxMol [Gao et al., 2023], MuMo exhibits more consistent performance across different tasks, validating the benefit of progressive and asymmetric integration of structural information. On conformer-sensitive regression tasks such as PPBR and LD50, MuMo maintains the lowest error, highlighting its robustness to geometric noise. As shown in Table 2 and 5, MuMo consistently outperforms other baselines on QM7/8/9 datasets (7 out of 10 tasks), which are known to be sensitive to conformer and molecular geometry, further highlighting its robustness to conformer-sensitive and superior 3D molecular modeling capability.

Table 5: Evaluation on QM7/8 and QM9 (HOMO/LUMO/GAP) benchmarks from MoleculeNet. Results are MAE ($\downarrow$). Standard deviations are in gray subscript. The top result is highlighted.

| Model | QM7 $\downarrow$ | QM8 $\downarrow$ | QM9 $\downarrow$ |
|---|---|---|---|
| GROVER | 92.0$_{0.9}$ | 0.0224$_{0.0003}$ | 0.0099$_{0.00025}$ |
| DMPNN | 103.5$_{8.6}$ | 0.0190$_{0.0001}$ | 0.0081$_{0.00001}$ |
| AttentiveFP | 72.0$_{2.7}$ | 0.0179$_{0.0010}$ | 0.0081$_{0.00001}$ |
| UniMol | 41.8$_{0.2}$ | 0.0156$_{0.0001}$ | 0.0047$_{0.00004}$ |
| **MuMo** | 42.8$_{0.6}$ | **0.0111**$_{0.0001}$ | **0.0030**$_{0.00001}$ |

## 4.3 Broader Chemical Benchmarks

Our original design focuses on single-molecule property prediction, which is why the baselines and benchmarks were selected accordingly. However, we also investigate the model's generalization to broader chemical domains beyond individual molecules. To this end, we extend MuMo

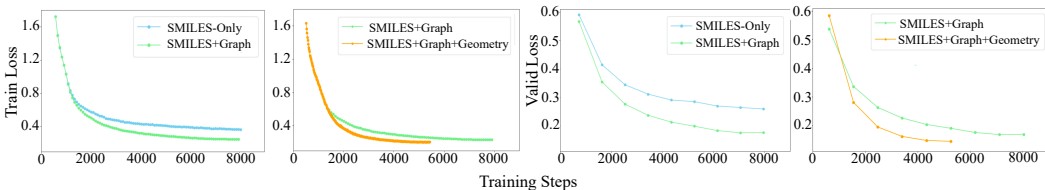

Figure 4: Pretraining loss curves under different modality configurations. Each part shows training (left two figures) and validation (right two figures) loss for a pairwise modality comparison.

to accept reaction-level inputs and evaluate it on four datasets from Reaxtica Lin et al. [2022], following their official data splits and reported baselines. These datasets cover two important domains: catalytic activity and reaction yield. As shown in Table 3, MuMo achieves the best results on three out of four tasks, surpassing prior methods such as Reaxtica, MFF, rxnfp, and DRFP.

### 4.4 Ablation Studies

**Contribution of components in Structural Fusion Pipeline.** We conduct ablation studies on MoleculeNet tasks to assess the effect of core components in SFP: "SUG" (structural unified graph representation for multimodal signals), and "GSP" (geometry-aware substructure partitioning). As shown in Table 4, using only sequence information results in the largest degradation (-13.11%), which shows the importance of leveraging both topological and geometric signals. Removing 3D geometry (no SUG) leads to a significant drop (-7.46%), and removing the GSP, which aggregates the local and global structural features, results in -3.00%. These results demonstrate not only the effectiveness of aligning 3D information into a unified representation and encapsulating multi-scale structural

Table 6: Impact of injection and its timing. Results on BBBP (AUROC) and ESOL (RMSE). "@a–b" denotes injection between layers a and b.

| Variant | BBBP ↑ | ESOL ↓ | Avg.Drop |
|---|---|---|---|
| MuMo @ 9–16 | **0.957**$_{0.011}$ | **0.536**$_{0.061}$ | 0.00% |
| Full-Inj @ 1–16 | 0.954$_{0.002}$ | 0.587$_{0.025}$ | -1.85% |
| Early-Inj @ 1–8 | 0.946$_{0.003}$ | 0.597$_{0.022}$ | -2.96% |
| Late-Inj @ 13–16 | 0.961$_{0.002}$ | 0.617$_{0.028}$ | -4.40% |
| None-Inj (Seq) | 0.928$_{0.004}$ | 0.939$_{0.030}$ | -18.91% |

Table 7: Impact of injection approaches (progressive injection vs. fixed injection).

| Variant | BBBP ↑ | ESOL ↓ | Avg.Drop |
|---|---|---|---|
| Progressive-Inj | 0.957$_{0.011}$ | 0.536$_{0.061}$ | 0.00% |
| Fixed-Inj | 0.946$_{0.008}$ | 0.597$_{0.051}$ | -6.28% |

signals, but also highlight that each component provides complementary benefits that are essential for the robustness of SFP. In particular, the synergy between SUG and GSP allows the model to capture richer chemical priors, yielding stronger generalization across diverse molecular benchmarks.

**Effects of components in Progressive Injection.** To evaluate the effectiveness of our injection strategy and the independent propagation of the structural prior, we conduct two ablation studies in Table 6 and 7. From Table 6, structural prior injection improves the performance by a wide margin regardless of timing (14.51%, 15.95% and 17.06% for late-/early-/full injection). However, early-/full-injection introduces modality collapse due to underdeveloped sequence semantics, while late-injection provides inadequate structural information. Our MuMo injects from layer 9, yielding the best results by balancing semantic establishment and structural guidance. As for propagation (Table 7), using the fixed structural prior throughout the inference will hinder the progressive refinement of structural representations. Fortunately, by propagating structural prior, MuMo improves 6.28%, demonstrating the benefits of the independent evolution of structural information across layers.

**Impact of input modalities on pretraining dynamics.** Figure 4 illustrates the impact of structural signals on pretraining loss curves. Compared to SMILES-only, adding 2D graph consistently accelerates convergence and lowers both training and validation loss, indicating that topological priors enhance early learning. Further incorporating geometry leads to the lowest loss across all steps, suggesting that spatial information provides strong inductive signals for alignment. These trends highlight the complementary role of each modality in guiding effective pertaining.

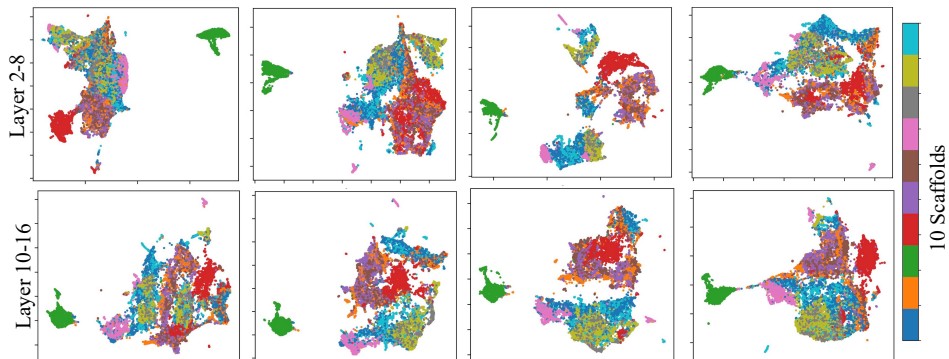

Figure 5: Layer-wise representation of the pretrained model. UMAP of embeddings across 10 selected scaffolds (5,000 molecules), showing scaffold-level separation at different layers.

## 4.5 Visualization Insights

**Molecular Similarity Analysis.** To assess the effectiveness of generated embeddings in distinguishing distinct molecules, we analyze their Pearson correlation with established molecular dissimilarity or similarity representations. Specifically, we randomly sample 20,000 molecules from the ZINC dataset [Gómez-Bombarelli et al., 2018] to construct molecule pairs. We then generate their embeddings by MolFormer and MuMo, and compute the embedding distance for each pair. We measure the Pearson correlation between the embedding distances and the distances from widely used structural metrics: Tanimoto distance and MCS substructure overlap. As shown in Figure 6, MuMo exhibits stronger correlations than MoLFormer, demonstrating its ability to produce robust molecular embeddings and reflect underlying structural relationships. See Appendix C.7 for details.

**Representation Distribution with Structural Prior.** To investigate how structural priors affect the evolution of molecular representations across network layers, we perform a layer-wise analysis of embeddings using Manifold Approximation and Projection (UMAP). As shown in Figure 5, in the early layers before injection (Layers 1–8), scaffold separation gradually emerges, indicating that the model is progressively extracting semantic features from the sequence stream. In the later layers (Layers 9–12), where structural priors are injected, the distributions become more compact and form clear scaffold-specific clusters. This indicates that the structural prior reinforces global perception without disrupting the semantic patterns learned before.

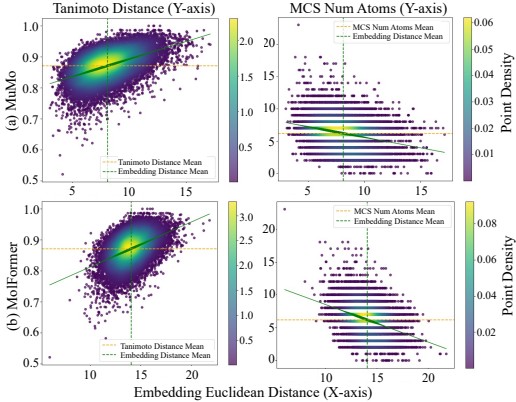

Figure 6: Visualization of similarity analysis of two models: MuMo and MoLFormer.

This validates our motivation for progressive injection: establishing sufficient modality-specific encoding before introducing cross-modal guidance for discriminative representations.

## 5 Conclusion

We introduce MuMo, a structured multimodal framework designed to address the unreliability of conformer-dependent fusion and the modality collapse caused by naive modality integration. MuMo includes the Structured Fusion Pipeline, which combines 2D topological and 3D geometric information into a stable structural prior, reducing sensitivity to noisy or inconsistent conformers. Progressive Injection (PI) mechanism then asymmetrically injects the structural prior into the sequence stream and evolves independently, enabling cross-modal enrichment while preserving semantic autonomy. Extensive experiments on a wide range of tasks show that MuMo consistently performs over various baselines and is robust on conformer-sensitive tasks. It highlights MuMo as a promising multimodal approach for building robust, geometry-aware molecular models. While MuMo is currently tailored to be a task-specific model, future work will focus on extending it into a general-purpose multimodal backbone for molecular representation learning.

## Acknowledgments and Disclosure of Funding

This work was supported in part by the Canada Research Chairs Tier II Program (CRC-2021-00482), the Canadian Institutes of Health Research (PLL 185683, PJT 190272, PJT204042), the Natural Sciences, Engineering Research Council of Canada (RGPIN-2021-04072) and The Canada Foundation for Innovation (CFI) John R. Evans Leaders Fund (JELF) program (#43481).

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

## Appendix Table of Contents

## A    Geometric Completeness Discussion

In this work, we introduced the MuMo model, which seamlessly integrates the topological and geometric information of molecules. The accuracy and completeness of geometric representations are crucial for capturing the underlying spatial properties. In the context of molecular representation, the ability to distinguish between stereoisomers is essential, as it highlights the importance of capturing fine-grained geometric detail, which is a key advantage of our geometric modeling.

### A.1    Proof of Rotational Invariance in the Unified Graph Structure

In this section, we present a rigorous proof that the Unified Graph structure framework, as defined by

$$\mathcal{T}_{Entity} = \big(\mathcal{V}, \mathcal{E}, \mathcal{G}\big) \quad \text{and} \quad \mathcal{T}_{Constraint} = \big(C_{\mathcal{E}}, C_{\mathcal{G}}\big), \tag{9}$$

exhibits invariance under arbitrary rotations in three-dimensional Euclidean space. We first restate the core components of the Unified Graph in a concise manner, then formally define rotational invariance, and finally offer a detailed proof, complete with references to fundamental geometric identities and transformations.

**Definitions**. For clarity and convenience, previously defined terms in Section 3.2.1 are restated. In the Unified Graph, $\mathcal{V}$ is the node set, where each node $v_i \in \mathcal{V}$ is endowed with a feature vector

$v_i \in \mathbb{R}^{d_V}$ and a spatial coordinate $p_i \in \mathbb{R}^3$. Edge set is $\mathcal{E}$, where each edge $e_{ij} \in \mathcal{E}$ connects the node pair $(v_i, v_j)$ and is described by an edge feature vector $e_{ij} \in \mathbb{R}^{d_E}$. In addition to these standard graph entities, Unified Graph introduces a geometric set $\mathcal{G}$ that provides essential spatial properties for each edge and its adjacent edges. Specifically, for an edge $e_{ij}$ sharing a common vertex $v_j$ with another edge $e_{jk}$, the corresponding geometric representation $g_{ij,k} \in \mathcal{G}$ is the triplet $g_{ij,k} = \left( l_{ij}, l_{jk}, \theta_{ij,k} \right)$, where

$$l_{ij} = \left\| p_i - p_j \right\|_2, \qquad l_{jk} = \left\| p_j - p_k \right\|_2, \qquad \cos(\theta_{ij,k}) = \frac{\langle\, p_i - p_j,\ p_k - p_j \,\rangle}{\left\| p_i - p_j \right\|_2 \left\| p_k - p_j \right\|_2}. \tag{10}$$

Here, $\| \cdot \|_2$ and $\langle \cdot, \cdot \rangle$ represent the Euclidean norm and the dot product in $\mathbb{R}^3$, respectively. The constraint group $\mathcal{T}_{Constraint} = (C_\mathcal{E}, C_\mathcal{G})$ codifies topological and geometric consistency via the edge index set $C_\mathcal{E}$ and the shared-vertex edge pair index set $C_\mathcal{G}$. Our focus is the invariance of $(l_{ij}, l_{jk}, \theta_{ij,k})$ under arbitrary rotations.

A representation in $\mathbb{R}^3$ is said to be rotationally invariant if, for any rotation matrix $R \in \mathbb{R}^{3\times3}$ that is orthonormal (i.e., $R^\top R = I$) and any translation vector $b \in \mathbb{R}^3$, the core geometric descriptors remain unchanged. Concretely, if one transforms the node coordinates via $p_i' = R p_i + b$, then the resulting triplets $g_{ij,k}' = \left( l_{ij}', l_{jk}', \theta_{ij,k}' \right)$ must satisfy

$$l_{ij}' = l_{ij}, \qquad l_{jk}' = l_{jk}, \qquad \theta_{ij,k}' = \theta_{ij,k}. \tag{11}$$

We next show that Unified Graph's definitions inherently guarantee this property.

**Proof of Rotational Invariance**. For any pair of nodes $(v_i, v_j)$, consider the original coordinate difference $p_i - p_j$ and its length $\left\| p_i - p_j \right\|_2$. Under the transformation $p_i' = R p_i + b$, we have

$$p_i' - p_j' = \left( R p_i + b \right) - \left( R p_j + b \right) = R( p_i - p_j ). \tag{12}$$

Hence the new length is

$$l_{ij}' = \left\| p_i' - p_j' \right\|_2 = \left\| R\left( p_i - p_j \right) \right\|_2 = \sqrt{\left( p_i - p_j \right)^\top R^\top R \left( p_i - p_j \right)}$$

$$= \sqrt{\left( p_i - p_j \right)^\top I \left( p_i - p_j \right)} = \left\| p_i - p_j \right\|_2 = l_{ij}.$$

Since the same argument applies for $(v_j, v_k)$, we obtain $l_{jk}' = l_{jk}$.

Then, the invariance of angles should be proved. We should show that $\theta_{ij,k}' = \theta_{ij,k}$ under the same transformation. Observe that

$$\cos(\theta_{ij,k}') = \frac{\langle p_i' - p_j',\ p_k' - p_j' \rangle}{\left\| p_i' - p_j' \right\|_2 \left\| p_k' - p_j' \right\|_2} = \frac{\langle\, R\left( p_i - p_j \right),\ R\left( p_k - p_j \right) \rangle}{\left\| R\left( p_i - p_j \right) \right\|_2 \left\| R\left( p_k - p_j \right) \right\|_2}. \tag{13}$$

The numerator of this fraction can be expanded using the invariance of the dot product under orthonormal transformations:

$$\langle\, R\left( p_i - p_j \right),\ R\left( p_k - p_j \right) \rangle = \left( p_i - p_j \right)^\top R^\top R \left( p_k - p_j \right) = \left( p_i - p_j \right)^\top \left( p_k - p_j \right) = \langle p_i - p_j,\ p_k - p_j \rangle. \tag{14}$$

Meanwhile, the denominator reduces precisely to $\left\| p_i - p_j \right\|_2 \left\| p_k - p_j \right\|_2$ by the argument in the proof of invariance of edge length. Consequently, we have

$$\cos(\theta_{ij,k}') = \frac{\langle\, R\left( p_i - p_j \right),\ R\left( p_k - p_j \right) \rangle}{\left\| R\left( p_i - p_j \right) \right\|_2 \left\| R\left( p_k - p_j \right) \right\|_2} = \frac{\langle\, \left( p_i - p_j \right),\ \left( p_k - p_j \right) \rangle}{\left\| \left( p_i - p_j \right) \right\|_2 \left\| \left( p_k - p_j \right) \right\|_2} = \cos(\theta_{ij,k}), \tag{15}$$

$$\theta_{ij,k}, \theta_{ij,k}' \in [0, \pi] \Rightarrow \quad \theta_{ij,k}' = \theta_{ij,k}. \tag{16}$$

By combining the above two results, we conclude that for every edge $e_{ij}$ and its adjacent edge $e_{jk}$, the geometric triplet $g_{ij,k}' = \left( l_{ij}', l_{jk}', \theta_{ij,k}' \right)$ remains identical to $g_{ij,k}$ under any spatial rotation (and translation). Hence, the Unified Graph structure fully preserves lengths and angles, guaranteeing invariance of its geometric descriptors with respect to orthonormal transformations in $\mathbb{R}^3$. Formally, for all rotation matrices $R$ with $R^\top R = I$ and translation vectors $b$, the Unified Graph definitions ensure $g_{ij,k}' = g_{ij,k}, \forall\, e_{ij}, e_{jk} \in \mathcal{E}$, which completes the proof of rotational invariance in the Unified data structure framework.

## A.2 Capability of the Unified Graph in Distinguishing Molecular Isomerism

The Unified Graph $\mathcal{T}$ proposed in this work combines topological and geometric features to represent molecules. It includes the topology of nodes and edges as well as geometric descriptors $\mathcal{G}$, which encode edge lengths and angles at shared vertices. This subsection evaluates the capability of the proposed representation in distinguishing different types of molecular isomerism [Axelrod and Gomez-Bombarelli, 2023].

- **Constitutional Isomers (Structural Isomers).** These isomers differ in the connectivity of atoms, i.e., their topological structures are distinct. Since the Unified Graph explicitly encodes edge connectivity relationships in $\mathcal{C_E}$, structural differences in connectivity are directly reflected in the graph, allowing effective differentiation between constitutional isomers [Datta and Limpanuparb, 2021].

- **Cis/Trans Isomers (Geometric Isomers, E/Z Isomers).** Geometric isomers share the same connectivity but differ in the spatial arrangement of substituents due to constraints such as double bonds or ring structures. These differences manifest as variations in certain interatomic distances or bond angles. The geometric descriptors $l_{ij}$ and $\theta_{ij,k}$ in the Unified Graph can capture these variations, enabling differentiation between cis/trans or E/Z isomers [Smith, 2009].

- **Diastereomers.** Diastereomers, especially those with multiple chiral centers, are not mirror images and often exhibit measurable differences in local geometric features such as bond lengths, bond angles, or interatomic distances. These differences are encoded in the geometric descriptors $\mathcal{G}$, allowing the Unified Graph to distinguish most diastereomers effectively.

- **Enantiomers (Optical Isomers).** Enantiomers are non-superimposable mirror images that are identical in connectivity, bond lengths, and bond angles but differ in their handedness. Since the Unified Graph only uses unsigned lengths and angles without encoding chirality or orientation explicitly, it cannot distinguish between enantiomers, as their representation in $\mathcal{T}$ would be identical [Mislow and Siegel, 1984].

- **Conformers (Conformational Isomers).** Conformational isomers arise from rotations around single bonds, typically resulting in different spatial arrangements of atoms. If these conformational changes do not significantly alter equilibrium bond lengths or angles, and if only one specific conformer is represented in the graph, such differences may not be captured. Hence, rapid interconversion between conformers is usually ignored in the Unified Graph representation [Lovell et al., 2000].

In summary, the Unified Graph $\mathcal{T}$ effectively distinguishes most isomer types, including constitutional isomers, geometric isomers, and many diastereomers. However, it has limitations in identifying enantiomers due to the absence of chirality-specific descriptors. Future extensions could incorporate chirality-sensitive features, such as signed dihedral angles or higher-dimensional orientation information, to enhance its capability to distinguish optical isomers.

## A.3 Are Explicit Torsion Angles Necessary?

**Step 1. Are torsion angles missing?** A torsion (dihedral) angle $\phi_{ijkl}$ is fully determined by the six inter-atomic distances of a four-atom chain $(i, j, k, l)$:

$$\phi_{ijkl} = \text{atan2}\big((\mathbf{r}_{ji} \times \mathbf{r}_{jk}) \cdot \mathbf{r}_{kl}, \ (\mathbf{r}_{ji} \times \mathbf{r}_{jk}) \cdot (\mathbf{r}_{jk} \times \mathbf{r}_{kl})\big),$$

where $\mathbf{r}_{ab} = \mathbf{x}_b - \mathbf{x}_a, \mathbf{x}_a = (x_a, y_a, z_a)$. Since our molecular graph already provides **all pairwise distances within a 6 Å cutoff**, and message passing runs for at least three layers, an $i \rightarrow l$ path that closes the $i-j-k-l$ quadrangle always exists. Thus, the network can **implicitly reconstruct torsion angles** without requiring explicit torsion features.

**Step 2. Why not add explicit torsion anyway?** To test the benefit of explicit torsion encoding, we added $[\sin \phi_{ijkl}, \cos \phi_{ijkl}]$ on every rotatable bond while keeping all other hyperparameters unchanged. As shown in Table 8, the performance difference on QM9 property benchmarks is minimal: only $C_v$ shows a moderate gain (+10.3%), while HOMO/LUMO/GAP performance even drops slightly (-6.67%). However, Table 9 shows that adding torsions sharply increases GPU memory and runtime (up to 2.6$\times$ slower), indicating that the marginal accuracy benefits do not justify the computational overhead.

Table 8: Explicit torsion ablation on QM9. Results are MAE (↓) with standard errors in gray subscript.

| Task (MAE) | w/o Torsion | w/ Torsion | $\Delta$ (%) |
|---|---|---|---|
| $\alpha$ (↓) | $0.283_{0.003}$ | $0.281_{0.003}$ | +0.71 |
| $C_v$ (↓) | $0.126_{0.003}$ | $\mathbf{0.113_{0.003}}$ | +10.3 |
| ZPVE (↓) | $0.0005_{6E\text{-}05}$ | $0.0005_{6E\text{-}05}$ | 0.0 |
| HOMO/LUMO/GAP (↓) | $\mathbf{0.00301_{1E\text{-}05}}$ | $0.00321_{1E\text{-}05}$ | -6.67 |
| **Average** | – | – | +1.00 |

Table 9: Efficiency comparison. GPU memory usage and step time are reported in Pretrain/SFT(QM8) format with global batch size 128/64.

| Model variant | GPU Mem (GB) ↓ | Step time (ms) ↓ |
|---|---|---|
| Ours (implicit) | **44.4/14.4** | **987/731** |
| +Explicit torsion | 64.6/29.4 | 2623/1316 |

**Step 3. Robustness to conformer sensitivity.** Beyond efficiency, we further validated robustness on QM7/8/9 conformer sensitivity benchmarks (see Section 4). MuMo consistently outperforms UniMol under perturbations, confirming that **implicit torsion modeling is sufficient** and our injection-enhanced design maintains strong robustness to 3D geometric noise.

# B  Relationship with Previous Methods

Understanding the relationship between our proposed method and prior approaches is crucial for situating our contributions within the broader research landscape. This section aims to highlight the key distinctions and improvements introduced by our model while also acknowledging the foundational principles laid by existing methodologies.

## B.1  Mamba State Space Model

**The Mamba Model**. State Space Models (SSMs) are a class of mathematical frameworks widely used for modeling temporal or sequential data by describing latent dynamics and observations [Gu and Dao, 2023]. An SSM typically consists of two components: a latent state evolution equation and an observation equation. Formally, let $h_t \in \mathbb{R}^d$ represent the latent state at time $t$, and let $y_t \in \mathbb{R}^o$ be the corresponding observation. The SSM is defined as:

$$h_{t+1} = Ah_t + Bu_t + \eta_t, \quad y_t = Ch_t + \epsilon_t, \tag{17}$$

where $u_t$ is an input sequence, $\eta_t$ and $\epsilon_t$ are process and observation noise, respectively, and $A, B, C$ are model parameters that govern the latent dynamics and the observation process. Mamba builds on the SSM framework and introduces significant advancements to enhance its efficiency and applicability to long-sequence modeling. By leveraging the SSM's inherent ability to capture long-range dependencies, Mamba employs a **selective scanning mechanism** that optimizes the representation of sequences across diverse time scales. Specifically, Mamba avoids the pitfalls of full dense computation by introducing a structured representation of state transitions that achieves logarithmic scaling in time complexity.

There are several core innovations of Mamba, and that is why we use Mamba as a core stacked module for fusion modeling. **a)** Logarithmic Scaling in Time Complexity. Mamba reformulates the SSM's computation by leveraging selective updates to the state vector $h_t$, reducing computational overhead from $O(T^2)$ (where $T$ is the sequence length) to $O(T \log T)$. This efficiency makes it suitable for applications involving very long sequences, such as large molecular data or genomic data.

**b)** Hardware-Aware Optimizations. Mamba introduces approximations to matrix exponentials that are hardware-friendly, enabling efficient computation on modern accelerators like GPUs and TPUs without sacrificing modeling accuracy.

**c)** General Applicability. The model supports diverse data modalities, such as text sequence, and time series, by adapting the SSM framework to handle modality-specific structures, making it a versatile tool for various sequence modeling tasks.

In Mamba, the continuous-time state evolution is modeled as:

$$\frac{d\boldsymbol{h}(t)}{dt} = \boldsymbol{A}\boldsymbol{h}(t) + \boldsymbol{B}\boldsymbol{u}(t), \tag{18}$$

where $\boldsymbol{A}$ is parameterized to ensure stability. This differential equation is solved efficiently using approximations of matrix exponentials:

$$\boldsymbol{h}(t + \Delta t) \approx e^{\boldsymbol{A}\Delta t}\boldsymbol{h}(t) + \int_0^{\Delta t} e^{\boldsymbol{A}s}\boldsymbol{B}\boldsymbol{u}(t + s)\, ds. \tag{19}$$

By discretizing the system with high precision and leveraging sparsity in $\boldsymbol{A}$, Mamba achieves efficient state transitions and improved memory usage. Mamba's selective state-space design enables it to handle sequences spanning thousands to millions of time steps while maintaining accuracy and efficiency. These features make it particularly suitable for tasks such as protein structure prediction, time-series forecasting, and for sure molecular representation learning.

**Discussion of Relationship.** Mamba serves as the sequence encoder in our multimodal molecular framework, offering efficient and scalable modeling of SMILES sequences via state-space dynamics. Its recurrent architecture not only improves computational efficiency but also aligns well with our Injection-Enhanced Attention (IEA, basic module of PI) design. By maintaining an evolving latent state, Mamba naturally accommodates injected structural priors without disrupting local token interactions.

However, Mamba is not central to the methodological contributions of this work. Our core innovations, the Structured Fusion Pipeline and asymmetric cross-modal injection, define the model's effectiveness in robust multimodal fusion. These techniques are model-agnostic and can be applied to other sequence encoders (e.g., Transformers). Mamba's role is to enhance the stability and propagation of injected priors with sequence stream, but it does not influence the fundamental novelty or adaptability of our approach.

## B.2 Breaking of Retrosynthetically Interesting Chemical Substructures

**Retrosynthetic analysis** is a systematic approach in organic synthesis that involves deconstructing a target molecule into simpler precursor structures by breaking bonds in a logical and chemically feasible manner [Law et al., 2009]. This process is guided by the identification of strategic bonds, which, when retrosynthetically cleaved simplify the molecule while preserving its essential functional groups. The ultimate goal is to map out potential synthetic routes, starting from readily available building blocks.

In retrosynthesis, disconnection is the conceptual reversal of a bond-forming reaction, often symbolized by a double-headed arrow ($\Rightarrow$). For instance, consider the retrosynthesis of benzyl alcohol ($C_6H_5-CH_2OH$):

$$C_6H_5-CH_2OH \Rightarrow C_6H_5-CH_2X + X-OH \tag{20}$$

In this example, a disconnection of the hydroxymethyl group ($-CH_2OH$) from the benzyl group ($C_6H_5-$) suggests two plausible precursors: benzyl halide ($C_6H_5-CH_2X$) and a nucleophile such as water ($H_2O$) or hydroxide ion ($OH^-$).

Another classic example is the retrosynthetic analysis of aspirin (acetylsalicylic acid, $C_9H_8O_4$):

$$C_9H_8O_4 \Rightarrow C_7H_6O_3 + CH_3COCl \tag{21}$$

Here, the ester bond ($-COO-$) in aspirin is retrosynthetically cleaved to yield salicylic acid ($C_7H_6O_3$) and acetyl chloride ($CH_3COCl$) as precursors. These intermediates suggest a forward synthesis involving esterification:

$$C_7H_6O_3 + CH_3COCl \xrightarrow{\text{Base}} C_9H_8O_4 + HCl \tag{22}$$

The disconnection approach is not arbitrary but relies on retrosynthetic transformations that correlate to known reaction types in synthetic chemistry, such as nucleophilic substitution, electrophilic

addition, or condensation reactions. By iteratively applying these transformations, a chemist can work backward from a complex molecule to identify feasible synthetic routes.

This method is particularly powerful when applied to complex natural products or pharmaceuticals, where the identification of key disconnections can dramatically simplify synthesis planning. For example, in the retrosynthesis of penicillin derivatives, the $\beta$-lactam ring is often identified as a core structural unit to preserve, while strategic disconnections focus on assembling the side chains and core step by step.

The BRICS (Breaking of Retrosynthetically Interesting Chemical Substructures) fragmentation method deconstructs complex molecules into chemically meaningful substructures by leveraging retrosynthetic principles. Through rule-based disconnection strategies, BRICS identifies synthetically accessible bond cleavages while preserving chemically stable moieties, such as aromatic rings, and targeting bonds like carbon-carbon single bonds or carbon-heteroatom bonds near functional groups. Each fragment is annotated with a placeholder atom (e.g., "*") to mark cleavage sites, enabling recombination in synthetic processes. For example, benzoic acid (SMILES: CC1=CC=CC=C1C(=O)O) is fragmented into [*]C1=CC=CC=C1 and [*]C(=O)O, retaining the functional features of the parent molecule.

The integration of BRICS into cheminformatics tools like RDKit has streamlined its application across large molecular datasets. With automated fragmentation processes, BRICS enables the efficient generation of annotated substructures for drug discovery, combinatorial library design, and virtual screening. In fragment-based drug discovery, BRICS facilitates the identification of minimal structural units critical for biological activity, supporting structure-activity relationship studies and lead optimization.

**Discussion of Relationship.** The BRICS fragmentation method plays a limited yet practical role in the molecular modeling framework presented in this work, serving primarily as a preprocessing module for extracting meaningful substructures from molecules. Its function is to provide a consistent and logical segmentation of molecular structures, supporting our geometry partitioning by instructing bond pruning. We use the instructions that describe which bond should be cut provided by BRICS to lead our geometry substructure partitioning, which is a part of our innovations.

Furthermore, it is crucial to emphasize that BRICS serves as an interchangeable, modular component within our workflow. While we have selected BRICS as a representative example for fragment generation, the framework is designed to accommodate alternative or more advanced fragmentation techniques. This flexibility ensures that researchers can integrate methods better suited to their specific molecular systems or scientific objectives. For instance, as new chemical structures and synthesis pathways are discovered, fragmentation rules may evolve to reflect these advancements, providing an avenue for continual improvement. However, the refinement of BRICS or its alternatives is not the focus of this work. Rather, our interest lies in demonstrating the versatility of our framework, allowing for the seamless substitution of fragmentation methods without affecting the validity or applicability of the overall system.

### B.3   Limitations and Broader Impact

**Broader Impact** MuMo explores a robust and asymmetric approach to multimodal molecular fusion, aiming to improve the reliability of structure-informed representation learning. Beyond its immediate performance gains, the design principles behind MuMo—such as late-stage modality injection and stable structural priors—may inspire future research in multimodal learning, especially in settings where modality-specific semantics must be preserved (e.g., vision-language tasks, protein-compound modeling, or biomedical imaging). We hope our work contributes to a broader understanding of how to design more interpretable and flexible fusion strategies in deep learning.

**Limitations** Like most molecular learning frameworks, MuMo requires fine-tuning for each downstream task, which can be resource-intensive in settings with limited data or computing power. In future work, we aim to develop a more generalizable multitask framework based on MuMo, enabling cross-task transfer and applicability to a broader range of real-world applications in drug discovery, including candidate prioritization, toxicity screening, and multi-objective molecular optimization.

We emphasize that MuMo is a research tool and does not provide direct clinical or regulatory advice. Responsible use of the model requires expert oversight, especially when applied to sensitive applica-

---

**Algorithm 3** Unified Graph Batching (Detailed)

---

**Input:** List of Unified Graphs $\{\mathcal{T}_1, \ldots, \mathcal{T}_N\}$

**Output:** Batched Unified Graph $\mathcal{T}^{(batch)} = (\mathcal{V}^{(batch)}, \mathcal{E}^{(batch)}, \mathcal{G}^{(batch)}, C_{\mathcal{E}}^{(batch)}, C_{\mathcal{G}}^{(batch)}, \boldsymbol{batch})$

1: Initialize $\mathcal{T}^{(batch)} \leftarrow$ new $\mathcal{T}()$, $\delta_v \leftarrow 0$, $\delta_e \leftarrow 0$       ▷ *Initialize offsets of nodes and edges*

2: **for** $k \leftarrow 1$ **to** $N$ **do**

3:      **Step 1: Merge Entity Features**

4:      $\mathcal{V}^{(batch)} \leftarrow \mathcal{V}^{(batch)} \cup \mathcal{T}_k.\mathcal{V}$       ▷ *Merge nodes (atoms)*

5:      $\mathcal{E}^{(batch)} \leftarrow \mathcal{E}^{(batch)} \cup \mathcal{T}_k.\mathcal{E}$       ▷ *Merge edges (bonds)*

6:      $\mathcal{G}^{(batch)} \leftarrow \mathcal{G}^{(batch)} \cup \mathcal{T}_k.\mathcal{G}$       ▷ *Merge geometry features*

7:      **Step 2: Adjust Constraints**

8:      $C_{\mathcal{E}}^{(batch)} \leftarrow C_{\mathcal{E}}^{(batch)} \cup \{(i + \delta_v, j + \delta_v) \mid (i, j) \in \mathcal{T}_k.C_{\mathcal{E}}\}$       ▷ *Update edge index offset*

9:      $C_{\mathcal{G}}^{(batch)} \leftarrow C_{\mathcal{G}}^{(batch)} \cup \{(\text{Idx}(e_{ij}) + \delta_e, \text{Idx}(e_{jk}) + \delta_e) \mid \{\text{Idx}(e_{ij}), \text{Idx}(e_{jk})\} \in \mathcal{T}_k.C_{\mathcal{G}}\}$       ▷ *For geometry*

10:     **Step 3: Update Offsets**

11:     $\delta_v \leftarrow \delta_v + |\mathcal{T}_k.\mathcal{V}|$       ▷ *Update node offsets*

12:     $\delta_e \leftarrow \delta_e + |\mathcal{T}_k.\mathcal{E}|$       ▷ *Update edge offsets*

13: **end for**

14: $\boldsymbol{batch} \leftarrow \left[ k \cdot \mathbf{1}_{|\mathcal{T}_k.\mathcal{V}|} \right]_{k=1}^{N}$       ▷ *Record batch index*

15: **return** $\mathcal{T}^{(batch)}$

---

tions such as toxicity prediction or candidate drug selection. Future work may explore integrating uncertainty quantification or domain adaptation techniques to further align model predictions with safety and ethical considerations.

## C    Implementation & Experiment Details

### C.1    Unified Graph Batching

In the Unified Graph $\mathcal{T}$, the entity group $\mathcal{T}_{Entity} = (\mathcal{V}, \mathcal{E}, \mathcal{G})$ includes the node set $\mathcal{V}$, edge set $\mathcal{E}$, and the geometric descriptors $\mathcal{G}$. Meanwhile, the constraint group $\mathcal{T}_{Constraint} = (C_{\mathcal{E}}, C_{\mathcal{G}})$ specifies topological connectivity through $C_{\mathcal{E}}$ and shared-vertex edge-pair relationships through $C_{\mathcal{G}}$. Algorithm 3 provides a procedure for merging multiple Unified Graphs $\{\mathcal{T}_1, \ldots, \mathcal{T}_N\}$ into a single batched graph $\mathcal{T}^{(batch)}$. By appropriately offsetting the node and edge indices and unifying the constraint sets, it ensures that each graph's internal structures and relationships remain consistent. Adopting such a Unified batching approach is essential when handling large-scale graph data, as it facilitates parallel processing and significantly improves both training and inference efficiency.

### C.2    Injection Enhanced Attention (IEA) within PI Implementation

A key challenge in multimodal learning with Unified Graphs is to effectively combine topological and geometric features with sequential embeddings. In this work, we propose an injection-enhanced attention (IEA) approach to address this challenge. As shown in Algorithm 4, after performing cross-attention between the sequence representation and the Unified Graph node embeddings, we further inject the globally aggregated features from the Unified graph into the global token $[GTK]$ via a residual connection. This injection, modulated by a learnable scalar $\alpha$, enriches the $[GTK]$ token with structural insights while preserving its original contextual content. As a result, the model acquires a more holistic understanding of both semantic and geometric aspects, thereby enabling more robust information fusion for tasks that require a unified representation of topology, geometry, and sequence semantics.

### C.3    Substructure-Level Tokenizer

A critical challenge when encoding molecular structures is capturing chemical nuances within the SMILES representation. To address this, we design a substructure-level tokenizer that segments SMILES strings based on chemically meaningful units (see Table 11). Rather than splitting strictly at character boundaries, we group tokens at natural substructures such as ring closures, chirality annotations, charged atoms, multi-letter elements, and specific isotopes. This ensures that each token remains a valid chemical entity, preserving the minimal functional meaning of each fragment. Consequently, our tokenizer aligns more closely with fundamental chemical principles, reduces the

**Algorithm 4** Injection Enhanced Attention (Detailed)

---

**Input:** Sequence hidden $h_S^{(t)}$, batched Unified graph $\mathcal{T}^{(batch)}$

**Output:** Updated sequence hidden $h_S^{(t+1)}$, updated Unified batch $\mathcal{T}'^{(batch)}$

1: $h_\mathcal{V}^{(t)} \leftarrow \mathcal{T}^{(batch)}.\mathcal{V}$ ▷ *Extract Unified graph node hidden*
2: **Step 1: Compute Queries, Keys, and Values for Cross-Attention**
3: $h_F^{(t)} \leftarrow$ graph2batch_sequence$(h_\mathcal{V}^{(t)}, \mathcal{T}^{(batch)}.\boldsymbol{batch})$ ▷ *Graph → sequence format*
4: $\boldsymbol{Q}_S, \boldsymbol{K}_S, \boldsymbol{V}_S \leftarrow$ Linear$(h_S^{(t)})$ ▷ *Sequence QKV*
5: $\boldsymbol{Q}_F, \boldsymbol{K}_F, \boldsymbol{V}_F \leftarrow$ Linear$(h_F^{(t)})$ ▷ *Integrated feature QKV*
6: **Step 2: Perform Symmetrized Cross-Attention**
7: $h_F^{(t+1)} \leftarrow$ CrossAttention$(\boldsymbol{Q}_F, \boldsymbol{K}_S, \boldsymbol{V}_S)$ ▷ *Learn from sequence hidden*
8: $h_S^{(t+1)} \leftarrow$ CrossAttention$(\boldsymbol{Q}_S, \boldsymbol{K}_F, \boldsymbol{V}_F)$ ▷ *Learn from fusion hidden*
9: **Step 3: Injection-Enhanced Feature Representation**
10: $h_\mathcal{V}^{(t+1)} \leftarrow$ sequence2graph_batch$(h_F^{(t+1)}, \mathcal{T}^{(batch)}.\boldsymbol{batch})$ ▷ *Sequence → graph format*
11: $h_\mathcal{V}^{\text{pooled}} \leftarrow$ GlobalAddPooling$(h_\mathcal{V}^{(t+1)}, \mathcal{T}^{(batch)}.\boldsymbol{batch})$ ▷ *Global graph pooling*
12: $h_S^{(t+1)}[\texttt{GTK}] \leftarrow$ Norm$(h_S^{(t+1)}[\texttt{GTK}] + \alpha\, h_\mathcal{V}^{\text{pooled}})$ ▷ *Inject pooled vector into [GTK]*
13: $\mathcal{T}'^{(batch)}.\mathcal{V} \leftarrow h_\mathcal{V}^{(t+1)}$ ▷ *Update graph hidden*
14: **return** $h_S^{(t+1)}, \mathcal{T}'^{(batch)}$

---

Table 10: Pretraining hyperparameters for MuMo.

| Hyperparameter | Value |
|---|---|
| Hidden size | 768 |
| Number of layers | 16 (Attention-Mamba) |
| Number of attention heads | 12 |
| Activation function | SILU |
| Normalization | LayerNorm |
| Dropout rate | 0.1 (attention) |
| Batch size | 512 |
| Learning rate | $1 \times 10^{-4}$ |
| Learning rate scheduler | Cosine with 2000 warmup steps |
| Epochs | 2 |
| Gradient accumulation | Enabled |
| Precision | Mixed precision (bf16) |
| Training time | ~5 hours on 4×A100-80GB GPUs |

loss of pertinent information, and handles elaborate notations (e.g., `[C@H]`, `[12C]`, and `%10`) in a chemically consistent manner. By retaining critical structural features within tokens, this approach not only enhances the interpretability of token sequences but also leads to improved performance across a wide range of molecular modeling tasks.

## C.4  MuMo Pretraining

**Pretraining Settings and Resources.** The basic model setup was configured with a hidden size of 768, 16 attention-mamba layers, and 12 attention heads, ensuring robust model capacity. The training batch size was set to 512, with a learning rate of 1e-4 and a cosine learning rate scheduler featuring 2000 warmup steps. We used SILU activation inside the Mamba module, layer normalization, and dropout rates of 0.1 for both attention layers. The training spanned 2 epochs with gradient accumulation and utilized mixed precision with bf16, optimizing computational efficiency. A single pretraining process will take around only 5 hours on 4xA100-80G GPUs.

**Pretraining Dataset.** We adopt the ChEMBL-1.6M dataset [Gaulton et al., 2012] for pretraining, which contains a curated set of bioactive molecules with experimentally validated properties. Compared to large-scale yet noisy corpora like ZINC (mostly synthetically accessible fragments) and PubChem (an extremely broad and noisy collection), ChEMBL provides high-quality, biologically relevant molecules that better reflect the structure-function distributions seen in real-world tasks.

Table 11: Token categorization in the substructure-level SMILES tokenizer. Tokens are grouped by structural or semantic function, including atomic symbols, ring closures, bond types, and model-reserved tokens. Examples and definitions are provided for clarity.

| CATEGORY | EXAMPLES | EXPLANATION |
|---|---|---|
| BASIC ATOMIC SYMBOLS | C, N, O, F, S, P, B, I, c, n, o, p, b, ... | SINGLE-LETTER ATOMIC SYMBOLS, INCLUDING LOWERCASE AROMATIC FORMS. FOR EXAMPLE, c TYPICALLY DENOTES AN AROMATIC CARBON. |
| HALOGENS, MULTI LETTER ELEMENTS | Cl, Br, Si, Na, Ca, Mg, Fe, Zn, Al, K, Li, Ag, Sn, ... | TWO-LETTER SYMBOLS FOR HALOGENS (E.G., Cl, Br) AND MULTI-LETTER ELEMENT SYMBOLS (E.G., Na, Fe), OFTEN REPRESENTING METALS OR METALLOIDS. |
| CHIRAL / CHARGED / ISOTOPIC ATOMS | [C@H], [C@@H], [N+], [O-], [13C], [nH], [B-], [Na+], [S@], [Si@@], [NH2+], [14C], ... | BRACKETED NOTATIONS INCORPORATING CHIRALITY (@, @@), CHARGES (+, -), ISOTOPES (E.G., [13C], [14C]), AND SPECIFIC HYDROGEN COUNTS (E.G., [nH]). |
| RING CLOSURES, BRANCHING | 1, 2, 3, 4, 5, 6, 7, 8, 9, %10, %11, (, ), ... | NUMERIC LABELS (1–9, %10, %11, ETC.) REPRESENT RING CLOSURES, WHILE PARENTHESES INDICATE BRANCHING IN MOLECULAR STRUCTURES. |
| BOND TYPES, SPECIAL SYMBOLS | -, =, #, /, \, :, ~, @, ?, >, *, $, % | VARIOUS SMILES BOND NOTATIONS: SINGLE (-), DOUBLE (=), TRIPLE (#), AND STEREOCHEMICAL (/, \). SPECIAL SYMBOLS LIKE :, ~, AND PUNCTUATION ($, >) ARE ALSO INCLUDED. |
| EXTENDED ATOMIC FORMS | [C-], [NH+], [CH2-], [S-], [n+], [I-], [Na], [C@], [C@@], [SiH], [Sn+2], [O+], [B-], ... | VARIATIONS COMBINING CHARGE STATES ([C-], [N+]), SPECIFIC HYDROGEN COUNTS ([CH2-], [nH]), OR HEAVY ATOMS REPRESENTED IN BRACKETED FORM. |
| MORE EXOTIC ISOTOPES/RADIONUCLIDES | [2H], [3H], [11C], [13C], [15N], [18F], [64Cu], [99Tc], [197Au], [238U], ... | TOKENS REPRESENTING SPECIFIC ISOTOPES AND RADIONUCLIDES IN BRACKET NOTATION. THESE OFTEN APPEAR IN SPECIALIZED DATASETS, SUCH AS RADIOTRACERS. |
| SPECIAL MODEL TOKENS | [GTK], [SEP], [MASK], [UNK], [PAD], [BOS], [EOS], ... | RESERVED TOKENS USED IN MACHINE LEARNING MODELS FOR SEQUENCE PROCESSING, INCLUDING CLASSIFICATION MARKERS, MASKS, UNKNOWN PLACEHOLDERS, AND PADDING SYMBOLS. |

This choice allows our model to learn from pharmacologically meaningful signals while avoiding excessive noise or chemical redundancy.

Importantly, we deliberately pretrain on a relatively small molecular corpus (1.6 million molecules) and still observe fast convergence within just 2 epochs. As demonstrated in later ablation studies (Section D.2), further scaling up pretraining data to larger datasets such as full PubChem (>10M molecules) does not yield consistent downstream improvement. This finding leads to a claim: **effective representation learning for molecules does not require massive-scale pretraining**, especially when the pretraining set is chemically diverse and task-relevant. The MuMo model, with its efficient IEA design and structural fusion pipeline, enables strong generalization from limited-scale pertaining.

**Pretraining Effectiveness.** We have also done pretraining ablation studies to see how the pretraining process contributes to the downstream performance. Please see Appendix D.2 for details.

## C.5  Molecular Properties Predction

### C.5.1  Baselines

We compare MuMo against a wide range of strong baselines, categorized into three primary groups: sequence-based models, graph-based networks, and 3D geometry-aware architectures.

Table 12: Overview of datasets from MoleculeNet for molecular properties prediction experiments.

| Dataset | Category | Task Type | Tasks | Molecules | Split | Metric |
|---|---|---|---|---|---|---|
| BACE | Biophysics | Classification | 1 | 1513 | Scaffold | AUROC |
| BBBP | Physiology | Classification | 1 | 2039 | Scaffold | AUROC |
| Tox21 | Physiology | Classification | 12 | 7831 | Random | AUROC |
| SIDER | Physiology | Classification | 27 | 1427 | Random | AUROC |
| ClinTox | Physiology | Classification | 2 | 1478 | Random | AUROC |
| ESOL | Physical Chemistry | Regression | 1 | 1128 | Random | RMSE |
| Lipophilicity | Physical Chemistry | Regression | 1 | 4200 | Random | RMSE |
| FreeSolv | Physical Chemistry | Regression | 1 | 642 | Random | RMSE |

**Sequence-based models** operate on SMILES strings and typically leverage Transformer-style encoders trained with masked language modeling. These include ChemBERTa-2 [Ahmad et al., 2022], a RoBERTa-style model pretrained on large-scale SMILES data, and MolBERT [Fabian et al., 2020], which incorporates chemically-informed masking strategies during pretraining. We also include MolFormer [Ross et al., 2022], a long-range Transformer architecture designed to capture global context in molecular sequences, as well as TranFoxMol [Gao et al., 2023], a SMILES Transformer that uses fine-grained tokenization schemes.

**Graph-based models** treat molecules as undirected graphs, where atoms are nodes and bonds are edges. Our evaluation includes FPGNN [Cai et al., 2022] (a fingerprint-enhanced GNN variant), D-MPNN [Yang et al., 2019], which propagates information along directed bonds, and GROVER [Rong et al., 2020], a graph Transformer pretrained with both contrastive and contextual prediction objectives. We also consider AttentiveFP [Xiong et al., 2019], which employs gated attention over atom neighborhoods, as well as two popular pretraining strategies from: AttrMasking [Hu et al., 2020b] (attribute prediction) and ContextPred [Hu et al., 2020a] (subgraph context prediction). Additionally, we include an AutoML-tuned GNN pipeline (DeepMol) [Correia et al., 2024] for automated baseline selection and optimization. For 3D geometry-aware baselines, we compare to Uni-Mol [Zhou et al., 2023], a SE(3)-equivariant molecular model that explicitly encodes atomic coordinates and interatomic distances using conformer input. It achieves strong performance on structure-sensitive tasks but is known to be susceptible to conformer variability.

All baselines are implemented using their original codebases or official checkpoints when available, with training and evaluation protocols aligned to ensure fair comparison.

### C.5.2 Datasets & Settings

**MoleculeNet** [Wu et al., 2018]. To evaluate the performance of our method, we utilized benchmark datasets from MoleculeNet, a widely recognized and authoritative resource for molecular property prediction tasks. As Table 12 shows, the selected datasets encompass a diverse range of molecular properties, covering both classification and regression tasks, as well as single-task and multi-task learning scenarios. Specifically, the classification datasets include biophysical and physiological properties, such as drug permeability across the blood-brain barrier (BBBP) and toxicity prediction (Tox21, SIDER, ClinTox), while the regression datasets focus on physical chemistry properties, such as solubility (ESOL) and lipophilicity.

Consistent with the official recommendation, for single-task classification problems, we adopt the **scaffold** split strategy, which ensures that structurally similar molecules are grouped into the same subsets. This approach enhances the robustness of model evaluation by simulating realistic generalization scenarios where models must predict molecular properties for novel scaffolds. In contrast, for multi-task classification and regression problems, we use a **random** split strategy. This is because multi-task models benefit from shared representations across tasks, and regression tasks often have fewer data points, making scaffold splitting too restrictive and leading to insufficient training samples, which is also not fair for the benchmarks.

To assess model performance on classification tasks, we use the area under the receiver operating characteristic curve (AUROC). This metric is preferred because it is insensitive to class imbalance, which is common in molecular datasets. Unlike accuracy, which can be misleading when dealing with imbalanced datasets, AUROC evaluates a model's ability to distinguish between positive and negative classes across different decision thresholds. For regression tasks, we use the root mean square error

(RMSE), which quantifies the magnitude of prediction errors and is well-suited for evaluating the continuous property predictions required in physical chemistry applications.

**Therapeutics Data Commons (TDC) [Huang et al., 2021]**. To further evaluate the robustness and generalizability of MuMo, we include benchmark datasets from the Therapeutics Data Commons (TDC), a comprehensive collection of machine learning-ready datasets for drug discovery and development. The selected datasets span various therapeutic tasks, including drug–target interaction (DTI), ADMET property prediction, and drug response estimation, reflecting the complexity and diversity of real-world pharmaceutical applications.

TDC provides standardized data splits and evaluation protocols, enabling reproducible benchmarking. For all included tasks, we adopt the **official 5-fold scaffold splits** provided by TDC to ensure fair comparison across models. This split strategy partitions molecules based on their core scaffolds, ensuring that test sets contain chemically distinct compounds not seen during training. Such a split is particularly challenging yet more realistic, as it simulates the practical scenario of predicting on structurally novel molecules.

For classification tasks in TDC (e.g., BBB, HIA), we report the area under the ROC curve (AUROC), consistent with MoleculeNet [Wu et al., 2018]. For regression tasks (e.g., LD50, Pgp), we use the mean absolute error (MAE), which is consistent with the metric on the leaderboard. All metrics are reported as the mean and standard deviation across five folds to ensure statistical robustness.

**Datasets Selection.** We selected 14 datasets from TDC because they are part of a well-curated benchmark suite **with standardized leaderboards**, allowing for direct comparison with state-of-the-art models. These datasets focus on ADMET-related tasks, which are critical for drug development, and have been widely adopted in recent multimodal and pretrained molecular modeling studies. Where available, we include leaderboard results for competitive baselines to ensure fairness and transparency in our evaluation.

From MoleculeNet, we selected commonly used datasets for both classification and regression tasks, ensuring broad coverage of molecular properties and compatibility with prior literature. We deliberately exclude datasets such as QM9, which provide precise atomic coordinates for small molecules, as our method treats 3D geometry as auxiliary information rather than a primary input. Moreover, QM9 conformers are computed via energy optimization and are known to be sensitive to conformer noise—an issue we aim to address rather than depend on. Our dataset choices thus prioritize both benchmarking relevance and alignment with the assumptions and goals of our framework.

**Computing Resources.** MuMo requires a minimum of 24 GB GPU memory for fine-tuning and can be trained on a single NVIDIA RTX 4090. The actual training time varies by task depending on the number and size of molecules. Empirically, training on a dataset with 1000 molecules typically takes 10–20 minutes. Larger GPUs or multi-GPU setups can further accelerate training.

**Hyperparameter Settings.** While the overall architecture and training strategy remain consistent, some hyperparameters may vary slightly across different downstream tasks depending on dataset size and task type. We do not enumerate all task-specific configurations here; please refer to our released code for full hyperparameter details and per-task settings.

### C.6 Insight of Representation Learning in Pretraining

To illustrate the layer-wise feature extraction capability of our model, we present a Uniform Manifold Approximation and Projection (UMAP) visualization of the hidden representations across all 16 layers, as shown in Figure 7. This visualization was generated by pooling the high-dimensional hidden states of molecular representations from a set of molecules and projecting them into two dimensions using UMAP [McInnes et al., 2018]. For this analysis, we selected ten distinct molecular scaffolds [Schuffenhauer et al., 2007], each represented by a unique color in the plot to facilitate differentiation.

Scaffold [Schuffenhauer et al., 2007], in cheminformatics, refers to the core chemical structure shared by molecules, typically including the central ring system and key functional groups. Scaffolds are widely used to group structurally similar molecules, as they often correlate with biological or chemical properties. Using scaffolds in this experiment allows us to evaluate the model's ability to capture and separate key structural features, aligning with their importance in drug discovery and molecular design.

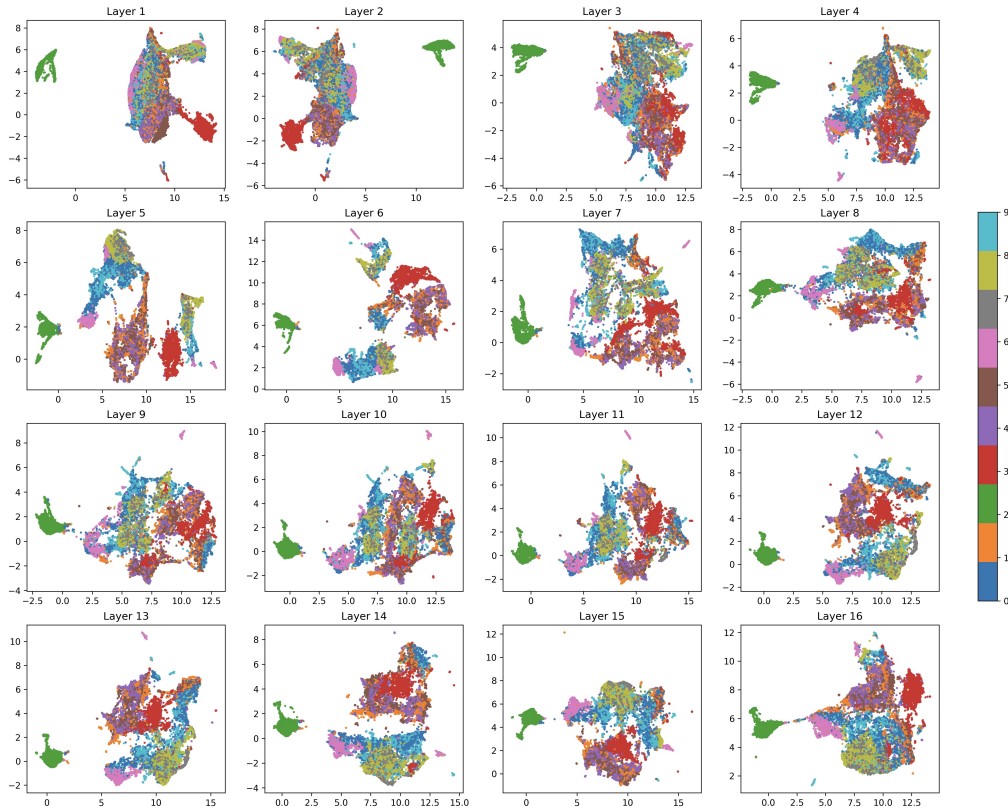

Figure 7: UMAP visualization of the model's sequence hidden representations, where each point represents a molecular scaffold and colors denote 10 distinct scaffold types. The projection illustrates the progression of feature separability and discriminative power across the model's 16 layers.

In our experiment, the first 8 layers of the model are dedicated to SMILES sequence modeling, while layers 9-16 progressively incorporate multimodal fusion through IEA. From the visualization, we observe that in the early layers (1-4), molecules from different scaffolds are mixed together, showing minimal differentiation. As the layers deepen, particularly in layers 5-8, clear boundaries begin to emerge, and molecules from the same scaffold are well-separated, indicating that the model has effectively captured the features of SMILES sequences. Starting from layer 9, as topological and geometric information is gradually integrated, the scaffold separability remains stable. This analysis highlights the rationale behind introducing multimodal fusion in the latter half of the model: by first independently modeling SMILES sequences and the unified representations of topology and geometry, we leverage the strengths of each modality without early-stage interference, ensuring optimal feature extraction for each data structure.

### C.7  Molecular Similarity Analysis

Molecular similarity analysis evaluates the effectiveness of pre-trained embeddings (dissimilarity) by assessing their correlation with molecular dissimilarity and similarity representations, serving as a benchmark for the embeddings' scientific validity and consistency [Ross et al., 2022]. For the experiments, we randomly sampled 20,000 molecules from the ZINC-250k dataset and generated 10,000 random molecule pairs. Each molecule was passed through the pre-trained model to infer its embeddings. For each molecule pair, the Euclidean distance between their embeddings was calculated and then correlated with four molecular dissimilarity and similarity representations: Tanimoto distance, the number of atoms in the Maximum Common Substructure (MCS), Dice similarity, and Cosine similarity. The representations were calculated based on the Morgan fingerprint

(radius=2) of molecules, a widely-used molecular descriptor that encodes structural features into fixed-length binary vectors [Morgan, 1965].

To ensure the reliability of the results, we randomly divided the dataset into five groups and ran each model on these splits, reporting the mean and standard error across the five runs. We selected the ZINC-250k dataset [Gómez-Bombarelli et al., 2018] for this experiment because neither our model nor the baselines were extensively pre-trained on this dataset, ensuring a fair comparison. Additionally, we chose three baseline models that are strong pre-trained models with demonstrated representation capability and robust performance on downstream molecular tasks. These baselines are well-suited for providing molecular embeddings, making them appropriate for evaluating the effectiveness of our approach. Pearson correlation was used to measure the relationship between the Euclidean embedding distance (dissimilarity) and each molecular dissimilarity or similarity representation. A strong correlation (larger absolute value of the correlation coefficient) indicates that the model's embeddings better capture the chemical similarity between molecules.

The four representations used in this analysis provide comprehensive and scientifically validated measures of molecular dissimilarity or similarity:

**(1) Tanimoto distance (dissimilarity).** Widely used in cheminformatics for virtual screening and chemical similarity searches, Tanimoto distance derives from the Tanimoto similarity, a fingerprint-based measure of overlap between two molecular representations. Let $f_A$ and $f_B$ be binary or real-valued fingerprint vectors for two molecules. The Tanimoto similarity $T$ is given by:

$$T(f_A, f_B) = \frac{f_A \cdot f_B}{\|f_A\|^2 + \|f_B\|^2 - f_A \cdot f_B}. \tag{23}$$

We define the Tanimoto distance $D$ as

$$D(f_A, f_B) = 1 - T(f_A, f_B). \tag{24}$$

A larger $D(f_A, f_B)$ (closer to 1) indicates greater dissimilarity, whereas smaller values (close to 0) signify higher similarity [Rogers and Hahn, 2010].

**(2) Number of atoms in Maximum Common Substructure (similarity).** Given two molecules $\text{Mol}_1$ and $\text{Mol}_2$, the MCS identifies the largest subgraph that is isomorphic in both. We record the number of atoms in this common subgraph as:

$$\text{MCS}(\text{Mol}_1, \text{Mol}_2) = \left|\text{MaxSub}(\text{Mol}_1, \text{Mol}_2)\right|, \tag{25}$$

where $\text{MaxSub}(\text{Mol}_1, \text{Mol}_2)$ is the maximum common substructure (in terms of atomic count). This metric provides an interpretable measure of the structural overlap (i.e., backbone or scaffold similarity) between two molecules [Raymond and Willett, 2002].

**(3) Dice Similarity (similarity).** Similar to the Tanimoto measure, Dice similarity emphasizes shared fingerprint features. For two fingerprint vectors $f_A$ and $f_B$, it is defined as:

$$\text{Dice}(f_A, f_B) = \frac{2(f_A \cdot f_B)}{\|f_A\|^2 + \|f_B\|^2}. \tag{26}$$

The Dice similarity often accentuates intersecting bits more strongly than Tanimoto, making it sensitive to certain molecular distributions [Willett et al., 1998].

**(4) Cosine Similarity (similarity).** Another vector-based metric is cosine similarity, which captures the cosine of the angle between two fingerprint vectors:

$$\text{Cosine}(f_A, f_B) = \frac{f_A \cdot f_B}{\|f_A\|\|f_B\|}. \tag{27}$$

This measure remains robust to scaling, focusing on the orientation of the vectors rather than their magnitude. A higher cosine similarity indicates a greater proportion of shared features between the two molecules [Bender and Glen, 2004].

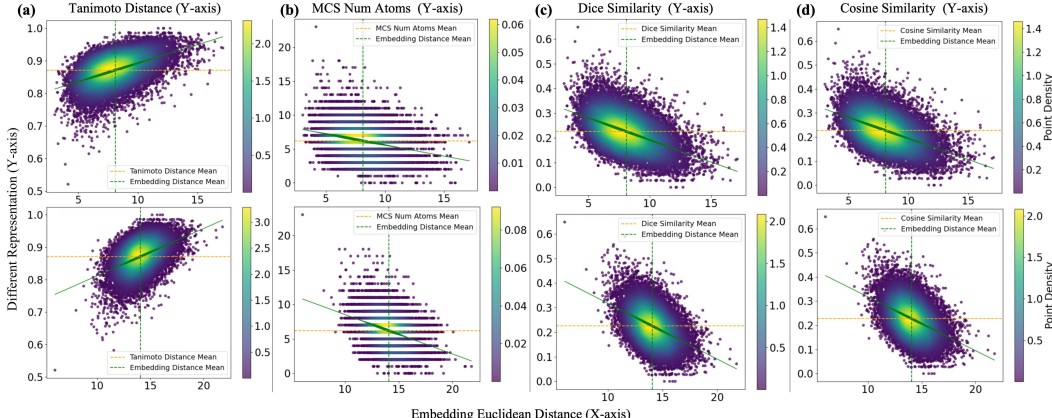

Figure 8: Visualization of molecular similarity results of MuMo (first row) and MoLFormer (second row). The plots show the relationship between embedding Euclidean distance and four molecular dissimilarity and similarity representations: (a) Tanimoto distance; (b) number of atoms in maximum common substructure (MCS); (c) Dice similarity; (d) cosine similarity for our model (first row) and MoLFormer (second row). Each plot includes scatter points, and a linear regression trend line (green).

**Euclidean Distance (dissimilarity).**    To assess the performance of the model-generated embeddings, we compute the Euclidean distance between the learned molecular representations (inferences from pretrained model). Given two embeddings $z_A, z_B \in \mathbb{R}^d$, the Euclidean distance is defined as:

$$d_E(z_A, z_B) = \|z_A - z_B\|. \tag{28}$$

Since Euclidean distance measures the separation between embeddings in the latent space, a larger $d_E$ value indicates greater dissimilarity, while a smaller $d_E$ suggests higher molecular similarity.

**Correlation with Molecular Similarity Representations.**    To evaluate how well the embedding space aligns with established molecular similarity measures, we compute the Pearson correlation coefficient between Euclidean distance $d_E$ and four molecular similarity/dissimilarity representations:

- **Tanimoto Distance** $D$ (dissimilarity): Positively correlated with $d_E$, as molecules with greater Tanimoto distance should have larger Euclidean distances in the embedding space.

- **Maximum Common Substructure (MCS) Size** (similarity): Negatively correlated with $d_E$, as molecules sharing larger common substructures should be mapped closer together.

- **Dice Similarity** (similarity): Negatively correlated with $d_E$, since a higher Dice similarity implies molecular resemblance.

- **Cosine Similarity** (similarity): Negatively correlated with $d_E$, as similar molecules should have embeddings with smaller Euclidean distances.

To quantify model performance, we compute the **Pearson absolute correlation coefficient** between $d_E$ and these four representations. A higher absolute correlation indicates that the learned embedding space is more aligned with traditional molecular dissimilarity or similarity representations, suggesting better representation learning.

Based on the results presented in Table 13, our model demonstrates competitive performance across the selected pre-trained models, achieving the highest correlation among 3/4 representations, with values approaching 0.5. This indicates a moderate correlation between the embedding Euclidean distance and molecular dissimilarity and similarity representations. The absence of a strong correlation suggests that our embeddings capture a broader range of molecular features beyond the structural similarities reflected in the selected representations.

Notably, MoLFormer proves to be a strong competitor, as evidenced by its comparable performance. It is important to highlight that in their paper [Ross et al., 2022], significantly higher correlation values, approaching 0.7, were reported. However, this can be attributed to their use of the same dataset,

Table 13: Experiemnt results of molecular similarity analysis. The Absolute Pearson correlation coefficient (between Euclidean distance and the four dissimilarity/similarity representations) is reported as the metric. Values represent the mean and standard deviation over five runs.

| MODEL | TANIMOTO DISTANCE | MCS NUM ATOMS | DICE SIMILARITY | COSINE SIMILARITY |
|---|---|---|---|---|
| MOLFORMER | $0.451_{(0.012)}$ | $\mathbf{0.363}_{(0.013)}$ | $0.453_{(0.007)}$ | $0.445_{(0.013)}$ |
| CHEMBERTA-2 | $0.338_{(0.008)}$ | $0.272_{(0.009)}$ | $0.339_{(0.013)}$ | $0.334_{(0.013)}$ |
| MOLBERT | $0.431_{(0.006)}$ | $0.275_{(0.010)}$ | $0.437_{(0.006)}$ | $0.414_{(0.006)}$ |
| **MUMO** | $\mathbf{0.490}_{(0.006)}$ | $0.290_{(0.010)}$ | $\mathbf{0.498}_{(0.008)}$ | $\mathbf{0.487}_{(0.006)}$ |

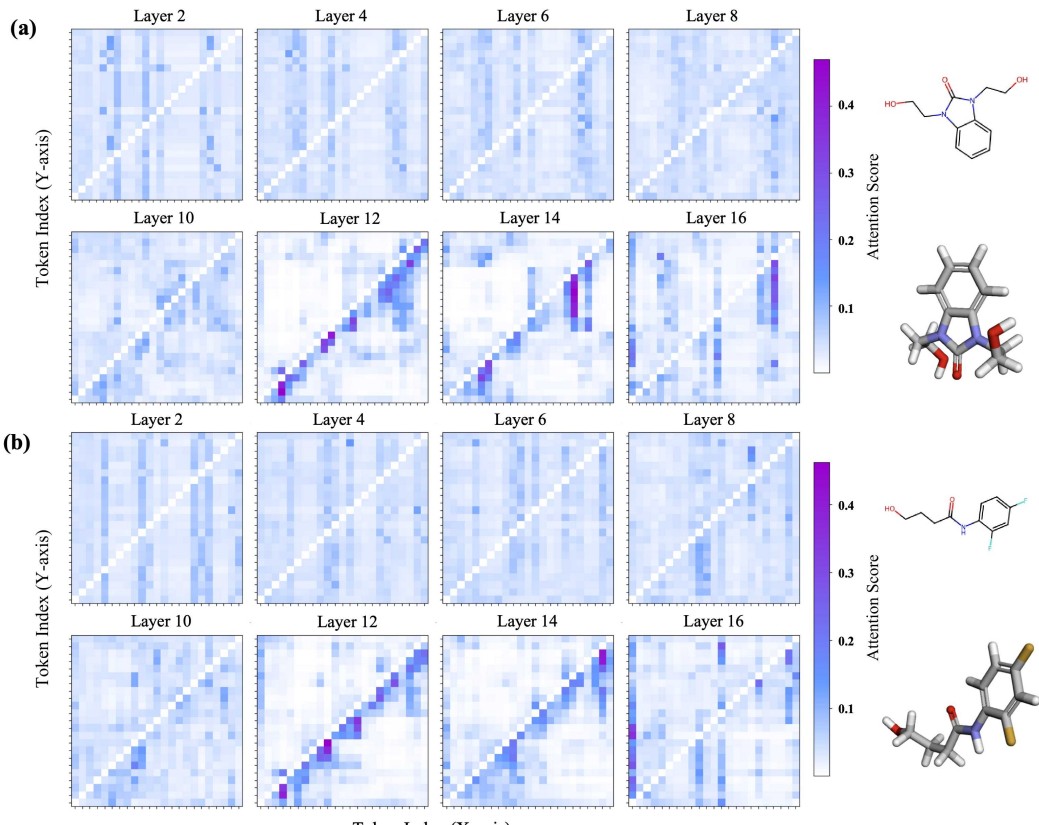

Figure 9: Attention visualization of MuMo for two molecules.
(a) O=C1N(CCO)C2=CC=CC=C2N1CCO and (b) O=C(CCCO)NC1=CC=C(F)C=C1.
Each molecule is analyzed across all 16 layers, with attention scores plotted every two layers. The MuMo model starts interacting with GGFN information from layer 9.

10M-PubChem [Kim et al., 2016], for both pre-training and evaluation, which likely introduces dataset overlap and inflates the results.

For a better comparison between MoLFormer and our model, we visualize results from both models in Figure 8. The first row of the figure represents the visualization for our model, while the second row corresponds to MoLFormer. Each subplot shows the relationship between the Euclidean embedding distance and a molecular dissimilarity or similarity representation. For our model, the distributions appear more elongated and flattened along the diagonal, indicating a stronger correlation. In contrast, the plots for MoLFormer exhibit more rounded distributions, reflecting weaker correlations. This difference suggests that our model's embeddings are more effective, as evidenced by the stronger alignment with the molecular dissimilarity and similarity representations.

Table 14: Computing cost comparison in the pretraining stage. We compare model scale, dataset size, and estimated pretraining time across different molecular foundation models.

| MODEL | SCALE | PRETRAIN DATASET SIZE | PRETRAIN TIME |
|---|---|---|---|
| MOLFORMER-BASE | 44.28M | 1.1B | 16×V100-32G, EST. 104H |
| MOLFORMER-XL | 86.75M | 1.1B | 16×V100-32G, 208H |
| UNIMOL | 47.61M | 19M | 4×V100-32G, 20H |
| UNIMOL-V2 | 1.1B | 884M | 64×A100-80G, EST. 2 WEEKS |
| MUMO | 505M | 1.6M | 4×A100-80G, 5H |

Table 15: Computing efficiency comparison with Uni-Mol-v2. MuMo achieves comparable performance with substantially fewer parameters, less pretraining data, and drastically reduced compute time.

| MODEL | MODEL SIZE | PRETRAIN DATA SIZE | COMPUTING RESOURCES |
|---|---|---|---|
| UNI-MOL-V2 | 1.1B | 884M FROM ZINC20 | 64×A100-80G, EST. 1–3 WEEKS |
| **MUMO** | 505M | 1.6M FROM CHEMBL | 4×A100-80G, 5 HOURS |

## C.8 Attention Analysis for Fusion Effectiveness

To evaluate the effectiveness of core stream fusion in our model, we visualized the self-attention scores across the 16 main layers of the MuMo model (comprising 8 Attention-Mamba layers and 8 multimodal fusion layers) when inferring two molecules randomly selected from the ZINC-250k dataset [Gómez-Bombarelli et al., 2018], as shown in Figure 9. Attention scores were plotted every two layers, providing a detailed view of how the model evolves through its hierarchical feature extraction process. This visualization enables us to track the progression of contextual relationships captured by the sequence flow along with its interaction with the structural fusion module using the Progressive Injection (PI) method.

In the first eight layers, the model primarily focuses on self-modeling within the sequence flow without incorporating topology or geometry information. During this stage, attention scores exhibit a relatively uniform distribution, indicating limited contextual understanding and a lack of meaningful interatomic relationships. Starting from layer 9, the model begins interacting with the outputs of the structural fusion module, starting the hierarchical injection with other structural modalities. This interaction enables the model to capture more localized information, as evidenced by the increased attention weights near the diagonal in the fifth subfigure for both molecules in Figure 9(a) and 9(b). These highlights correspond to strong pairwise dependencies within local regions of the molecular structure. In subsequent layers, the sequence flow continues to interact with topology and geometry flows while independently refining its representations. By the final layer, the model demonstrates the ability to capture long-range contextual relationships, as seen in the attention scores for distant atom pairs. Additionally, the attention score of the [$GLK$] token (the first token) becomes significantly stronger through the last 8 layers, showing it has captured rich representation in the molecular by the Progressive Injection (PI) method, which enhances the model's capacity for robust global representations. This progression underscores the model's ability to integrate multimodal information effectively.

## C.9 Training Cost Analysis

**High efficiency in pretraining.** As shown in Table 14, MuMo is highly efficient in the pretraining stage, requiring only 5 hours on a 1.6M-scale dataset using 4×A100 GPUs, significantly less than prior models that consume days or weeks but got better performance. In Table 15, we compare the pretraining cost to the UniMol-v2 [Ji et al., 2024] model, showing our significant efficiency in pretraining stage.

**High efficiency in finetuning.** In the finetuning stage (Table 16), even on our largest benchmark dataset QM9 (133k samples), MuMo completes training in just 3.3 hours with modest GPU memory (14 GB per card). For smaller datasets (few thousand samples), training typically finishes in under 30 minutes. These results demonstrate that MuMo offers strong performance without prohibitive resource demands, making it practical for real-world use.

Table 16: Computing cost on QM9 in the finetuning stage. We report the hardware configuration, dataset statistics, and training runtime for fine-tuning MuMo (505M) on QM9.

| ITEM | VALUE |
|---|---|
| DATASET | QM9 |
| SAMPLE COUNT | 133,855 |
| CPU CORES | 32 |
| CPU | INTEL(R) XEON(R) PLATINUM 8480+ |
| GPU COUNT | 2 |
| GPU TYPE | NVIDIA A100-SXM4-80GB |
| MODEL PARAMETERS | 505M |
| TOTAL FLOPS | 65,218,081 GF |
| EPOCHS | 10 |
| GLOBAL BATCH SIZE | 64 |
| MAX GPU MEMORY USAGE | 14G PER GPU |
| DATA PREPROCESS TIME | 5 MIN |
| TRAINING TIME | 3H 21M 51S |

# D    Extended Experiments and Ablation Studies

To comprehensively evaluate the effectiveness of our proposed method and share our training experience with other researchers, particularly the innovations in the structural fusion pipeline and PI (Progressive Injection), we conducted an extensive series of ablation studies. These experiments not only validate the contributions of our approach but also provide valuable insights into training strategies and model development. Our analysis systematically investigates the impact of various components and innovations, including pretraining strategies, multimodal interaction mechanisms, sequence data types, pretraining datasets, model capacity, and some hyperparameter configurations.

To ensure the rigor and scientific validity of our experiments, we began by focusing on multimodal fusion. Specifically, starting from the best-performing model, we progressively removed one modality at a time to establish three baseline models: the complete model (our proposed method), a model fusing sequences and 2D graphs, and a sequence-only model. Based on comparisons of these three baselines, we further conducted ablation studies by reducing or modifying individual modules within each model to highlight the contributions of different components to the final performance. In this section, we primarily use five medium-sized datasets from MoleculeNet, applying the recommended split methods: scaffold split for BACE and BBBP, and random split for the others.

## D.1    Contribution of Multimodal Combinations

In this section, we demonstrate the effectiveness of the proposed MuMo model and use the best model as a baseline for ablation experiments. The full MuMo model represents our unaltered approach. To investigate the impact of different modalities, we conducted the following variations: (1) removing geometric information and the fusion of geometric and 2D topological flow, and (2) utilizing only SMILES sequences for modeling without using any fusion methods. These two models are labeled as Ablation Baseline 1 and 2 (AB1 and AB2) for other ablation experiments, with MuMo itself designated as Ablation Baseline 3 (AB3). All other conditions were kept identical. After the same pretraining process, the models were evaluated on five datasets, comprising three classification tasks and two regression tasks from MoleculeNet, to ensure a comprehensive assessment.

From the results presented in Table 17, it is evident that the model without geometric information experiences varying degrees of performance degradation across datasets, with a particularly significant drop observed on the BACE dataset. This indicates that geometric information plays a critical role in capturing subtle molecular properties for certain classification tasks. Comparing the SMILES-only model with the sequence+2D graph fusion model reveals minimal differences in performance on classification tasks but a substantial decline in regression tasks. This suggests that accurate numerical predictions in regression heavily depend on the model's ability to capture the full 2D topological structure of molecules. These findings underscore the indispensable roles of both 2D topology and 3D modeling in molecular representation learning.

Table 17: Ablation results on multimodal combinations. BACE, BBBP, and Clintox are classification tasks (AUROC, higher is better), while ESOL and LIPO are regression tasks (RMSE, lower is better). The three methods include AB1: SMILES Only (sequence-based), AB2: SMILES + Graph (sequence and topology), and MuMo (sequence, topology, geometry).

| | CLASSIFICATION | | | REGRESSION | |
|---|---|---|---|---|---|
| METHOD | BACE | BBBP | CLINTOX | ESOL | LIPO |
| AB1 - SMILES ONLY | $0.798_{(0.015)}$ | $0.931_{(0.005)}$ | $0.958_{(0.021)}$ | $1.793_{(0.055)}$ | $0.844_{(0.033)}$ |
| AB2 - SMILES + GRAPH | $0.780_{(0.022)}$ | $0.946_{(0.008)}$ | $0.971_{(0.015)}$ | $0.597_{(0.051)}$ | $0.596_{(0.035)}$ |
| AB3 - **MuMo** | $\mathbf{0.849}_{(0.014)}$ | $\mathbf{0.957}_{(0.011)}$ | $\mathbf{0.985}_{(0.011)}$ | $\mathbf{0.536}_{(0.061)}$ | $\mathbf{0.577}_{(0.027)}$ |

Table 18: Ablation results of pretraining contribution. Results are presented with AUROC for classification tasks (BACE, BBBP, CLINTOX) and RMSE for regression tasks (ESOL, LIPO). Standard deviations are reported over three runs.

| | | CLASSIFICATION | | | REGRESSION | |
|---|---|---|---|---|---|---|
| PRETRAINING | FINETUNING | BACE | BBBP | CLINTOX | ESOL | LIPO |
| $\surd$ | $\surd$ | $\mathbf{0.849}_{(0.014)}$ | $\mathbf{0.957}_{(0.011)}$ | $\mathbf{0.985}_{(0.011)}$ | $\mathbf{0.536}_{(0.061)}$ | $\mathbf{0.577}_{(0.027)}$ |
| $\times$ | $\surd$ | $0.807_{(0.003)}$ | $0.930_{(0.002)}$ | $0.928_{(0.015)}$ | $2.568_{(0.030)}$ | $0.887_{(0.017)}$ |

## D.2 Pretraining Ablation Studies

Recent works, such as MolBERT [Fabian et al., 2020] and MoLFormer [Ross et al., 2022], have shown that the effectiveness of large-scale molecular pretraining relies heavily on massive datasets, often exceeding 1 billion molecules. In contrast, our proposed method, MuMo, achieves superior performance on downstream tasks with significantly less pretraining data. By leveraging only 1.6M molecules from the ChEMBL dataset [Gaulton et al., 2012], MuMo demonstrates that pretraining on a much smaller scale can be both efficient and highly effective. Furthermore, ablation studies on our pretraining approach validate the robustness and efficiency of MuMo, highlighting its ability to extract meaningful representations even with limited data.

### D.2.1 Pretraining Effectiveness

To evaluate the necessity of pretraining, we compared two models: one fully pretrained on the 1.6M ChEMBL dataset and the other directly fine-tuned on downstream tasks without any pretraining (zero-shot fine-tuning). This comparison isolates the impact of pretraining by examining its influence on downstream performance and training dynamics. Both models share identical architectures, hyperparameters, and training configurations during fine-tuning to ensure a fair and controlled evaluation.

The results in Table 18 demonstrate the critical role of pretraining in enhancing molecular property prediction, with particularly significant effects on regression tasks. For classification tasks such as BACE and BBBP, pretraining yields consistent but modest improvements, increasing AUROC scores by 5.9% and 3.5%, respectively. These gains suggest that pretraining effectively captures foundational molecular patterns and functional group features, which improve generalization. However, its impact is most pronounced in regression tasks like ESOL and LIPO, where RMSE reductions from 2.568 to 0.536 and 0.887 to 0.577. This reflects the importance of pretraining in encoding high-resolution spatial and geometric molecular features, essential for accurately modeling continuous properties such as solubility and lipophilicity.

The differential impact of pretraining stems from the varying nature of the tasks. Classification tasks rely on discrete structural patterns, which fine-tuning alone can partially capture, whereas regression tasks demand precise geometric and spatial modeling, making pretraining indispensable. By leveraging the Unified Graph and molecular segmentation, pretraining enables the model to encode both local interactions and global molecular topology effectively. These findings underscore the necessity of pretraining for complex molecular representations, particularly in tasks requiring high-resolution structural information.

Table 19: Ablation results on pretrain datasets. Downstream task performance comparison of models using different pretraining datasets. The metrics for BACE, BBBP, and Clintox are AUROC (higher is better), while ESOL and LIPO use RMSE (lower is better). The models include SMILES-Only (AB1, sequence-based), and MuMo (sequence, topology, and geometry).

| | CLASSIFICATION | | | REGRESSION | |
|---|---|---|---|---|---|
| SIZE OF PRETRAIN DATASETS | BACE | BBBP | CLINTOX | ESOL | LIPO |
| 10M PUBCHEM - SEQUENCE ONLY | $0.849_{(0.036)}$ | $0.856_{(0.037)}$ | $0.677_{(0.122)}$ | $2.543_{(0.849)}$ | $0.797_{(0.125)}$ |
| 1.6M CHEMBL - SEQUENCE ONLY | $0.798_{(0.015)}$ | $0.931_{(0.005)}$ | $0.958_{(0.021)}$ | $1.793_{(0.055)}$ | $0.844_{(0.033)}$ |
| MIXDATASETS - MUMO | $0.853_{(0.035)}$ | $0.956_{(0.038)}$ | $0.972_{(0.110)}$ | $0.607_{(1.058)}$ | $0.604_{(0.150)}$ |
| 10M PUBCHEM - MUMO | $0.784_{(0.038)}$ | $0.946_{(0.038)}$ | $0.979_{(0.134)}$ | $0.635_{(0.930)}$ | $0.611_{(0.104)}$ |
| **1.6M CHEMBL - MUMO** | $\mathbf{0.849}_{(0.014)}$ | $\mathbf{0.957}_{(0.011)}$ | $\mathbf{0.985}_{(0.011)}$ | $\mathbf{0.536}_{(0.061)}$ | $\mathbf{0.577}_{(0.027)}$ |

### D.2.2 Scale and Source of Pretrain Datasets

To evaluate the influence of the scale and type of pretraining datasets on downstream molecular property prediction tasks, we conducted experiments using models pretrained on datasets of varying sizes and sources. Specifically, we compared two model architectures: SMILES-Only (AB1), which relies solely on sequence representations, and MuMo, which is our best model. Pretraining datasets included PubChem (10M molecules) [Kim et al., 2016], ChEMBL (1.6M molecules) [Gaulton et al., 2012], and a curated "MixDatasets", which integrates these three datasets. We assessed model performance on five benchmark datasets (BACE, BBBP, Clintox, ESOL, and LIPO), covering both classification and regression tasks. This experiment is necessary to investigate how pretraining data characteristics—such as scale, diversity, and feature richness—affect the generalization and robustness of models in molecular property prediction, which is critical for advancing drug discovery and materials science applications.

The results in Table 19 highlight the significant impact of pretraining dataset scale and type on downstream task performance. Models relying solely on SMILES sequences perform reasonably well on certain tasks, such as the AB1 (sequence only) model trained on the 10M PubChem dataset achieving an AUROC of 0.849 on BACE. However, their performance on other tasks is relatively limited. This suggests that sequence-only representations may lack the expressiveness needed for diverse property predictions. In contrast, the introduction of the multimodal fusion strategy in the MuMo models substantially improves performance across tasks. For instance, the MuMo model trained on the 1.6M ChEMBL dataset achieves leading results on Clintox and ESOL, indicating superior capability in toxicity and solubility predictions.

Interestingly, while the performance differences between models pre-trained on different datasets are not particularly large, the 1.6M ChEMBL dataset consistently outperforms others, especially when used with the MuMo model. This can be attributed to ChEMBL's curated nature, as it specifically focuses on bioactive molecules with experimentally validated activity against biological targets. This high-quality, domain-relevant data likely provides more meaningful molecular patterns and task-specific signals for the model to learn. In contrast, larger datasets like the 10M PubChem, which contain more diverse and potentially noisy molecular structures, may introduce heterogeneity that dilutes the relevance of features for downstream tasks. The ChEMBL dataset, combined with our multimodal representation learning approach, allows the model to effectively generalize and achieve strong performance across various tasks. This highlights the importance of data quality and relevance over sheer size in molecular pretraining.

### D.2.3 Pretraining Strategy

This experiment explores the impact of different pretraining tasks and pooling methods on the performance of the sequence-based ablation baseline model (AB1). Two pretraining tasks are considered: Masked Language Modeling (MLM), which involves masking a portion of the input sequence and training the model to predict the masked tokens based on the context, and Next Token Prediction (NTP), where the model predicts the next token in the sequence to learn sequential dependencies. For MLM, two pooling methods are tested during the fine-tuning phase: mean pooling, which averages all token embeddings in the sequence to form the final representation, and GTK token pooling, which uses the embedding of the first special token (GTK) as the final representation. The

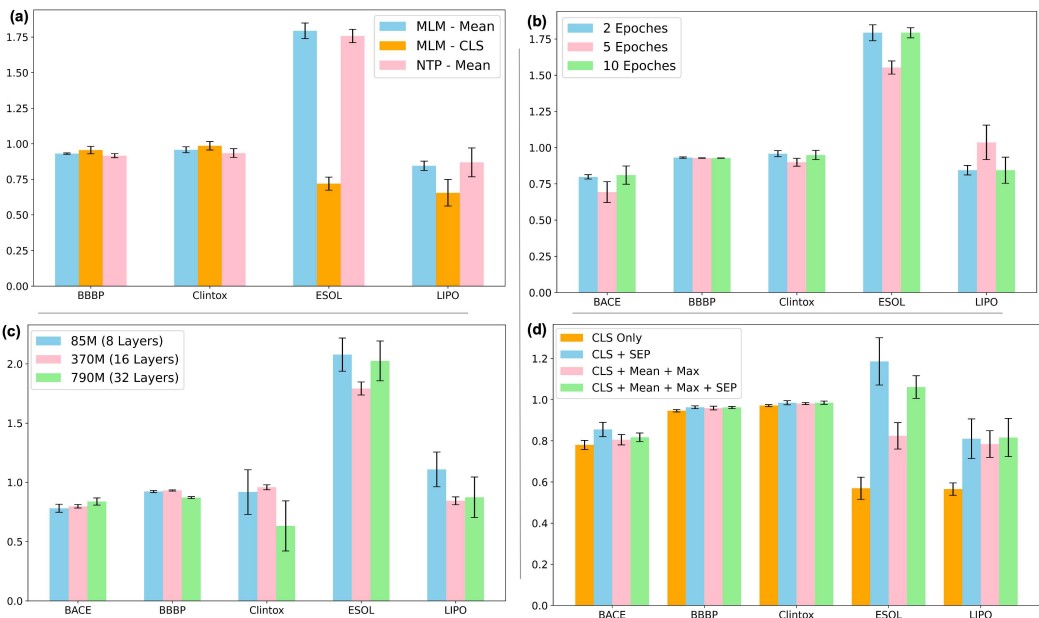

Figure 10: Ablation results on pretraining Epochs, pooling methods, model size, and pretraining strategies. (a) The impact of pretraining epochs on model performance. (b) The effect of pooling methods during fine-tuning. GTK (first token), SEP (last token), mean pooling, max pooling, and combined methods (GTK + Mean + Max + SEP) are compared. (c) Influence of model size. Compared to the larger model using 32 layers and the smaller model using 8 layers. (d) The effect of pretraining strategies - Masked Language Modeling (MLM) and Next Token Prediction (NTP) with pooling methods. Metrics: AUROC (higher is better) for classification datasets (BACE, BBBP, Clintox) and RMSE (lower is better) for regression datasets (ESOL, LIPO).

performance of these variations is evaluated across five datasets, with classification tasks (BBBP, Clintox, and ESOL) using AUROC and regression tasks (LIPO) using RMSE. This setup aims to assess how pretraining tasks and pooling strategies influence downstream performance.

The results for Figure 10 (a) show that the NTP pretraining task generally underperforms compared to MLM, achieving the lowest overall performance across the evaluated datasets. This can be attributed to the nature of the task itself. NTP focuses on predicting the next token in the sequence, which emphasizes sequential dependencies but lacks the capacity to capture broader structural information about the molecule. In contrast, MLM allows the model to learn richer contextual relationships by considering both preceding and following tokens, which is crucial for representing the complex structural and chemical properties of molecules. By masking random tokens, MLM forces the model to integrate information across the entire sequence, enabling a more holistic understanding of molecular features.

Among the pooling methods, GTK token pooling consistently outperforms mean pooling. This difference can be explained by the way each method aggregates information. GTK token pooling uses a dedicated GTK token trained explicitly to encode a global representation of the input sequence, allowing it to serve as a focused summary of the molecule's overall properties. In contrast, mean pooling averages the embeddings of all tokens, which can dilute critical information and make it harder for the model to distinguish important structural or functional features. The superior performance of GTK pooling suggests that the specialized GTK representation is better suited for tasks requiring a compact yet informative molecular encoding.

### D.2.4 Pretraining Epochs

Results from Figure 10(b) show that the difference between 2 and 5 epochs of pretraining is negligible across all tasks, suggesting that the model converges quickly. However, extending pretraining to 10

epochs negatively impacts regression tasks, particularly ESOL, where performance drops significantly. This trend aligns with the loss curves from Figure 4, which demonstrate that the model converges rapidly and maintains stable loss values with minimal fluctuation. Prolonged pretraining provides diminishing returns and can even degrade performance on certain tasks, likely due to overfitting or excessive adaptation to the pretraining data. This behavior highlights a key advantage of our model: its ability to achieve strong performance with minimal pretraining epochs, making it computationally efficient and robust.

### D.3 Impact of Model Size

Results from Figure 10(c) demonstrate that model size has a significant but nuanced impact on performance. Smaller models, such as the 85M parameter model with 8 layers, generally underperform due to limited capacity to capture the complexity of molecular representations. However, excessively large models, like the 790M parameter model with 32 layers, also fail to consistently outperform the 370M parameter model with 16 layers, particularly on tasks like Clintox and LIPO. This suggests that overly large models may suffer from overfitting or diminishing returns when scaling parameters. The 370M model strikes a balance between capacity and generalization, achieving competitive performance across both classification (e.g., BBBP) and regression (e.g., ESOL) tasks. This indicates that there is an optimal model size for capturing molecular structure and properties without incurring computational inefficiency or overfitting risks.

### D.4 Impact of Pooler Methods in Finetuning

This experiment shown in Figure 10(d) evaluates the impact of different pooling strategies during the fine-tuning phase on downstream task performance. For regression tasks the addition of the SEP token increases the values, indicating a negative effect on regression accuracy. Similarly, the inclusion of mean and max pooling does not outperform the simpler GTK-only (global token) approach, as the performance remains worse when combining multiple pooling strategies. This outcome underscores the importance of the GTK token for providing a compact and enriched global representation, which is particularly critical for regression tasks. Regression tasks are more sensitive to the accuracy of global representations and demand higher precision in capturing geometric and topological information. The MuMo model's multimodal fusion process explicitly strengthens the GTK token through injected information during pretraining, making the GTK token crucial for achieving accurate global molecular property predictions in regression tasks.

For classification tasks, the performance differences across pooling methods are relatively minor. This is because the Injection-Enhanced mechanism during pretraining plays a dominant role in strengthening the GTK token with rich multimodal information, surpassing the contribution of the symmetric cross-attention interaction. As a result, the GTK token alone provides a sufficiently strong and compact representation for coarse-grained classification tasks. The addition of other pooling methods, such as SEP, mean, and max pooling, adds minimal benefit, as the GTK token already captures the essential sequence-level features needed for classification.

### D.5 Impact of Sequence Data Type

This experiment evaluates the performance of a sequence-only model (AB1) using three different molecular sequence representations: SMILES, SELFISH, and Morgan fingerprint while keeping all other conditions identical. The goal is to understand how different sequence data types influence the performance of molecular representation learning.

SMILES is a widely used linear notation that encodes molecular structures as text strings, capturing atom connectivity and bond information. SELFISH is a sequence representation derived from SMILES, optimized for specific tasks by restructuring the sequence to enhance information extraction. Morgan fingerprint, in contrast, is a fixed-length vector representation generated from molecular graphs, capturing substructural patterns but lacking the sequential structure of SMILES.

Table 20 shows that SMILES achieves the best overall performance, particularly excelling in classification tasks. While SMILES demonstrates some limitations in regression tasks, such as ESOL, its performance remains competitive, showcasing its versatility as a molecular representation. SELFISH also performs well but slightly trails SMILES, particularly in classification tasks, while Morgan

Table 20: Ablation results on different sequence data types. Performance comparison of models using SMILES, SELFISH, and Morgan fingerprint. The metrics for dataset BBBP and Clintox are AUROC (higher is better), while ESOL and LIPO use RMSE (lower is better).

| | CLASSIFICATION | | REGRESSION | |
|---|---|---|---|---|
| SEQUENCE DATA TYPE | BBBP | CLINTOX | ESOL | LIPO |
| **SMILES (AB1)** | $\mathbf{0.931}_{(0.005)}$ | $\mathbf{0.958}_{(0.021)}$ | $1.793_{(0.055)}$ | $\mathbf{0.844}_{(0.033)}$ |
| SELFISH | $0.914_{(0.020)}$ | $0.910_{(0.151)}$ | $1.793_{(0.436)}$ | $\mathbf{0.844}_{(0.018)}$ |
| MORGAN FINGERPRINT | $0.903_{(0.016)}$ | $0.707_{(0.135)}$ | $\mathbf{0.939}_{(0.480)}$ | $0.880_{(0.019)}$ |

Table 21: Ablation results on data type combinations for multimodal modeling. Full MuMo is our best model, which combines SMILES, 2D graph, and 3D Geometry information. The metrics for BACE, BBBP, and Clintox are AUROC (higher is better), while ESOL and LIPO use RMSE (lower is better).

| | CLASSIFICATION | | | REGRESSION | |
|---|---|---|---|---|---|
| MULTIMODAL CHOICE | BACE | BBBP | CLINTOX | ESOL | LIPO |
| SMILES+GRAPH | $0.780_{(0.022)}$ | $0.946_{(0.008)}$ | $0.971_{(0.015)}$ | $0.597_{(0.051)}$ | $0.596_{(0.035)}$ |
| SMILES+FINGERPRINT+GRAPH | $0.745_{(0.026)}$ | $0.919_{(0.007)}$ | $0.969_{(0.012)}$ | $0.646_{(0.055)}$ | $0.633_{(0.033)}$ |
| FINGERPRINT+GRAPH+GEOMETRY | $0.824_{(0.031)}$ | $0.908_{(0.007)}$ | $0.996_{(0.012)}$ | $1.030_{(0.055)}$ | $0.845_{(0.034)}$ |
| **FULL MUMO** | $\mathbf{0.849}_{(0.014)}$ | $\mathbf{0.957}_{(0.011)}$ | $\mathbf{0.985}_{(0.011)}$ | $\mathbf{0.536}_{(0.061)}$ | $\mathbf{0.577}_{(0.027)}$ |

fingerprint struggles in classification tasks and shows inconsistent results in regression tasks. This suggests that the sequential information in SMILES provides a more robust foundation for general molecular property prediction across both task types.

Results from Table 21 demonstrate that incorporating Morgan fingerprint into the multimodal combinations does not significantly improve performance and, in some cases, leads to a slight degradation. For instance, the combination of SMILES + Fingerprint + Graph underperforms compared to SMILES + Graph alone, particularly on regression tasks. This suggests that the Morgan fingerprint, being a fixed-length vector representation, may not effectively complement other modalities like 2D topology graphs and 3D geometry, which provide richer structural and spatial information. The best results are achieved by the MuMo model, which combines SMILES, 2D graph, and 3D geometry, highlighting the importance of these complementary modalities in capturing molecular features comprehensively.

### D.6 Impact of Key Modules

In this section, we present five modules that have been shown to enhance the representational effectiveness of the model during training. To demonstrate the impact of these modules and share our insights with peers, we conducted a series of experiments. Under identical conditions, we modified only one module of the model at a time and evaluated its performance on the corresponding datasets.

Table 22: Ablation results of three small modules. The effect of the substructure-level tokenizer, bi-attention, and different graph processing modules (MPNN and GCN). The metrics for BACE, BBBP, and Clintox are AUROC (higher is better), while ESOL and LIPO use RMSE (lower is better).

| MODULE | DETAIL | AVG. DROP - CLASSIFICATION | AVG. DROP - REGRESSION |
|---|---|---|---|
| ATTENTION | SELF-ATTENTION | 0.00% | 0.00% |
| | BI-ATTENTION | 0.88% | -18.68% |
| GRAPH PROCESS | MPNN | 0.00% | 0.00% |
| | GCN | -4.08% | -11.37% |
| TOKENIZER | SUBSTRUCTURE-LEVEL | 0.00% | 0.00% |
| | CHARACTER-LEVEL | -4.28% | -9.05% |

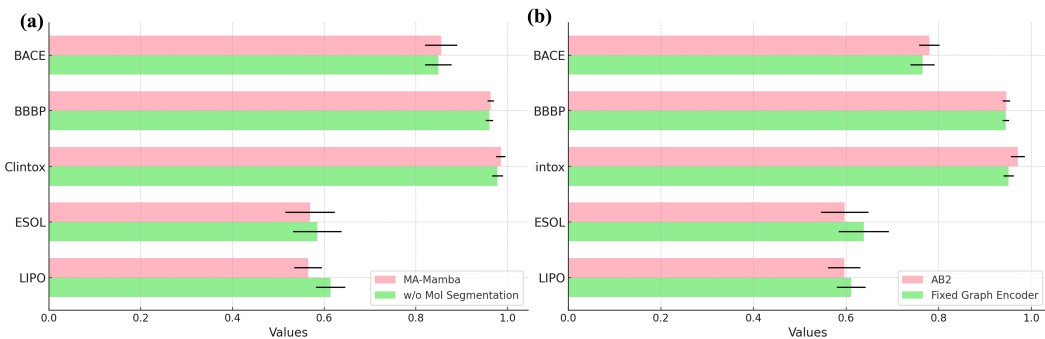

Figure 11: Impact of (a) molecular segmentation and (b) graph learning strategy. The metrics for BACE, BBBP, and Clintox are AUROC (higher is better), while ESOL and LIPO use RMSE (lower is better).

**Substructure Partitioning.** In this study, we proposed the Geometry Substructure Partitioning method to capture molecular features at the geometric substructure level. To evaluate its importance, we conducted an ablation study where the MuMo model served as the baseline, and the experimental group removed the fusion of 2D graph and geometry substructure modeling, retaining only the interaction of their global information with sequence information. The results illustrated in Figure 11(a) showed performance drops across all five datasets, highlighting the critical role of this module, particularly for regression tasks that demand precise and fine-grained modeling.

**Graph Learning Strategy.** This strategy enhances the model's ability to dynamically learn graph representations across layers by updating the Unified node features after the symmetric Cross-Attention fusion at each layer, rather than relying on static processing (fixed encoder). The updated node features are further refined in subsequent layers, allowing the model to better capture intricate and continuous relationships within molecular graphs. As shown in Figure 11(b), this strategy leads to performance gains across all datasets, particularly in regression tasks, by leveraging the progressively enriched node features for more precise and detailed molecular representation.

**Bi-Attention.** This method extends standard self-attention by introducing a bidirectional mechanism, where sequence features are processed with normal self-attention and then passed through a reversed sequence self-attention layer, followed by fusion of the two outputs. While this approach adds cross-directional context, our experiments show that it slightly improves classification tasks but negatively impacts regression tasks, which is illustrated in Table 22. This suggests that the added bidirectional processing may introduce noise or unnecessary complexity, particularly for tasks requiring precise global property predictions. Consequently, this method was not adopted in our model.

**Graph Processing Module (MPNN vs. GCN).** In the Unified module, we evaluated the choice of Message Passing Neural Networks (MPNN) versus Graph Convolutional Networks (GCN) for message passing. MPNN consistently demonstrated superior performance across all tasks due to its ability to effectively capture intricate node relationships and propagate richer information through the graph. In contrast, GCN introduced significant performance degradations, which are shown in Table 22, particularly in regression tasks, where precise modeling of molecular structures is critical. This highlights the importance of using MPNN for robust and detailed feature extraction in the Unified module.

**Substructure-Level Tokenizer.** The substructure-level tokenizer for the SMILES sequence outperforms the character-level tokenizer, which struggles with both classification and regression tasks, as Table 22 shows. Unlike the character-level approach, which splits SMILES into individual characters and loses critical structural context, the substructure-level tokenizer treats special ions and atoms as individual tokens, preserving their unique chemical and structural properties. This results in significantly better performance by capturing meaningful substructure information essential for downstream tasks.

Table 23: Impact of the ratio of hierarchical fusion layers. The metrics for BACE, BBBP, and Clintox are AUROC (higher is better), while ESOL and LIPO use RMSE (lower is better).

| | CLASSIFICATION | | | REGRESSION | |
|---|---|---|---|---|---|
| LAYER NUM FUSION: SEQUENCE | BACE | BBBP | CLINTOX | ESOL | LIPO |
| 0.75 | $0.818_{(0.032)}$ | $0.954_{(0.006)}$ | $0.992_{(0.010)}$ | $0.587_{(0.017)}$ | $0.588_{(0.038)}$ |
| 0.25 | $0.841_{(0.032)}$ | $0.961_{(0.008)}$ | $0.994_{(0.012)}$ | $0.617_{(0.016)}$ | $0.602_{(0.022)}$ |
| **0.5** (MuMo) | $\mathbf{0.849}_{(0.014)}$ | $\mathbf{0.957}_{(0.011)}$ | $\mathbf{0.985}_{(0.011)}$ | $\mathbf{0.536}_{(0.061)}$ | $\mathbf{0.577}_{(0.027)}$ |

Table 24: Overlapping evaluation on MoleculeNet datasets (classification). "Overlap Rate" denotes the proportion of molecules overlapping with the pretraining corpus. Results are AUROC (↑) with standard errors in gray subscript.

| MODEL: MuMo | BACE ↑ | BBBP ↑ | CLINTOX ↑ | SIDER ↑ | TOX21 ↑ |
|---|---|---|---|---|---|
| OVERLAP RATE | 0.00% | 4.61% | 1.42% | 1.60% | 67.54% |
| ORIGINAL PERFORMANCE | $0.849_{0.014}$ | $0.957_{0.011}$ | $0.985_{0.011}$ | $0.677_{0.009}$ | $0.834_{0.009}$ |
| CLEAN PERFORMANCE | $0.849_{0.014}$ | $0.960_{0.009}$ | $0.996_{0.011}$ | $0.675_{0.011}$ | $0.839_{0.009}$ |
| PERFORMANCE CHANGE | 0.00% | **+0.31%** | **+1.12%** | -0.30% | **+0.60%** |

Table 25: Overlapping evaluation on MoleculeNet datasets (regression). "Overlap Rate" denotes the proportion of molecules overlapping with the pretraining corpus. Results are RMSE (↓) with standard errors in gray subscript.

| MODEL: MuMo | ESOL ↓ | FREESOLV ↓ | LIPO ↓ | QM7 ↓ | QM8 ↓ | QM9 ↓ |
|---|---|---|---|---|---|---|
| OVERLAP RATE | 37.67% | 29.28% | 6.90% | 4.61% | 2.09% | 0.58% |
| ORIGINAL PERFORMANCE | $0.536_{0.061}$ | $1.082_{0.088}$ | $0.448_{0.007}$ | $42.80_{0.6}$ | $0.0111_{0.0001}$ | $0.0030_{0.00001}$ |
| CLEAN PERFORMANCE | $0.342_{0.031}$ | $0.796_{0.056}$ | $0.439_{0.006}$ | $45.67_{0.5}$ | $0.0112_{0.0001}$ | $0.0030_{0.00001}$ |
| PERFORMANCE CHANGE | **+36.2%** | **+26.4%** | **+2.01%** | **+6.70%** | **+0.90%** | 0.00% |

Table 26: Ablation on Backbone Generalizability. Results on BBB (AUROC ↑), PGP (AUROC ↑), Caco2 (RMSE ↓), and PPBR (RMSE ↓). Standard errors are shown in gray subscript.

| MODEL | BBB ↑ | PGP ↑ | CACO2 ↓ | PPBR ↓ | AVG. DROP |
|---|---|---|---|---|---|
| MuMo (ATTN_MAMBA, OURS) | $0.899_{0.014}$ | $0.942_{0.019}$ | $0.315_{0.055}$ | $7.324_{0.323}$ | 0.00% |
| MuMo (TRANSFORMER) | $0.867_{0.051}$ | $0.899_{0.049}$ | $0.317_{0.095}$ | $8.489_{0.591}$ | 6.66% |
| MuMo (VANILLA MAMBA) | $0.869_{0.011}$ | $0.908_{0.020}$ | $0.396_{0.045}$ | $7.465_{0.333}$ | 8.65% |
| MAMBA-ONLY (W/O FUSION) | $0.813_{0.019}$ | $0.876_{0.023}$ | $0.832_{0.075}$ | $11.54_{0.788}$ | 59.55% |
| MuMo (TRANSFORMER, W/O FUSION) | $0.843_{0.021}$ | $0.889_{0.013}$ | $0.644_{0.077}$ | $9.892_{0.701}$ | 37.84% |

## D.7 Impact of Number of Fusion Layers

This experiment evaluates the impact of the number of fusion layers in the model on its performance across various molecular property prediction tasks. The results from Table 23 indicate that the optimal number of fusion layers is crucial for achieving the best performance, as neither too few nor too many layers consistently yield superior results. The model demonstrates a clear peak in performance with a balanced fusion layer configuration, highlighting the importance of effective feature integration for multimodal molecular representations.

## D.8 Backbone Generalizability

**Ablation on the backbone.** As shown in Table 26, MuMo with Transformer and Vanilla Mamba backbones both achieve competitive results, confirming that our fusion design is indeed model-agnostic.

**Ablation on fusion.** In contrast, when the fusion module is removed, both Mamba-only and Transformer-only baselines suffer large performance drops, which are up to 59.55% and 37.84%,

Table 27: Ablation on asymmetrical vs. symmetrical fusion approaches. Results are reported on BBB (AUROC ↑), PGP (AUROC ↑), Caco2 (RMSE ↓), and PPBR (RMSE ↓). Standard deviations are shown in gray subscript. The best result is highlighted.

| MODEL | TYPE | BBB ↑ | PGP ↑ | CACO2 ↓ | PPBR ↓ | AVG. DROP |
|---|---|---|---|---|---|---|
| **MuMo** | ASYMMETRIC | $0.899_{0.014}$ | $0.942_{0.019}$ | $0.315_{0.055}$ | $7.324_{0.323}$ | 0.00% |
| MuMo-FG | ASYMMETRIC | $0.878_{0.014}$ | $0.901_{0.011}$ | $0.412_{0.051}$ | $7.935_{0.297}$ | 11.96% |
| MuMo-CE | SYMMETRIC | $0.889_{0.019}$ | $0.917_{0.016}$ | $0.475_{0.059}$ | $8.273_{0.312}$ | 16.88% |
| MuMo-CB | SYMMETRIC | $0.849_{0.011}$ | $0.891_{0.023}$ | $0.601_{0.066}$ | $9.132_{0.359}$ | 31.60% |

Table 28: Evaluation on ESOL and BBBP datasets using different conformer optimization tools. Results are reported with standard errors in gray subscript. Std. Dev denotes the standard deviation across conformer settings, and Max Δ is the largest difference observed.

| TASK | MODEL | MMFF94 | UFF | NO-OPTIMIZE | STD. DEV ↓ | MAX Δ ↓ |
|---|---|---|---|---|---|---|
| ESOL (↓) | UNIMOL | $0.769_{0.153}$ | $0.790_{0.113}$ | $0.939_{0.191}$ | 0.0927 | 0.170 |
| ESOL (↓) | **MuMo** | $0.536_{0.061}$ | $0.550_{0.072}$ | $0.585_{0.065}$ | 0.0252 | 0.049 |
| BBBP (↑) | UNIMOL | $0.889_{0.025}$ | $0.831_{0.021}$ | $0.733_{0.006}$ | 0.0789 | 0.156 |
| BBBP (↑) | **MuMo** | $0.962_{0.007}$ | $0.952_{0.009}$ | $0.941_{0.009}$ | 0.0105 | 0.021 |

Table 29: Robustness to 3D conformer noise. Performance under increasing levels of Gaussian noise (in Å) added to atom coordinates. Results are reported with standard deviation in gray subscript.

| MODEL - DATASET | 0Å | 0.05Å | 0.1Å | 0.2Å | STD. DEV ↓ | MAX Δ ↓ |
|---|---|---|---|---|---|---|
| UNIMOL - ESOL (↓) | $0.769_{0.153}$ | $0.765_{0.144}$ | $0.760_{0.141}$ | $0.831_{0.220}$ | 0.0334 | 0.071 |
| **MuMo - ESOL (↓)** | $0.536_{0.061}$ | $0.530_{0.069}$ | $0.540_{0.040}$ | $0.544_{0.071}$ | 0.006 | 0.014 |
| UNIMOL - BBBP (↑) | $0.889_{0.025}$ | $0.878_{0.032}$ | $0.890_{0.031}$ | $0.787_{0.039}$ | 0.0496 | 0.103 |
| **MuMo - BBBP (↑)** | $0.962_{0.007}$ | $0.960_{0.007}$ | $0.961_{0.008}$ | $0.954_{0.010}$ | 0.0036 | 0.008 |

respectively (Table 26). This highlights that the key performance gains come from our structured fusion strategy, not from the choice of backbone alone.

Overall, while our hybrid Attn-Mamba design yields the strongest results, our ablations demonstrate that the proposed fusion scheme can generalize across architectures, validating its standalone effectiveness.

### D.9 Impact of Asymmetric Fusion

To investigate the role of fusion strategy, we compare our asymmetric injection-enhanced design against symmetric concatenation-based alternatives. As shown in Table 27, the full MuMo with asymmetric fusion consistently achieves the best performance across all benchmarks. By contrast, symmetric strategies that concatenate modalities either at the beginning (MuMo-CB) or the end (MuMo-CE) lead to large performance drops, especially on regression tasks such as Caco2 and PPBR. The fixed-graph variant (MuMo-FG) also underperforms due to its inability to jointly adapt representations. These results confirm that asymmetric injection not only preserves modality-specific information but also enables more effective cross-modal enrichment, leading to superior generalizability.

### D.10 Potential Data Overlap Analysis

**Experiment Results.** We analyzed all MoleculeNet datasets for potential overlap with our ChEMBL-1.6M pretraining corpus, and re-ran experiments after removing overlapping molecules. Surprisingly, the average performance increased by **6.81%** across datasets rather than dropping (Tables 24 and 25).

**Explanation.** This confirms that our pretraining task (masked language modeling) does not leak label information even under worst-case overlap scenarios (e.g., Tox21, where the overlap reached

67.5%). Since no downstream supervision is involved during pretraining, the model cannot memorize labels from ChEMBL. Moreover, given the scale of ChEMBL and our relatively short training schedule (2 epochs), it is unlikely that the model memorized specific molecular structures.

**Observation.** Notably, the largest gains appeared in datasets with the highest overlap, such as ESOL (+36.2%) and Freesolv (+26.4%). We hypothesize that overlapping molecules may have been harder examples, and their removal led to cleaner evaluation splits. This highlights both the robustness of MuMo and the soundness of our pretraining corpus design.

### D.11 Ablation on Conformer Settings and Robustness to Geometric Perturbations

**Effect of Conformer Generation Tools.** We evaluated MuMo and UniMol under three common conformer settings: (1) MMFF94-minimized, (2) UFF-minimized, and (3) ETKDG without optimization. As shown in Table 28, MuMo maintains stable performance across all conformer sources and exhibits notably smaller fluctuations compared to UniMol. For example, on ESOL, MuMo's standard deviation across conformer sources is only 0.0252, much lower than UniMol's 0.0927. This indicates that our injection-enhanced design provides robustness against geometric inconsistencies, ensuring reliable predictions even when different conformer generation pipelines are used.

**Impact of Conformer Noise.** We further test sensitivity to conformer noise by perturbing atom positions with Gaussian noise of varying magnitudes. As summarized in Table 29, UniMol suffers significant degradation as noise increases (e.g., ESOL from 0.769 to 0.831 RMSE), while MuMo's performance remains nearly unchanged (Std. Dev = 0.006, Max $\Delta$ = 0.014). Similar trends are observed on BBBP. These results confirm that MuMo is more robust to 3D geometry noise, making it better suited for real-world scenarios where generated conformers may be approximate.

