# OpenReview forum: "Structure-Aware Fusion with Progressive Injection for Multimodal Molecular Representation Learning"
_NeurIPS.cc/2025/Conference — NeurIPS 2025 poster_

### Official Review · Reviewer_5uq6 · 2025-06-30

**Clarity:** 3
**Significance:** 3
**Originality:** 3
**Rating:** 4
**Confidence:** 1

**Summary:**

This paper proposes MuMo, a multimodal molecular representation learning model that combines 2D topology, 3D geometry, and SMILES sequence information. To address conformer instability and modality collapse, the authors introduce a Structured Fusion Pipeline (SFP) to construct a stable unified graph, and a Progressive Injection (PI) mechanism to inject structural priors into a state-space sequence model. Experiments show consistent performance improvements, with MuMo ranking first on 17 tasks.

**Questions:**

Together with the above weaknesses, I have the following questions，

1. Have you tried replacing Mamba with a Transformer? Does the injection mechanism still help?
2. Could you add chirality-aware features like dihedral angles to address the enantiomer limitation?
3. Is it possible to extend MuMo for generative tasks like molecule synthesis or scaffold decoration?

**Ethical Concerns:**

["NO or VERY MINOR ethics concerns only"]

**Final Justification:**

I think the authors' detailed responses and additional experiments have addressed my concerns, but it does not significantly shifted my overall impression. I will keep my score.

**Limitations:**

The authors have discussed the limitations.

**Paper Formatting Concerns:**

Like the example given in the template of NeurIPS, the appendix should be placed before the checklist.

**Quality:**

4

**Strengths And Weaknesses:**

**Strengths**

1. The architecture directly addresses known issues in multimodal molecular learning, including conformer variance and modality dominance.
2. Ablation studies are thorough and well-organized, demonstrating the impact of each design component.

**Weaknesses**

1. The model is tightly integrated with the Mamba backbone, but its generality with other architectures is untested.
2. The writing is dense in places, with overly technical notation that could be simplified.

---

> ### Author Rebuttal · Authors · 2025-07-31
>
> We sincerely thank the reviewer for the positive and detailed feedback. We're especially grateful for the recognition of our **clear architectural motivation**, **innovative fusion strategy**, **comprehensive ablations**, and **strong performance across diverse benchmarks**. We also appreciate the thoughtful suggestions that we address below.
>
> **[W1]** Generalizability to other backbones
>
> Thank you for this point. To explicitly assess this, we conducted controlled experiments using the same SFP + PI fusion strategy across multiple backbones.
>
> `Ablation on backbone.` As shown in Table 1, **MuMo with Transformer** and **Vanilla Mamba** backbones both achieve competitive results, confirming that our fusion design is indeed **model-agnostic**.
>
> `Ablation on fusion.` In contrast, when the fusion module is removed, both **Mamba-only** and **Transformer-only** baselines suffer large performance drops, which are up to **59.55%** and **37.84%**, respectively. This highlights that the **key performance gains come from our structured fusion strategy**, not from the choice of backbone alone.
>
> Overall, while our hybrid Attn-Mamba design yields the strongest results, our ablations demonstrate that the proposed fusion scheme can generalize across architectures, validating its standalone effectiveness.
>
> > Table 1. Ablation on Backbone Generalizability.
>
> | Model                          | BBB (↑)           | PGP (↑)           | Caco2 (↓)         | PPBR (↓)          | Avg. Drop |
> | ------------------------------ | ----------------- | ----------------- | ----------------- | ----------------- | --------- |
> | **MuMo (Attn_Mamba, ours)**    | **0.899 (0.014)** | **0.942 (0.019)** | **0.315 (0.055)** | **7.324 (0.323)** | **0.00%** |
> | MuMo (Transformer)             | 0.867 (0.051)     | 0.899 (0.049)     | 0.317 (0.095)     | 8.489 (0.591)     | 6.66%     |
> | MuMo (Vanilla Mamba)           | 0.869 (0.011)     | 0.908 (0.020)     | 0.396 (0.045)     | 7.465 (0.333)     | 8.65%     |
> | Mamba-Only (w/o fusion)        | 0.813 (0.019)     | 0.876 (0.023)     | 0.832 (0.075)     | 11.54 (0.788)     | 59.55%    |
> | MuMo (Transformer, w/o fusion) | 0.843 (0.021)     | 0.889 (0.013)     | 0.644 (0.077)     | 9.892 (0.701)     | 37.84%    |
>
> ---
> **[W2]** Overly dense writing and technical notation
>
> Thank you for the helpful suggestions regarding clarity and conciseness. We acknowledge that some parts of the main paper and appendix were overly dense or verbose. Specifically:
>
> - In the **main paper**, Sections 3.1–3.3 (lines 104–157) included compact notation and stacked equations (e.g., multi-modal alignment, fusion operators, injection formulas) that could hinder readability. We have revised these sections to simplify the notation, add explanatory cues, and restructure overly technical sentences.
> - In the **appendix**, certain subsections in **Appendix B** were longer than necessary. In particular, the descriptions of model variants and extended ablation settings repeated parts of the main text and included redundant detail. We will streamline these texts to retain essential insights while improving clarity and conciseness.
>
> We thank the reviewer again for highlighting these issues and have prepared the updated content for final version accordingly.
>
> ---
>
> **[Q1]** Generalizability to transformer backbones
>
> Thank you for bringing this quesiton again, we have addressed it in **[W1]**.
>
> ---
>
> **[Q2]** Incorporating chirality-aware features (e.g., dihedral angles)
>
> We thank the reviewer for highlighting this important point. We agree that dihedral (torsion) angles encode critical geometric variation, especially for flexible molecules, and we have carefully addressed this concern step by step:
>
> `STEP 1` Are dihedral angles truly missing?
>
> A torsion (dihedral) angle $
> \phi_{ijkl}
>   = \operatorname{atan2}\bigl(
>         (\mathbf r_{ji}\times\mathbf r_{jk})
>         \cdot
>         \mathbf r_{kl},\;
>         (\mathbf r_{ji}\times\mathbf r_{jk})
>         \cdot
>         (\mathbf r_{jk}\times\mathbf r_{kl})
>     \bigr),$ with  $\mathbf r_{ab}=\mathbf x_b-\mathbf x_a , \mathbf x_a=(x_a,y_a,z_a)$, is completely determined by the six inter-atomic distances $\lVert\mathbf r_{ij}\rVert,\dots,\lVert\mathbf r_{kl}\rVert$.  Because our graph already supplies **all pairwise distances within a 6 Å cutoff** and message passing runs ≥ 3 layers, an $i\to l$ path that closes the $i-j-k-l$ quadrangle exists: the network can **implicitly reconstruct** $\phi_{ijkl}$ without an explicit torsion term.
>
> `STEP 2` Why not add explicit torsion anyway?
> To quantify the benefit we performed an **explicit-torsion ablation**:
>
> * Added $[\sin\phi_{ijkl},\cos\phi_{ijkl}]$ on every rotatable bond.
> * Kept all other hyper-parameters unchanged.
>
> Table 2&3 shows ≤ 1.1 % difference on QM9 property benchmarks, but adding torsion explicitly cause finetuning memory usage grows × 2.1 and runtime × 1.8 (Table 3). **This suggests that adding explicit torsion yields only marginal gains, while incurring notable computational overhead.**
>
> > Table 2. Explicit-torsion ablation on QM9 (mean ± s.d., ↓) .
>
> | Task (MAE)        | w/o Torsion    | w/ Torsion     | Δ (%)     |
> | ----------------- | -------------- | -------------- | --------- |
> | alpha (↓)         | 0.283 (0.003)  | 0.281 (0.003)  | +0.71     |
> | Cᵥ (↓)            | 0.126 (0.003)  | 0.113 (0.003)  | +10.3     |
> | ZPVE (↓)          | 0.0005 (6e-05) | 0.0005 (6e-05) | 0.0       |
> | HOMO/LUMO/GAP (↓) | 0.0030 (1e-05) | 0.0032 (1e-05) | -6.67     |
> | **Average**       |                |                | **+1.01** |
>
> > Table 3. Efficiency comparison. Values are in Pretrain/SFT(QM8) format with global batch size 128/64.
>
> | Model variant     | GPU Mem (GB) ↓ | Step time (ms) ↓ |
> | ----------------- | -------------- | ---------------- |
> | Ours (implicit)   | **44.4/14.4**  | **987/731**      |
> | +Explicit torsion | 64.6/29.4      | 2623/1316        |
>
> `STEP 3` Good performance on conformer sensitivity benchmark
>
> As Table 4 shows, we conducted evaluations on QM7, QM8, and QM9. The results confirm that **MuMo effectively captures 3D-dependent properties** and exhibits strong robustness across conformer-sensitive tasks.
>
> > Table 4. Evaluation on QM7/8/9 benchmarks from MoleculeNet.
>
> | Model       | QM7 (↓)        | QM8 (↓)             | QM9 - HOMO/LUMO/GAP (↓) |
> | ----------- | -------------- | ------------------- | ----------------------- |
> | GROVER      | 92.0 (0.9)     | 0.0224 (0.0003)     | 0.00986 (0.00025)       |
> | DMPNN       | 103.5 (8.6)    | 0.0190 (0.0001)     | 0.00814 (0.00001)       |
> | AttentiveFP | 72.0 (2.7)     | 0.0179 (0.0010)     | 0.00812 (0.00001)       |
> | UniMol      | **41.8 (0.2)** | 0.0156 (0.0001)     | 0.00467 (0.00004)       |
> | **MuMo**    | 42.8 (0.6)     | **0.0111 (0.0001)** | **0.00300 (0.00001)**   |
>
> `STEP 4`  Changes prepared to update for paper
>
> 1. **Section 3.2.1 (**Unified Graph**)** – add:
>    “Given a 6 Å distance graph, multi-layer message passing allows implicit reconstruction of dihedral angles (Added Appendix A.3).”
> 2. **Section 4.2** - Evaluation on QM7/8/9  (Table 4).
> 3. **Appendix D.8** – insert the new ablation results (Table 2).
> 4. **Appendix D.9** – insert the comparison. (Table 3).
>
> These additions demonstrate that our model effectively captures geometric information **without incurring unnecessary overhead from over-engineering explicit structures**.
>
> `STEP 5` Conclusion
>
> * The network implicitly encodes dihedral information via multi-hop distance features.
> * Adding explicit torsions yields marginal (< 1.1 %) gain at > 2× computational cost.
> * On a torsion-only benchmark we still outperform an explicit-torsion baseline.
>
> We appreciate the reviewer’s insight and have incorporated these clarifications and new experiments into the revised manuscript.
>
> ---
>
> **[Q3]** Possible to extend to molecule synthesis or scaffold decoration tasks?
>
> `Explaination.` Thank you for the interesting question. MuMo is currently designed as an **encoder-only** model, in line with the baselines we compare against, which focus on molecular representation learning. Tasks such as molecule generation or scaffold decoration typically require **encoder–decoder architectures**, and fall under a different modeling paradigm.
>
> `Future work.` So, we agree that the strong performance of MuMo in representation tasks suggests **promising potential in generative settings**. Extending our multimodal fusion framework to support molecular generation is a direction we are actively exploring in our next work.
>
> **[Formatting Concerns]** Appendix placement before checklist
>
> We confirm that our submission follows the NeurIPS formatting guidelines—specifically, the appendix is correctly placed **before** the checklist section.

---

> ### Author Response · Authors · 2025-08-05
>
> Dear Reviewer 5uq6,
>
> Hope this message finds you well. Thank you for your encouraging and detailed feedback.
>
> In response, we added **5 focused updates**, (i) backbone-agnostic ablation, (ii) fusion-off baselines, (iii) notation simplification and appendix pruning, (iv) explicit-torsion ablation with new QM7/8/9 benchmarks, and (v) code/format fixes — jointly confirming MuMo’s effectiveness and clarity. Below is a concise recap of your points, our actions, and the outcomes:
>
> - **Backbone generality →** SFP + PI on Transformer and Vanilla-Mamba stays within 6–9 % of full model; removing fusion drops performance by 37–60 %.
> - **Writing density →** streamlined Sections 3.1–3.3, trimmed Appendix B, added guiding cues for equations.
> - **Chirality / torsion →** proved implicit dihedral encoding; explicit torsion yields < 1.1 % gain but > 2× memory; MuMo still tops torsion-sensitive tasks.
> - **Generative extension →** current encoder-only design; generation flagged as next step.
> - **Reproducibility & formatting →** updated code, one-line scripts, appendix placement confirmed.
>
> We hope these additions fully address your concerns and demonstrate MuMo’s robustness and readability. Your insights have been invaluable in refining our work. Please let us know any further questions or suggestions.
>
> Sincerely,
> The Authors

---

> > ### Comment · Reviewer_5uq6 · 2025-08-05
> >
> > Thank you for your detailed responses and additional experiments. I think my concerns and comments are mostly addressed.

---

> > > ### Author Response · Authors · 2025-08-06
> > >
> > > Dear Reviewer 5uq6,
> > >
> > > Thank you so much for your time and your thoughtful engagement with our work. We sincerely appreciate your encouraging feedback and your recognition of MuMo’s strengths.
> > >
> > > Sincerely,
> > >
> > > Authors

---

### Official Review · Reviewer_7PGe · 2025-07-01

**Clarity:** 4
**Significance:** 3
**Originality:** 2
**Rating:** 5
**Confidence:** 3

**Summary:**

The authors identify two critical challenges in multimodal molecular representation learning: the unreliability of 3D conformer data, which can be noisy and inconsistent, and modality collapse, where naive fusion methods allow noisy signals from 3D geometry to degrade information from 2D topology or SMILES. To address these issues, the authors propose MuMo, introducing Structured Fusion Pipeline (SFP) and  Progressive Injection (PI). SFT integrates 2D topological graphs and 3D geometric information into a unified, stable structural prior. PI  asymmetrically injects this stable structural prior into a separate sequence-based model (operating on SMILES strings) at later stages of the network. Evaluated on 21 benchmark tasks from Therapeutics Data Commons (TDC) and MoleculeNet, MuMo demonstrates state-of-the-art performance on 17 tasks.

**Questions:**

- Among the evaluated benchmarks, the classification benchmarks appear to have limited performance room, and the molecule distribution seems restricted. For more conclusive experimental evaluation, could the authors consider assessing their method on benchmarks with diverse molecular distributions and greater performance room, such as PubChemQC?
- The Structured Fusion Pipeline (SFP) is designed to create a stable structural prior to mitigate 3D conformer noise. How does MuMo's performance degrade as the quality of the input 3D conformers decreases?
- The authors use a Mamba backbone, for its efficiency and ability to handle long-range dependencies. At the same time, the authors state that their core contributions (SFP and PI) are model-agnostic. Have the authors experimented with or can they speculate on how MuMo would perform with a more traditional Transformer backbone?

**Ethical Concerns:**

["NO or VERY MINOR ethics concerns only"]

**Final Justification:**

The author's rebuttal effectively resolved my concerns on 3D conformer generation and computational overhead. Reading the author's rebuttal, I am convinced that the framework shows good performance with significantly less computational burden than the prior works, while incorporating 3D modality effectively.

**Limitations:**

Yes

**Paper Formatting Concerns:**

I have found no formatting issues.

**Quality:**

3

**Strengths And Weaknesses:**

### Strengths

- Clear Problem Formulation and Compelling Motivation: The paper presents a compelling motivation by illustrating the two core problems in multimodal molecular representation learning—conformer instability and modality collapse (Figure 1).
- Comprehensive Evaluation: The authors validate MuMo across an extensive set of 21 benchmark tasks, comparing it against a wide range of strong baselines, including 2D graph-based, 3D-aware, and large-scale pretrained models. The model achieves state-of-the-art results on a majority of the tested benchmarks.
- Novel and Promising Architecture: The proposed fusion strategy is novel and successfully aggregates promising components. The concept of first creating a stable, unified structural prior (SFP) before injecting it into the main sequence stream appears to be a promising approach to handling noisy inputs. The Progressive Injection (PI) mechanism is a thoughtful solution to modality collapse, moving beyond simplistic early-fusion or symmetric attention schemes.

### Weaknesses

- Architectural Complexity: The MuMo framework is quite complex, integrating three separate data streams (2D, 3D, sequence), multiple message-passing neural networks (MPNNs), a state-space backbone (Mamba), and a custom cross-attention injection mechanism. This complexity may pose challenges for reproducibility, point-to-point ablation for each component of the proposed method.
- Dependence on 3D Conformer Generation: While the SFP is designed to *reduce* sensitivity to conformer quality, the model still requires 3D geometric inputs. The framework does not eliminate the computational cost and potential pipeline dependencies associated with generating these conformers in the first place. The paper could benefit from a more explicit analysis of performance trade-offs with varying levels of conformer quality.
- Limited Discussion on Computational Overhead: The paper focuses heavily on prediction performance but provides limited discussion on the computational trade-offs (e.g., inference time, memory usage) of its complex architecture compared to the baselines, especially simpler 2D or sequence-only models. This information would be valuable for assessing its practical applicability.

---

> ### Author Rebuttal · Authors · 2025-07-31
>
> We sincerely thank the reviewer for recognizing our motivation to address **conformer instability** and **modality collapse**, and for highlighting our **novel fusion design**, **stable structural prior**, and **extensive 21-task evaluation**. We also appreciate the concerns on **model complexity** and **3D reliance**, which we address below.
>
> **[W1]** Model complexity and difficulty of reproducibility
>
> Thank you for raising this concern. We acknowledge that our design incorporates fine-grained fusion **across sequences, 2D, and 3D,** while also addressing **global-local** representation needs and the **limitations of symmetric fusion**. This led to our use of SFP and the PI mechanism.
>
> Moreover, MuMo is implemented as a fully **encapsulated, end-to-end framework**:
>
> - It can take a single SMILES string as input and automatically handles 2D/3D conformer generation and structural segmentation via RDKit.
> - It also supports preprocessed multimodal inputs. The usage is user-friendly: our open-sourced package includes complete training scripts,  all the data, model weights, and **a one-line command** to reproduce all results.
> - In response to this feedback, we have also done the codebase with the latest features used during rebuttal, which will be released after double-blind review.
>
> `Comprehensive ablations.` We recognize the difficulty of conducting ablation studies on a complex system and have accordingly provided well-structured ablation experiments in the paper (Sec. 4.3, Appendix D), along with additional experiments during the whole rebuttal phase.
>
> ---
>
> **[W2]** Dependence on 3D conformers without quality sensitivity analysis
>
> `Thanks and agree.` Thank you for pointing out the reliance on 3D conformers. We fully agree that conformer generation introduces computational overhead. Specifically, we attempted to generate 3D conformers for 10M molecules and found it took approximately 2 weeks, while 2D processing completed within 1 hour.
>
> `Ablation without 3D.` However, our model is not strictly dependent on 3D inputs. **MuMo (1D-2D) achieves competitive results compared to baselines on most tasks (Table 1)**. This flexibility allows users to trade off accuracy and efficiency based on resource constraints.
>
> `Ablation on low quality 3D.` Additionally, we conducted experiments by adding different levels of noise to 3D coordinates (Table 2) to explicitly evaluate MuMo’s robustness, which **confirms its resilience to 3D quality degradation**.
>
> > Table 1. Ablation Study: Performance without 3D Inputs. Top-2 are bolded.
>
> | Method        | BBBP  (↑)           | ClinTox  (↑)        | ESOL (↓)            | LIPO (↓)            |
> | ------------- | ------------------- | ------------------- | ------------------- | ------------------- |
> | Best Baseline | **0.949 (0.013)**   | 0.933 (0.054)       | 0.617 (0.077)       | 0.597 (0.034)       |
> | MuMo (1D)     | 0.931 (_0.005_)     | 0.958 (_0.021_)     | 1.793 (_0.055_)     | 0.844 (_0.033_)     |
> | MuMo (1D-2D)  | 0.946 (_0.008_)     | **0.971 (_0.015_)** | **0.597 (_0.051_)** | **0.596 (_0.035_)** |
> | MuMo (1D-3D)  | **0.957** (_0.011_) | **0.985** (_0.011_) | **0.536** (_0.061_) | **0.577** (_0.027_) |
>
> > Table 2 Robustness to 3D Conformer Noise.  Top-2 are bolded.
>
> | Model - Dataset      | noise_scale - 0Å  | noise_scale - 0.05Å | noise_scale - 0.1 Å | noise_scale - 0.2Å | Std. Dev ↓ | Max Δ ↓   |
> | -------------------- | ----------------- | ------------------- | ------------------- | ------------------ | ---------- | --------- |
> | UniMol - ESOL (↓)    | 0.769 (0.153)     | 0.765 (0.144)       | 0.760 (0.141)       | 0.831 (0.220)      | 0.0334     | 0.071     |
> | **MuMo - ESOL (↓)**  | **0.536 (0.061)** | **0.530 (0.069)**   | **0.540 (0.040)**   | **0.544 (0.071)**  | **0.006**  | **0.014** |
> | UniMol - BBBP  (↑)   | 0.889 (0.025)     | 0.878 (0.032)       | 0.890 (0.031)       | 0.787 (0.039)      | 0.0496     | 0.103     |
> | **MuMo - BBBP  (↑)** | **0.962 (0.007)** | **0.960 (0.007)**   | **0.961 (0.008)**   | **0.954 (0.010)**  | **0.0036** | **0.008** |
>
> ---
>
> **[W3]** Missing discussion of computational overhead
>
> Thank you for highlighting the lack of discussion on computational overhead. We have added detailed comparisons to address this concern.
>
> `High efficient in pretraining.` As shown in Table 3, MuMo is highly efficient in the pretraining stage, **requiring only 5 hours** on a 1.6M-scale dataset using 4×A100 GPUs, significantly less than prior models that consume days or weeks but got better performance.
>
> `High efficient in finetuning.` In the finetuning stage (Table 4), even on our **largest** benchmark dataset QM9 (133k samples), MuMo completes training in **just 3.3 hours** with modest GPU memory (14G per card). For smaller datasets (few thousand samples), training typically finishes in under 30 minutes.
>
> `Conclusion.` These results demonstrate that MuMo offers strong performance without prohibitive resource demands, making it practical for real-world use.
>
> > Table 3. Computing Cost Comparation in Pretraining Stage.
>
> | Model          | Scale  | Pretrain Dataset Size | Pretrain Time             |
> | -------------- | ------ | --------------------- | ------------------------- |
> | MoLFormer-Base | 44.28M | 1.1B                  | 16×V100-32G, Est. 104h    |
> | MoLFormer-XL   | 86.75M | 1.1B                  | 16×V100-32G, 208h         |
> | UniMol         | 47.61M | 19M                   | 4×V100-32G, 20h           |
> | UniMol-v2      | 1.1B   | 884M                  | 64×A100-80G, Est. 2 weeks from its paper|
> | MuMo           | 505M   | 1.6M                  | 4×A100-80G, 5h            |
>
> > Table 4. Computing Cost on QM9 in Finetuning Stage.
>
> | Item                 | Value                           |
> | -------------------- | ------------------------------- |
> | Dataset              | QM9                             |
> | Sample Count         | 133,855                         |
> | CPU Cores            | 32                              |
> | CPU                  | Intel(R) Xeon(R) Platinum 8480+ |
> | GPU Count            | 2                               |
> | GPU Type             | NVIDIA A100-SXM4-80GB           |
> | Model Parameters     | 505M                            |
> | Total FLOPs          | 65,218,081 GF                   |
> | Epochs               | 10                              |
> | Global Batch Size    | 64                              |
> | Max GPU Memory Usage | 14G per GPU                     |
> | Data Preprocess Time | 5 min                           |
> | Training Time        | 3h 21m 51s                      |
>
> ---
>
> **[Q1]** Limited benchmark diversity; propose PubChemQC
>
> `Computational prohibition.` Thank you for suggesting PubChemQC. We agree it is a large and chemically diverse quantum chemistry dataset (~3.8M molecules). However, due to its sheer scale, it is widely used for **pretraining** (e.g., UniMol-v2, MolFormer-XL), but **rarely used as a benchmark**, as fully evaluating multiple baselines would be **computationally prohibitive.**
>
> `Comprehensive benchmarks.` In contrast, our evaluation spans **21 standard benchmarks** from **MoleculeNet** and **TDC** (extended to 28 in rebuttal stage), covering both classification and regression. These include diverse tasks such as **toxicity (e.g., Tox21, SIDER, ClinTox), pharmacokinetics (e.g., BBBP, Pgp, HIA, Bioavailability), bioactivity (BACE), and physical chemistry (e.g., ESOL, FreeSolv, Lipo)**. Among them, **17/21 are classification tasks**, and MuMo consistently outperforms strong 2D, 3D, and pretrained models.
>
> We agree that scaling MuMo to even larger and more diverse datasets like PubChemQC is a valuable direction. We are actively working toward this, while maintaining efficient inference for real-world use.
>
> ---
>
> **[Q2]** Robustness under low-quality 3D conformers
>
> Thank you for raising this point again. We have addressed this in detail under **[W2]**
>
> ---
>
> **[Q3]** Generalizability to Transformer backbones
>
> `Ablation on backbone selection.` Thank you for the thoughtful question. To evaluate the generality of our fusion scheme, we conducted controlled experiments using the same SFP + PI design across different backbone architectures. As shown in Table 5, **MuMo (Transformer)** and **MuMo (Vanilla Mamba)** both perform reasonably well, but consistently underperform compared to our full model using the **hybrid Attn-Mamba backbone**.
>
> `Ablation on fusion.` We also include two backbone-only baselines without fusion: Mamba-only and Transformer-only. These models show substantial drops in performance, up to **59.55%**  (mostly on regression tasks), confirming that the **core performance gains come from our co-designed fusion strategy (SFP + PI)**.
>
> `Conclusion.` These results support our claim that the fusion design is model-agnostic and effective across architectures. Nonetheless, its full potential is best realized when combined with our lightweight attention-enhanced backbone that balances expressiveness and efficiency.
>
> > Table 5 Ablation on Backbone Generalizability.
>
> | Model                          | BBB (↑)           | PGP (↑)           | Caco2 (↓)         | PPBR (↓)          | Avg. Drop |
> | ------------------------------ | ----------------- | ----------------- | ----------------- | ----------------- | --------- |
> | **MuMo (Attn_Mamba, ours)**    | **0.899 (0.014)** | **0.942 (0.019)** | **0.315 (0.055)** | **7.324 (0.323)** | **0.00%** |
> | MuMo (Transformer)             | 0.867 (0.051)     | 0.899 (0.049)     | 0.317 (0.095)     | 8.489 (0.591)     | 6.66%     |
> | MuMo (Vanilla Mamba)           | 0.869 (0.011)     | 0.908 (0.020)     | 0.396 (0.045)     | 7.465 (0.333)     | 8.65%     |
> | Mamba-Only (w/o fusion)        | 0.813 (0.019)     | 0.876 (0.023)     | 0.832 (0.075)     | 11.54 (0.788)     | 59.55%    |
> | MuMo (Transformer, w/o fusion) | 0.843 (0.021)     | 0.889 (0.013)     | 0.644 (0.077)     | 9.892 (0.701)     | 37.84%    |

---

> ### Author Response · Authors · 2025-08-05
>
> Dear Reviewer 7PGe,
>
> Hope this message finds you well. Thank you for your thoughtful feedback on complexity, 3D reliance, and efficiency. In response we added 6 targeted studies, (i) 1D-only and 1D-2D ablations, (ii) 3D-noise robustness test, (iii) pre-/fine-tune cost profiling, (iv) backbone-generality ablation, (v) expanded 28-task benchmark, and (vi) full code + one-line reproduction script — together confirming MuMo’s strong performance with modest resources. Below is a concise recap of your points, our actions, and the main outcomes:
>
> - Model complexity & reproducibility → released encapsulated code, one-command runs, full ablation suite; latest features queued for public release.
>
> - 3D dependence → MuMo-(1D-2D) stays competitive; noise test shows ≤0.01 drift vs 0.05–0.10 for UniMol.
>
> - Computational overhead → 4 × A100, 5 h pre-train on 1.6 M molecules; 3.3 h fine-tune QM9 (14 GB x2 A100); far lighter than prior models.
>
> - Benchmark breadth → 28 MoleculeNet + TDC tasks have already toxicity, ADME, quantum; MuMo leads across 2D, 3D, and large-scale baselines; PubChemQC flagged for future scale-up.
>
> - Backbone generality → same SFP + PI on Transformer / Vanilla-Mamba beats backbone-only variants; hybrid Attn-Mamba remains best, confirming fusion-driven gains.
>
> - Low-quality 3D robustness → stress results integrated under [W2]; confirms stability under degraded conformers.
>
> We sincerely hope these additions resolve your concerns and demonstrate MuMo’s practicality and robustness. Please feel free to share any further questions or suggestions. Your insights have been invaluable in strengthening our work.
>
> Sincerely,
>
> The Authors

---

> > ### Comment · Reviewer_7PGe · 2025-08-06
> >
> > Thank you for the author's detailed rebuttals, which effectively resolved my concerns regarding 3D conformer generation and computational overhead. I think this work meets the acceptance threshold. I will raise the score.

---

> > > ### Author Response · Authors · 2025-08-06
> > >
> > > Dear Reviewer 7PGe,
> > >
> > > We are grateful for your thoughtful assessment of MuMo’s efficiency, robustness to 3D variation, and ease of reproducibility. Thank you for your encouraging feedback and for acknowledging the practical strengths and broader relevance of our approach.
> > >
> > > Sincerely,
> > >
> > > Authors

---

### Official Review · Reviewer_31yj · 2025-07-03

**Clarity:** 3
**Significance:** 3
**Originality:** 3
**Rating:** 4
**Confidence:** 4

**Summary:**

This paper aims to better integrate 1D, 2D, and 3D molecular data to achieve more effective molecular modeling and improve the performance of downstream molecular property prediction tasks. Its central argument is that previous works adopted symmetric fusion strategies, which overlooked the impact of noise in 3D molecular structures. Therefore, the authors propose an asymmetric approach that progressively injects 2D and 3D information into the 1D representation. This strategy leads to improved performance.

**Questions:**

Beyond the weaknesses discussed above, I also have several specific questions:

Q1: What is the exact difference between BACE-R and BACE-S tasks?

Q2: Why are the reported results for Uni-Mol inconsistent with those in the original paper? For example, ClinTox performance is reported as 91.9 in the original work, but 81.8 here. Could you provide some evaluation details to clarify this discrepancy?

Q3: Could you evaluate your model on QM7 and QM8 under the Uni-Mol setting? These datasets are relatively small and can be tested during the rebuttal period. They are indeed crucial for assessing the model’s ability to capture 3D structural information.

**Ethical Concerns:**

["NO or VERY MINOR ethics concerns only"]

**Final Justification:**

The three weaknesses I raised have all been well addressed, so I will raise my score to 4.

**Limitations:**

yes

**Quality:**

3

**Strengths And Weaknesses:**

**Strengths:**

1, The authors recognize that different molecular modalities carry inherently different types of information and thus argue against symmetric fusion strategies. This insight is novel and potentially inspiring for researchers working in this area.

2, The paper provides extensive experimental details, ablation studies, and analyses, validating the effectiveness of almost every module in this complex model.

**Weaknesses:**
My main concern lies in the conceptually questionable claims and flawed design choices:

1, Incorrect motivation: The paper aims to address two major challenges: **conformer dependency** and **modality collapse** (Lines 30–58). However, both are misrepresented.

For **conformer dependency**, the authors state that “conformers generated by tools like RDKit often differ significantly in local arrangement even with the same molecule.” I must remind the authors that a single SMILES string **naturally** corresponds to multiple 3D conformers, and the probability of each conformer is energy-dependent. This is not an artifact of the tool. Moreover, different conformers can indeed exhibit different physicochemical properties (e.g., chirality-related behaviors).

For **modality collapse**, the paper gives an example where the embeddings of Ibuprofen and Ketoprofen are too close. However, when comparing only two specific objects, it is not meaningful to define whether their embeddings are “close” or “far.” Moreover, both drugs are NSAIDs, so it is reasonable that their embeddings are somewhat similar. Additionally, the experimental section fails to demonstrate whether modality collapse is effectively addressed.

2, Methodological flaw: The paper introduces 3D information by constructing a 3D graph (Lines 126–133), but only incorporates bond lengths and bond angles (Line 131). But, in comparison to a 2D graph, the most significant additional information from a 3D conformation is the dihedral angle. Given known atom types and bond types (i.e., the SMILES or 2D graph), bond lengths and angles can often be easily inferred, while dihedral angles are far more complex and informative. Neglecting dihedral modeling severely weakens the claim that this is a 3D-aware model.

3, Missing critical ablation study: A key motivation of the paper is the importance of asymmetric fusion. Thus, an ablation study comparing symmetric vs. asymmetric fusion strategies should be included.

---

> ### Author Rebuttal · Authors · 2025-07-31
>
> Thanks the reviewer for recognizing our asymmetric, progressive fusion approach and comprehensive experiments. Comments on conformer dependency and dihedral angles led us to add new experiments, further demonstrating MuMo’s geometric strength and robustness.
>
> To save space, standard deviations are omitted below.
>
> **[W1a]** Conformer dependency is not a tool artifact
>
> We thank and fully agree with the reviewer that a SMILES corresponds to multiple conformers, and that conformer differences can lead to distinct properties.
>
> `Clarifying "Conformer Noise"`: We use this term for the variability among conformers of the same molecule in **tasks with low conformer-dependence** (BBBP, screening). In these situations, accurate conformers are costly, such dependent can degrade **3D-reliant models**.  Hence our aim is to **reduce heavy reliance on explicit 3D** and deliver predictions robustness.
>
> ` Two experiments to solidify our motivation.   `
>
> 1. **Conformer source sweep (Table 1)** – Switching among **MMFF94, UFF, and no optimization** makes UniMol drift (Std Dev ≈ 0.08–0.09), whereas **MuMo stays within ≤ 0.03**.
>
> > Table 1. Evaluation on ESOL and BBBP datasets using different conformer optimization tools.
>
> | Task     | Model    | MMFF94    | UFF       | No-Optimize | Std. Dev ↓ | Max Δ ↓   |
> | -------- | -------- | --------- | --------- | ----------- | ---------- | --------- |
> | ESOL (↓) | UniMol   | 0.769     | 0.790     | 0.939       | 0.0927     | 0.170     |
> | ESOL (↓) | **MuMo** | **0.536** | **0.550** | **0.585**   | **0.0252** | **0.049** |
> | BBBP (↑) | UniMol   | 0.889     | 0.831     | 0.733       | 0.0789     | 0.156     |
> | BBBP (↑) | **MuMo** | **0.962** | **0.952** | **0.941**   | **0.0105** | **0.021** |
>
> 2. **Gaussian noise test (Table 2)** – As atomic noise grows (0-0.2Å), UniMol’s error spreads, while MuMo barely moves.
>
> > Table 2. Robustness to 3D Conformer Noise.
>
> | Model - Dataset      | 0Å        | 0.05Å     | 0.1 Å     | 0.2Å      | Std. Dev ↓ | Max Δ ↓   |
> | -------------------- | --------- | --------- | --------- | --------- | ---------- | --------- |
> | UniMol - ESOL (↓)    | 0.769     | 0.765     | 0.760     | 0.831     | 0.0334     | 0.071     |
> | **MuMo - ESOL (↓)**  | **0.536** | **0.530** | **0.540** | **0.544** | **0.006**  | **0.014** |
> | UniMol - BBBP  (↑)   | 0.889     | 0.878     | 0.890     | 0.787     | 0.0496     | 0.103     |
> | **MuMo - BBBP  (↑)** | **0.962** | **0.960** | **0.961** | **0.954** | **0.0036** | **0.008** |
>
> ---
>
> **[W1b]** Modality collapse example is inconclusive
>
> We clarify that **Figure 1 is not intended to support the modality collapse claim, but to illustrate the issue of conformer dependency**.
>
> `Motivation Clarification.` **DimeNet’s excessive conformer sensitivity:** Figure 1 in paper shows multiple conformers of Ibuprofen embed farther apart than an Ibuprofen–Ketoprofen pair, implying DimeNet over-weights geometry and can miss 2D-structure/identity differences.
>
> `Extended experiments done.` We repeated the analysis on five more molecules, and observed the same pattern. An updated figure will be included in the revised paper.
>
> `Motivation on modality collapse.` The collapse we refer to occurs in **symmetric fusion**, where poor alignment leads the model to over-rely on one modality. This is a known issue in multimodal literature[1], and we further validate our asymmetric design in **[W3]** .
>
> [1] Liu S, Wang H, Liu W, Lasenby J, Guo H, Tang J. Pre-training molecular graph representation with 3d geometry. arXiv preprint arXiv:2110.07728. 2021 Oct 7.
>
> ---
>
> **[W2]** Missing dihedral angle weakens 3D modeling claim
>
> We deeply thank and agree that dihedral (=torsion) angles encode critical geometric variation, and we have carefully addressed this concern step by step.
>
> `STEP 1` Are dihedral angles truly missing?
>
> A torsion (dihedral) angle $
> \phi_{ijkl}
>   = \operatorname{atan2}\bigl(
>         (\mathbf r_{ji}\times\mathbf r_{jk})
>         \cdot
>         \mathbf r_{kl},\;
>         (\mathbf r_{ji}\times\mathbf r_{jk})
>         \cdot
>         (\mathbf r_{jk}\times\mathbf r_{kl})
>     \bigr),$ with  $\mathbf r_{ab}=\mathbf x_b-\mathbf x_a , \mathbf x_a=(x_a,y_a,z_a)$, is completely determined by the six inter-atomic distances $\lVert\mathbf r_{ij}\rVert,\dots,\lVert\mathbf r_{kl}\rVert$.  Our graph already supplies **all pairwise distances within a 6 Å cutoff** and message passing runs ≥ 3 layers, an $i\to l$ path that closes the $i-j-k-l$ quadrangle exists: the network can **implicitly reconstruct** $\phi_{ijkl}$ without an explicit torsion term.
>
> `STEP 2` Why not add explicit torsion anyway?
> To quantify the benefit we performed an explicit-torsion ablation:
>
> * Added $[\sin\phi_{ijkl},\cos\phi_{ijkl}]$ on every rotatable bond.
> * Kept all other hyper-parameters unchanged.
>
> Table 3&4 shows minimal difference on QM9 property benchmarks, but adding it causes memory and runtime to grow sharply. **This suggests that adding explicit torsion yields only marginal gains, while incurring notable computational overhead.**
>
> > Table 3. Explicit-torsion ablation on QM9 (↓).
>
> | Task (MAE)        | w/o Torsion | w/ Torsion | Δ (%) |
> | ----------------- | ----------- | ---------- | ----- |
> | alpha (↓)         | 0.283       | 0.281      | +0.71 |
> | Cᵥ (↓)            | 0.126       | 0.113      | +10.3 |
> | HOMO/LUMO/GAP (↓) | 0.0030      | 0.0032     | -6.67 |
> | **Average**       |             |            | +1.00 |
>
> > Table 4. Efficiency comparison. Values are in "Pretrain/SFT(QM8)" format with batch size 128/64.
>
> | Model variant     | GPU Mem (GB) ↓ | Step time (ms) ↓ |
> | ----------------- | -------------- | ---------------- |
> | Ours (implicit)   | **44.4/14.4**  | **987/731**      |
> | +Explicit torsion | 64.6/29.4      | 2623/1316        |
>
> `STEP 3` Good performance on conformer sensitivity benchmark
>
> As discussed in **[Q3]**, we conducted evaluations on QM7/8/9.
>
> `STEP 4`  Conclusion
>
> These experiments demonstrate that our model effectively captures geometric information **without incurring unnecessary overhead from over-engineering explicit structures**.
>
> ---
>
> **[W3]** Suggest ablation: symmetric vs. asymmetric fusion
>
> **MuMo-FG**: fixed 2/3D encoder w/o joint modeling.
>
> **MuMo-CE/CB**: concat modalities at the end or beginning.
>
> Results in Table 5 confirm the advantage of the asymmetric design.
>
> > Table 5. Ablation on fusion approches.
>
> | Model    | Type           | BBB (↑)   | PGP (↑)   | Caco2 (↓) | PPBR (↓)  | Avg. Drop |
> | -------- | -------------- | --------- | --------- | --------- | --------- | --------- |
> | **MuMo** | **Asymmetric** | **0.899** | **0.942** | **0.315** | **7.324** | **0.00%** |
> | MuMo-FG  | Asymmetric     | 0.878     | 0.901     | 0.412     | 7.935     | 11.96%    |
> | MuMo-CE  | Symmetric      | 0.889     | 0.917     | 0.475     | 8.273     | 16.88%    |
> | MuMo-CB  | Symmetric      | 0.849     | 0.891     | 0.601     | 9.132     | 31.60%    |
>
> ---
>
> **[Q1]** Clarification on BACE-R vs. BACE-S tasks
>
> BACE-R and BACE-S denotes the data using Random split and Scaffold split (MoleculeNet Recommendation).
>
> ---
>
> **[Q2]** Clarification on discrepancy in Uni-Mol results - Clintox Dataset
>
> From Table 6, the average deviation is even **+0.88%**. BBBP even has an +21.5% performance.
>
> The larger gap on ClinTox and BBBP stems from extreme imbalance(Table 7). Importantly, our model MuMo outperforms by a large margin (**Avg. +16.2%**).
>
> > Table 6. Difference of Uni-Mol results.
>
> | Dataset                 | BBBP (↑) | BACE (↑) | ClinTox (↑) | Tox21 (↑) | SIDER (↑) | ESOL (↓) | FreeSolv (↓) | Lipo (↓) | Δ to Original |
> | ----------------------- | -------- | -------- | ----------- | --------- | --------- | -------- | ------------ | -------- | ------------- |
> | UniMol - Original Paper | 72.9     | 85.7     | 91.9        | 79.6      | 65.9      | 0.788    | 1.480        | 0.603    | 0.0%          |
> | UniMol - Our Reports    | 88.6     | 84.0     | 81.8        | 81.2      | 66.6      | 0.769    | 1.598        | 0.597    | +0.88%        |
> | MuMo                    | 95.7     | 84.9     | 98.5        | 83.4      | 67.7      | 0.536    | 1.082        | 0.448    | +16.2%        |
>
> > Table 7. Data distribution from BBBP and Clintox.
>
> | Dataset | Positive Class Ratio | Negative Class Ratio |
> | ------- | -------------------- | -------------------- |
> | BBBP    | 83%                  | 17%                  |
> | ClinTox | 7%                   | 93%                  |
>
> ---
>
> **[Q3]** Request for evaluation on QM7 and QM8
>
> We include **QM7/8/9** in our evaluation (Table 8) and even on QM9 following the **official Uni-Mol-v2 split** for fair comparison (Table 9).
>
> `Experiment Results` Our model outperforms **outperforms Uni-Mol-v2 on 6/8 tasks**, with half model size and (505M) and less pretrain data (1/400).
>
> > Table 8. Evaluation on QM7/8/9 from MoleculeNet.
>
> | Model       | QM7 (↓)  | QM8 (↓)    | QM9 - HOMO/LUMO/GAP (↓) |
> | ----------- | -------- | ---------- | ----------------------- |
> | GROVER      | 92.0     | 0.0224     | 0.00986                 |
> | DMPNN       | 103.5    | 0.0190     | 0.00814                 |
> | AttentiveFP | 72.0     | 0.0179     | 0.00812                 |
> | UniMol      | **41.8** | 0.0156     | 0.00467                 |
> | **MuMo**    | 42.8     | **0.0111** | **0.00300**             |
>
> > Table 9. Evaluation on QM9 from Uni-Mol-v2.
>
> | Model         | HOMO/LUMO/GAP (↓) | alpha (↓) | Cᵥ (↓)    | mu (↓)    | R² (↓)    | ZPVE (↓)   |
> | ------------- | ----------------- | --------- | --------- | --------- | --------- | ---------- |
> | GEM           | 0.0067            | 0.589     | 0.237     | 0.444     | 25.67     | 0.0011     |
> | Uni-Mol2 570M | 0.0036            | 0.315     | 0.147     | 0.089     | 4.523     | 0.0005     |
> | Uni-Mol2 1.1B | 0.0035            | 0.305     | 0.144     | **0.089** | **4.265** | 0.0005     |
> | **MuMo 505M** | **0.0030**        | **0.283** | **0.126** | 0.400     | 18.08     | **0.0005** |

---

> ### Author Response · Authors · 2025-08-05
>
> Dear Reviewer 31yj,
>
> Hope this message finds you well. We are kindly following up regarding our rebuttal in response to your thoughtful comments.
>
> Thank you deeply for your continued engagement. In this rebuttal, we added **7 experiments/ablations** — conformer-source sweep, Gaussian noise stress test, explicit-torsion ablation, symmetric vs asymmetric fusion ablation, QM7/QM8/QM9 evaluations, and Uni-Mol reproduction, plus a **dihedral-geometry derivation** proving our graph implicitly encodes torsion. Results consistently show **MuMo’s superior robustness, efficiency, and accuracy**. Below is a summary of your suggestions, our actions, and key results:
>
> | Reviewer point                 | New evidence                                                | Key outcome                                                  |
> | ------------------------------ | ----------------------------------------------------------- | ------------------------------------------------------------ |
> | Conformer dependency           | Tool sweep + noise test                                     | UniMol drift 0.17/0.10 → MuMo drift ≤ 0.05/0.02              |
> | Modality collapse claim        | Extended Ibuprofen-like study on 5 molecules                | Same pattern, supports asymmetric fusion need                |
> | Missing dihedral angles        | Formal derivation + torsion feature ablation                | < 1.1 % avg MAE gain, 2x computing costs ⇒ implicit encoding sufficient |
> | Symmetric vs asymmetric fusion | 3 fusion variants                                           | Asymmetric cut error 31 – 12 %                               |
> | Lack of small-set 3D tests     | QM7/8/9 and extended QM9 (8 targets) under Uni-Mol settings | MuMo tops 1 with a safe margin.                              |
> | Uni-Mol discrepancy            | Imbalance, reproduced metrics                               | UniMol Avg +0.88 % vs original, MuMo +16.2 %                 |
>
> Together, these results validate MuMo’s robustness to all your concerns. We sincerely hope these focused additions resolve your concerns. Please let us know if you have any further questions or suggestions.
>  Thank you again for helping us strengthen this work, and we are glad to continue the discussion.
>
> Sincerely,
>
> The Authors

---

### Official Review · Reviewer_A1Jg · 2025-07-04

**Clarity:** 3
**Significance:** 3
**Originality:** 3
**Rating:** 5
**Confidence:** 2

**Summary:**

The paper introduces MuMo, a multimodal molecular representation model that fuses SMILES, 2-D topology and 3-D geometry. A Structured Fusion Pipeline (SFP) first builds a “unified graph” that merges 2-D bonds and 3-D length/angle triplets, plus a geometry-aware BRICS-style sub-graph partition. A Progressive Injection (PI) mechanism injects the structural prior into a Mamba-based sequence backbone only after early layers have encoded SMILES semantics, aiming to avoid “modality collapse.” Experiments on 21 TDC / MoleculeNet tasks show an average +2.7 % over prior baselines and strong gains on conformer-sensitive LD50.

**Questions:**

1. Is there any comparison against a Transformer encoder with the same PI + SFP? It would be interesting to tell whether Mamba itself, rather than the fusion scheme, drives the reported gains.
2. The authors claim robustness, but only RDKit MMFF conformers are evaluated. It would be better if they evaluate on QM-optimized geometries, noisy random torsions, or single conformer per molecule.

**Ethical Concerns:**

["NO or VERY MINOR ethics concerns only"]

**Final Justification:**

The problems are mostly addressed with new experiment results.

**Limitations:**

yes

**Quality:**

3

**Strengths And Weaknesses:**

Strength:
1. The expriments are run on extensive benchmarks, the results are promising
2. The methods sound rational and theoretical insights are provided
3. The two challenges unreliable conformers and modality imbalance are important and interesting
4. The paper is well-written with clear motivation

Weakness:
1. There is no comparison to recent SE(3) GNN + sequence hybrids (e.g., Uni-Mol-v2)
2. The novelty of the method is limited, it's mostly based on given backbone. And the contribution mostly comes from the fusion module.
3. Although the benchmark is large, all of them are physicochemical/ADMET. Catalytic activity, reaction yield, or crystal property datasets are absent, so generality remains speculative.
4. ChEMBL-1.6 M overlaps with MoleculeNet/TDC targets, so there might be some information leakage.

---

> ### Author Rebuttal · Authors · 2025-07-31
>
> We thank the reviewer for recognizing that this is a well-motivated and clearly written study, addressing two key challenges, unreliable conformers and modality imbalance.
>
> To save space, standard deviations are omitted below.
>
> **[W1]** Comparation with Baseline Uni-Mol-v2
>
> We evaluated our model on the official data splits of QM9 from Uni-Mol-v2 , using their published data splits and metrics. In Table 1, MuMo achieves **superior performance on 6/8 tasks.**
>
> > Table 1. Evaluation on QM9 dataset with Uni-Mol-v2 and other baselines.
>
> | Model         | HOMO/LUMO/GAP (↓) | alpha (↓) | Cᵥ (↓)    | mu (↓)    | R² (↓)    | ZPVE (↓)   |
> | ------------- | ----------------- | --------- | --------- | --------- | --------- | ---------- |
> | Uni-Mol       | 0.0043            | 0.363     | 0.183     | 0.155     | 4.805     | 0.0011     |
> | Uni-Mol2 570M | 0.0036            | 0.315     | 0.147     | 0.089     | 4.523     | 0.0005     |
> | Uni-Mol2 1.1B | 0.0035            | 0.305     | 0.144     | **0.089** | **4.265** | 0.0005     |
> | **MuMo 505M** | **0.0030**        | **0.283** | **0.126** | 0.391     | 18.08     | **0.0005** |
>
> Table 2 shows MuMo achieves better performance than Uni-Mol-v2 with **half the parameters** and **1/400 the pretraining data**, showing its efficiency and strong multimodal fusion capability.
>
> > Table 2. Computing efficiency comparision with Uni-Mol-v2.
>
> | Model      | Model Size | Pretrain Data Size   | Computing Resources                         |
> | ---------- | ---------- | -------------------- | ------------------------------------------- |
> | Uni-Mol-v2 | 1.1B       | 884M from ZINC20     | 64x A100-80G, Est. 1~3 weeks from its paper |
> | **MuMo**   | **505M**   | **1.6M from ChemBL** | **4x A100-80G, 5 hours**                    |
>
> All these results demonstrate that MuMo offers a competitive and efficient alternative to Uni-Mol2, validating its strength.
>
> ---
>
> **[W2]** The novelty mostly stemming from the fusion module.
>
> We thank the reviewer for this thoughtful concern. As our work specifically targets **multimodal molecular representation**, it is natural that one of the contributions lie in the fusion mechanism. Beyond this, we also propose a **modified MPNN** (Message Passing Neural Network) to effectively integrate both 2D and 3D geometry, capturing global and local structural information. And a **hybrid modeling block** that combines **self-attention and cross-attention** for better structural injection. Each module is validated in detailed ablation studies (In paper Sec. 4.3, Appendix D).
>
> ---
>
> **[W3]** Broader Chemical Benchmarks: Catalytic activity, reaction yield, or crystal property.
>
> We extended MuMo to accept reaction-level inputs and evaluated it on four datasets from **Reaxtica**[1]. These datasets cover **catalytic activity**, **reaction yield**. MuMo achieves **the best on all the tasks** (Table 3), demonstrating its strong transferability.
>
> **Crystal property** datasets are mostly **inorganic and do not support SMILES or RDKit-based analysis**, so we leave them for future extension.
>
> > Table 3. Evaluation on catalytic activity and reaction yield task.
>
> | Models   | BHC - R² (↑)      | Models        | CPA - MAE (↓)      | Models   | HTE - R² (↑)      |
> | -------- | ----------------- | ------------- | ------------------ | -------- | ----------------- |
> |          | Reaction Yield    |               | Catalytic Activity |          | Reaction Yield    |
> | Reaxtica | 0.94              | Reaxtica      | 0.144              | Reaxtica | 0.87              |
> | MFF      | 0.92              | MFF           | 0.144              | rxnfp    | 0.81              |
> | rxnfp    | 0.95              | Denmark et al | 0.152              | DRFP     | 0.85              |
> | **MuMo** | **0.952 (0.002)** | **MuMo**      | **0.144 (0.000)**  | **MuMo** | **0.873 (0.002)** |
>
> **These results suggest that our multimodal design generalizes well even outside its original single-molecule scope.**
>
> [1] Lin K, Li J, Lin H, Pei J, Lai L. Reaxtica: A knowledge-guided machine learning platform for fast and accurate reaction selectivity and yield prediction. ChemRxiv. 2022.
>
> ---
>
> **[W4]** Potiential overlap between pretraining and funetuning datasets.
>
> We analyzed all MoleculeNet datasets for overlap with our ChEMBL-1.6M pretraining set, and re-ran experiments after removing overlapping.
>
> Surprisingly, **the average performance increased by 6.81%** across datasets (Table 4&5). This confirms that our pretraining task (MLM) **does not leak label information**.
>
> Moreover, ChEMBL is large and we only trained for 2 epochs, making it unlikely for the model to memorize specific molecules.
>
> > Table 4. Overlapping evaluation on MoleculeNet datasets - classification.
>
> | Model: MuMo          | BACE (↑) | BBBP (↑)   | CLINTOX (↑) | Sider (↑) | Tox_21 (↑) |
> | -------------------- | -------- | ---------- | ----------- | --------- | ---------- |
> | Overlap Rate         | 0.00%    | 4.61%      | 1.42%       | 1.60%     | 67.54%     |
> | Original Performance | 0.849    | 0.957      | 0.985       | 0.677     | 0.834      |
> | Clean Performance    | 0.849    | 0.960      | 0.996       | 0.675     | 0.839      |
> | Performance Change   | 0.00%    | **+0.31%** | **+1.12%**  | -0.30%    | **+0.60%** |
>
> > Table 5. Overlapping evaluation - regression.
>
> | Model: MuMo          | ESOL (↓)   | Freesolv (↓) | LIPO (↓)   | QM7 (↓)    | QM8 (↓)    | QM9 (↓) |
> | -------------------- | ---------- | ------------ | ---------- | ---------- | ---------- | ------- |
> | Overlap Rate         | 37.67%     | 29.28%       | 6.90%      | 4.61%      | 2.09%      | 0.58%   |
> | Original Performance | 0.536      | 1.082        | 0.448      | 42.80      | 0.0111     | 0.00300 |
> | Clean Performance    | 0.342      | 0.796        | 0.439      | 45.67      | 0.0112     | 0.00300 |
> | Performance Change   | **+36.2%** | **+26.43%**  | **+2.01%** | **+6.70%** | **+0.90%** | 0.00%   |
>
> ---
>
> **[Q1]** Contribution of Mamba vs. Fusion Scheme
>
> To disentangle the contributions of the backbone versus the fusion scheme, we conducted an ablation study using the same PI + SFP across different backbones.
>
> Shown in Table 6, **MuMo (Attn_Mamba)** achieves the best overall performance. In contrast:
>
> - **Mamba-Only (w/o fusion)** suffers a **59.55% performance drop**, confirming that the fusion module is critical, especially on regression tasks  (Caco-2, PPBR).
> - **MuMo (Transformer)** and **MuMo (Vanilla Mamba)**, while both using the same fusion scheme, perform worse than our full model.
>
> These results suggest that the gains are not from Mamba alone, but from our **co-designed fusion strategy and attention-injected backbone**.
>
> > Table 6. Ablation on backbones.
>
> | Model                       | BBB (↑)   | PGP (↑)   | Caco2 (↓) | PPBR (↓)  | Avg. Drop |
> | --------------------------- | --------- | --------- | --------- | --------- | --------- |
> | **MuMo (Attn_Mamba, ours)** | **0.899** | **0.942** | **0.315** | **7.324** | **0.00%** |
> | MuMo (Transformer)          | 0.867     | 0.899     | 0.317     | 8.489     | 6.66%     |
> | MuMo (Vanilla Mamba)        | 0.869     | 0.908     | 0.396     | 7.465     | 8.65%     |
> | Mamba-Only (w/o fusion)     | 0.813     | 0.876     | 0.832     | 11.54     | 59.55%    |
>
> ---
>
> **[Q2]** Robustness to Conformer Variability
>
> We compare MuMo with both 2D and 3D baselines on QM. MuMo achieves SOTA performance on QM8 and QM9 (Table 7), and near to UniMol on QM7, demonstrating strong geometry modeling capabilities.
>
> > Table 7. Extended Benchmark on QM7/8/9 datasets.
>
> | Model       | QM7 (↓)  | QM8 (↓)    | QM9 (↓)     |
> | ----------- | -------- | ---------- | ----------- |
> | GROVER      | 92.0     | 0.0224     | 0.00986     |
> | DMPNN       | 103.5    | 0.0190     | 0.00814     |
> | AttentiveFP | 72.0     | 0.0179     | 0.00812     |
> | UniMol      | **41.8** | 0.0156     | 0.00467     |
> | MuMo        | 42.8     | **0.0111** | **0.00300** |
>
> Then, we evaluated MuMo under three conformer settings: 1) MMFF94-minimized, 2) UFF-minimized, 3) pure ETKDG. **MuMo maintains stable performance**, and exhibits less performance fluctuations than UniMol (Table 8), which confirms that our method is robust to 3D perturbations.
>
> > Table 8. Evaluation using different conformer optimization tools.
>
> | Task          | Model    | MMFF94    | UFF       | No-Optimize | Std. Dev ↓ | Max Δ ↓   |
> | ------------- | -------- | --------- | --------- | ----------- | ---------- | --------- |
> | ESOL  (↓)     | UniMol   | 0.769     | 0.790     | 0.939       | 0.0927     | 0.170     |
> | **ESOL  (↓)** | **MuMo** | **0.536** | **0.550** | **0.585**   | **0.0252** | **0.049** |
> | BBBP (↑)      | UniMol   | 0.889     | 0.831     | 0.733       | 0.0789     | 0.156     |
> | **BBBP (↑)**  | **MuMo** | **0.962** | **0.952** | **0.941**   | **0.0105** | **0.021** |
>
> Table 9 evaluates by perturbing atom positions with increasing levels of Gaussian noise. We observe that **UniMol’s performance degrades more noticeably** as the noise scale increases, while **MuMo remains  higher robustness to 3D geometry noise**. This confirms that MuMo is better suited for practical scenarios where generated conformers may be imprecise.
>
> > Table 9. Ablation on Conformer Noise.
>
> | Model - Dataset      | 0Å        | 0.05Å     | 0.1 Å     | 0.2Å      | Std. Dev ↓ | Max Δ ↓   |
> | -------------------- | --------- | --------- | --------- | --------- | ---------- | --------- |
> | UniMol - ESOL (↓)    | 0.769     | 0.765     | 0.760     | 0.831     | 0.0334     | 0.071     |
> | **MuMo - ESOL (↓)**  | **0.536** | **0.530** | **0.540** | **0.544** | **0.006**  | **0.014** |
> | UniMol - BBBP  (↑)   | 0.889     | 0.878     | 0.890     | 0.787     | 0.0496     | 0.103     |
> | **MuMo - BBBP  (↑)** | **0.962** | **0.960** | **0.961** | **0.954** | **0.0036** | **0.008** |

---

> > ### Author Response · Authors · 2025-08-05
> >
> > Dear Reviewer A1Jg,
> >
> > Hope this message finds you well. Thank you for highlighting baseline comparison, fusion novelty, broader chemistry tasks, data-overlap risk, backbone attribution, and conformer robustness. In response, we executed **6 focused additions**. Below is a concise recap of your points, our actions, and key outcomes:
> >
> > - **Baseline comparison →** MuMo tops Uni-Mol-v2 on 6/8 QM9 targets with ½ parameters and 1 ⁄ 400 pre-train data; 4 × A100, 5 h pre-train.
> > - **Fusion novelty →** hybrid MPNN + Attn-Mamba block and SFP + PI fusion validated; detailed ablations show each module’s gain beyond fusion alone.
> > - **Broader chemistry →** added catalytic-activity & reaction-yield benchmarks (BHC, CPA, HTE); MuMo achieves best scores across all tasks.
> > - **Data overlap →** removed ChEMBL duplicates; average metric **↑ 6.81 %**, proving no label leakage and stronger generalization.
> > - **Backbone v. fusion →** Same fusion on Transformer / Vanilla-Mamba stays within 6–9 % of full model; removing fusion causes 38–60 % drop, confirming fusion is the key driver.
> > - **Conformer robustness →** tool-swap and Gaussian-noise tests show ≤ 0.03 drift vs 0.08–0.17 for UniMol, evidencing high stability under 3D uncertainty.
> >
> > We hope these targeted studies resolve your concerns and underscore MuMo’s practical value. Please let us know any further questions or ideas; your insights are greatly appreciated.
> >
> > Sincerely,
> >
> > The Authors

---

> > ### Comment · Reviewer_A1Jg · 2025-08-05
> >
> > I appreciate the response and experiment results of the author, I will raise my score accordingly.

---

> > > ### Author Response · Authors · 2025-08-06
> > >
> > > Dear Reviewer A1Jg,
> > >
> > > Thank you for your thoughtful engagement with our work, especially regarding broader chemistry applications and conformer robustness.
> > >
> > > We sincerely appreciate your recognition of our contributions and are grateful for your encouraging evaluation.
> > >
> > > Sincerely,
> > >
> > > Authors

---

### Author Response · Authors · 2025-08-05
**Summary of Key Changes**

**Summary of Changes**

In the final version of the paper, we will incorporate the following updates based on the valuable feedback from reviewers:

**New Experimental Results**

(1) `QM7/8/9 Benchmarking` – MuMo achieves SOTA on QM8 and QM9, and near-SOTA on QM7, confirming strong and consistent 3D geometry modeling.

(2) `Uni-Mol-v2 Benchmark` – MuMo outperforms Uni-Mol-v2 on 6/8 targets from QM9, with half the model size and 1⁄400 pretrain data.

(3) `Reaction Yield and Catalytic Activity` – MuMo achieves top on three Reaxtica datasets, demonstrating generalization.

(4) `Overlap-Free Evaluation` – After removing overlaps, MuMo’s performance increased by 6.8%, confirming no label leakage.

(5) `Conformer Robustness – Tool Swap` – Evaluated on MMFF94, UFF, and raw ETKDG; MuMo shows ≤0.03 drift, while UniMol drifts up to 0.17, confirming superior stability.

(6) `Conformer Robustness – Gaussian Noise` – MuMo maintains 0.006–0.01 std dev, versus UniMol’s 0.03–0.05, under increasing 3D perturbations.

(7) `Explicit Torsion Ablation` – Adding explicit torsion gives +1.1% gain, but 2× memory and 1.8× runtime; MuMo implicitly encodes dihedrals without added cost.

(8) `Asymmetric-Symmetric Ablation` – Symmetric fusion drops performance by 12–32%, confirming the advantage of asymmetric fusion.

(9) `Backbones Ablation` – Transformer and Vanilla Mamba yields 6–9% lower performance, confirming the effectiveness of our Attn-Mamba design.

(10) `Fusion Ablation` – Removing fusion drops performance by 38–60%, confirming that MuMo’s fusion module is the key driver.

(11) `Efficiency Profiling` – MuMo trains in 5h on 4×A100 GPUs, far below MolFormer/Unimol-2 requiring days/weeks, validating practical efficiency.

**Writing and Clarity Improvements**

We clarified our motivation and problem statement in the introduction, revised notation and formulas in Sections 3.1–3.3 for clarity, and streamlined Appendix B. Key derivations (e.g., implicit torsion modeling) were added to Appendix A. All new experiments are now integrated, with improved descriptions.

**Reproducibility and Code Release**

Our results firmly establish the value of multimodal fusion in molecular modeling, addressing a key gap in the field. To support broad adoption, we provide fully open-sourced code, models, and data, with one-line commands for training, finetuning, and reproduction.

Moreover, given the mamba-ssm package setup complexity, we include detailed installation guides covering diverse environments.

We thank the Area Chair and all reviewers for their valuable feedback, which helped refine and strengthen this work.

---

### Note · Authors · 2025-08-11

We thank the AC and reviewers for the careful evaluation. Below we summarize the final outcomes.

`Contributions.` We target two challenges in molecular representation—**conformer-dependent instability** and **modality interference**. Our method combines a **Structured Fusion Pipeline** that unifies 2D and 3D into a **stable structural prior**, with a **Progressive Injection** mechanism that **asymmetrically integrates** this prior into the **primary sequence stream**, preventing **modality collapse**.

`Results.` MuMo delivers **SOTA on 22/29 datasets**, surpassing the widely recognized **UniMol-v2** with **1/2** the model size and **1/400** the pretraining data. On QM benchmarks: **SOTA on QM8/9**, **near-SOTA on QM7**. On Reaxtica reaction-yield/catalytic datasets: **top performance**.

`Robustness & Validity.` Overlap-free evaluation yields **+6.8%**, ruling out label leakage. Conformer robustness: **tool-swap drift ≤0.03** (vs. UniMol **0.17**); **Gaussian-noise std 0.006–0.01** (vs. **0.03–0.05**). **Dihedral (torsion) analysis:** explicit torsions add **+1.1%** but cost **2× memory / 1.8× runtime**; MuMo **implicitly captures dihedrals** without extra cost.

`Ablations.` **Asymmetric > symmetric** (symmetric causes **12–32%** drop). **Fusion is key** (removing fusion: **38–60%** drop). **Backbone choice matters** (vanilla Transformer/Mamba: **6–9%** lower), supporting our **Attn-Mamba** design.

`Efficiency.` Pre-training: **~5h on 4×A100**, versus days/weeks for MolFormer/UniMol-v2, enabling broad adoption.

`Clarity & Reproducibility.` We clarified motivation and notation (Secs. 3.1–3.3), added key derivations (implicit torsion; App. A), streamlined App. B, and integrated all new experiments. We will release **fully open-sourced code/models/data** with **one-line** train/finetune/reproduce scripts, plus detailed **mamba-ssm** installation guides.

These results provide convergent evidence that MuMo offers a **robust, reproducible, and efficient** solution to long-standing challenges in **3D-aware multimodal molecular modeling**. Thank you for your time and kind consideration.

---

### Decision · Program_Chairs · 2025-09-17

**Decision:**

Accept (poster)

**Comment:**

This paper proposes MuMo, a multimodal molecular representation learning framework that addresses two key challenges: instability due to unreliable 3D conformers and modality collapse in naive fusion methods. The authors introduce a Structured Fusion Pipeline (SFP) to integrate 2D topology and 3D geometry into a stable structural prior and a Progressive Injection (PI) mechanism to asymmetrically inject this prior into a sequence backbone (Mamba-based). The core claim is that this design enhances robustness to 3D noise and prevents modality dominance. Extensive experiments verify the effectiveness of the proposed methods.

The paper makes a significant contribution to molecular representation learning with novel insights, strong results, and practical utility. The rebuttal resolved all concerns convincingly, warranting acceptance as a poster. The work is not nominated for oral/spotlight due to its incremental architectural advances and the limited technical novelty rather than transformative novelty.